# Systematic analyses of lipid mobilization by human lipid transfer proteins

Kevin Titeca[1,2,18], Antonella Chiapparino[2,10,18], Marco L. Hennrich[2,11,18 ✉], Dénes Türei[3,4], Mahmoud Moqadam[5,6], Reza Talandashti[5,6,12], Camille Cuveillier[1], Larissa van Ek[1], Joanna Zukowska[2], Sergio Triana[2,13,14], Florian Echelard[5,6], Inger Ødum Nielsen[7], Mads Møller Foged[7], Charlotte Gehin[2,15], Kliment Olechnovic[8,9], Sergei Grudinin[9], Julio Saez-Rodriguez[3,4], Theodore Alexandrov[2,16,17], Kenji Maeda[7], Nathalie Reuter[5,6] & Anne-Claude Gavin[1 ✉]

Lipid transfer proteins (LTPs) maintain the specialized lipid compositions of organellar membranes[1,2]. In humans, many LTPs are implicated in diseases[3], but the cargo and auxiliary lipids that facilitate the transfer of the majority of LTPs remain unknown. Here we combined biochemical, lipidomic and computational methods to systematically characterize LTP–lipid complexes[4] and measure how LTP gains of function affect cellular lipidomes. We identified bound lipids for around half of the hundreds of LTPs that we analysed, confirming known ligands and identifying new ones across most LTP families. Gains in LTP function affected the cellular abundance of both their known and newly identified lipid ligands, indicating comparable functional relevance of the two ligand sets. Using structural bioinformatics, we characterized mechanisms that contribute to lipid selectivity and identified preferences based on headgroup or acyl chain. We demonstrate some basic principles of how LTPs mobilize their ligands. They commonly interact with several classes of lipids and exhibit broad but selective preference for particular headgroups and for lipid species with shorter acyl chains that contain one or two unsaturated carbons, suggesting that only subsets of lipid species are efficiently mobilized. The datasets represent a resource for further analysis in different cell types and states, such as those associated with pathologies.

Human cells generate thousands of different lipids[5], which constitute the lipidome, whose composition is adapted to cellular needs and contributes to establishing cellular identity and functional specialization[6,7]. All aspects of lipid function rely on their heterogeneous distribution, whereby lipids accumulate locally and define the membranes of specific organelles or microdomains[5]. Maintaining optimal functional membrane composition involves compartmentalized lipid metabolism, which is associated with a variety of lipid sorting and transport systems, which can be provided, among other mechanisms, by LTPs[1,2]. LTPs have diverse structures, but many share a common mode of action: they extract specific lipids from membrane bilayers and load them into a hydrophobic pocket, forming water-soluble protein–lipid complexes that isolate cargoes from the aqueous phase, a step known as lipid mobilization. In addition to their cargo, some LTPs mobilize auxiliary lipids that function as exchange currencies or cofactors[8–13]. They facilitate the uptake or release of cargo, thereby ensuring the directionality of transport and its coupling to metabolism[14,15].

LTP functions are conserved in all kingdoms of life[1]. There are at least 131 LTPs in humans, whose dysfunctions are often associated with diseases[3]. However, in most cases, the identity of cargo and auxiliary lipids remains unknown, limiting our ability to understand how LTPs function in cells and adapt to the state of membrane lipidomes. We have developed methods based on affinity purification and mass spectrometry (AP–MS) to study the mobilization of lipids by LTPs[4], applied them in proof-of-principle studies in a eukaryote model, *Saccharomyces cerevisiae*, and demonstrated the feasibility of systematic analyses of LTP–lipid complexes[16]. We therefore set out to characterize LTP–lipid complexes assembled in humans and to systematically record the consequences of gain of LTP function on the whole-cell lipidome. Owing to its unprecedented scale, covering nine LTP families of divergent

[1]Department of Cell Physiology and Metabolism, University of Geneva, Geneva, Switzerland. [2]European Molecular Biology Laboratory, EMBL, Heidelberg, Germany. [3]Institute for Computational Biomedicine, Faculty of Medicine, Heidelberg University and Heidelberg University Hospital, Heidelberg, Germany. [4]EMBL European Bioinformatics Institute (EMBL-EBI), Wellcome Genome Campus, Hinxton, UK. [5]Department of Chemistry, University of Bergen, Bergen, Norway. [6]Computational Biology Unit, Department of Informatics, University of Bergen, Bergen, Norway. [7]Cell Death and Metabolism group, Center for Autophagy, Recycling and Disease, Danish Cancer Institute, Copenhagen, Denmark. [8]Institute of Biotechnology, Life Sciences Center, Vilnius University, Vilnius, Lithuania. [9]CNRS, Grenoble INP, LJK, Université Grenoble Alpes, Grenoble, France. [10]Present address: AB Sciex Germany, Darmstadt, Germany. [11]Present address: Absea Biotechnology, Berlin, Germany. [12]Present address: Department of Biochemistry, University of Oxford, Oxford, UK. [13]Present address: Institute for Medical Engineering and Science (IMES) and Department of Chemistry, Massachusetts Institute of Technology, Cambridge, MA, USA. [14]Present address: Broad Institute of MIT and Harvard, Cambridge, MA, USA. [15]Present address: École Polytechnique Fédérale de Lausanne (EPFL), Lausanne, Switzerland. [16]Present address: Department of Pharmacology, University of California, San Diego, La Jolla, CA, USA. [17]Present address: DeepCyte, San Diego, CA, USA. [18]These authors contributed equally: Kevin Titeca, Antonella Chiapparino, Marco L. Hennrich. ✉e-mail: marco.hennrich@abseabio.com; anne-claude.gavin@unige.ch

origins and with different folds, the resulting resource captures some general biochemical principles of LTP-mediated lipid mobilization in humans. It represents a useful resource for follow-up structural analyses[13,17,18] and for systematic analyses of LTP-dependent cellular lipid fluxes[19].

## A systematic resource on human LTPs

The lipid-binding properties of LTPs are essential to their function. Here we measured the ability of human LTPs to mobilize specific lipids. We adapted AP–MS methods to characterize human soluble LTP–lipid complexes, and applied them to complementary approaches (Fig. 1 and Supplementary Methods). We measured the ability of affinity-tagged LTPs that were overexpressed in HEK293 cells to associate stably with lipids in a physiological context (Fig. 1a, in cellulo). We also studied the ability of recombinant LTPs expressed in *Escherichia coli* to extract lipids from simplified artificial membranes composed of lipids extracted from bovine liver and porcine brain (Fig. 1a, in vitro).

In total, 101 human LTPs were cloned, and we were able to express 86 in HEK293 cells and 71 in *E. coli* (Fig. 1a and Supplementary Tables 1 and 2). We focused on non-transmembrane box-like LTPs, excluding the few bridge-like LTPs, which mediate bulk lipid transport through long hydrophobic grooves[20]. We successfully purified 110 LTP–lipid complexes assembled in cellulo or in vitro (counting the redundancy of the two screens) by affinity and size-exclusion chromatography (SEC) (Fig. 1a and Extended Data Fig. 1a) (for gel source data see Supplementary Fig. 1a (in cellulo) and 1b (in vitro)). The SEC fractions were analysed by SDS–PAGE and liquid chromatography–tandem mass spectrometry (LC–MS/MS)- or high-performance thin-layer chromatography (HPTLC)-based lipidomics[16] (Supplementary Methods). To filter out non-specific background, we matched LTP abundance in SEC fractions with lipid abundance. Only lipids identified in fractions containing LTPs and showing an elution profile similar to that of LTPs were considered as potential lipid binders (Supplementary Methods and Extended Data Fig. 1a). The work consisted of more than 600 LC–MS/MS runs, and the analysis of the resulting large datasets required the development of semiautomatic pipelines, incorporating quality filtering and manual processing of the spectra (Extended Data Fig. 1b–d and Supplementary Table 3). Notably, the LTP-associated lipidome revealed lipid species that were barely, if at all, detectable in the total lipidome of HEK293 cells (Supplementary Figs. 2 and 3), such as rare, long-chain ceramide species with 46 or 48 carbons (fatty acid plus long-chain base, see below) and ceramide 1-phosphate (Supplementary Table 4). The enrichment for rare and low-abundance lipids, which is unlikely to come from non-specific contamination of the total lipidome, provides an additional level of confidence in the dataset.

We identified lipid ligands for 45 of the LTPs that we could affinity purify (Fig. 1a and Supplementary Table 5). The datasets cover nine of the ten LTP families, with up to ten representatives for the lipocalins[2]. Although most of the LTPs analysed were successfully expressed and purified in HEK293 and *E. coli* systems, only six showed lipid mobilization activity in both assays (Fig. 1a and Supplementary Table 6). For example, the retinol-binding proteins RBP1 and RBP4 formed complexes with vitamin A only in the cellular context, as their assembly requires active metabolism and transmembrane transport systems, which are absent in the simplified in vitro assay[21] (Supplementary Tables 4–6). The class I phosphatidylinositol transfer proteins (PITPs) PITPNA and PITPNB, which are known to transfer phosphatidylinositol and phosphatidylcholine between membranes[22], associated with phosphatidylcholine in both assays, but associated with phosphatidylinositol only in the in vitro biochemical assay (Supplementary Tables 4–6). This discrepancy may reflect the absence of active phosphatidylinositol-related pathways in unstimulated HEK293 cells, such as the phosphatidylinositol cycle that is activated downstream of G protein–coupled receptors[22]. By contrast, in vitro conditions bypass cellular regulatory mechanisms and are not subject to context-dependent limitations in lipid availability. The cell-based analyses concern only the biology of dividing HEK293 cells, in which specific pathways and functions may not be involved. This illustrates that both approaches have specific limitations and complementary capabilities for detecting different sets of LTP–lipid complexes.

## Data quality and functional relevance

The scale of this study is unprecedented, and there were no established strategies for assessing the overall quality of both datasets. We therefore developed our own benchmarks after acquiring and integrating structural and functional data.

The structural benchmark was based on known[23] or predicted[24] structures, from which we estimated the volume of lipid-binding pockets in LTPs[25]. We determined the extent to which the sizes of lipid ligands observed in our assays fitted in these volumes (Fig. 1b,c, Extended Data Fig. 2a, Supplementary Table 7a and Supplementary Methods). For the subset of LTPs with both known and novel ligands, we observed that the known ligands occupied less than 42.5% of the volume of the binding pocket (Fig. 1c), revealing the existence of a 'buffer zone' (region of unfilled space), similar to that observed in the ligand-binding cavity of enzymes[26]. Of note, the volumes occupied by newly discovered LTP ligands did not significantly differ from those of known ligands ($P = 0.252$) (Fig. 1b). Out of the 756 LTP–lipid pairs analysed (277 previously known and 479 novel), only 56 had ligands occupying the buffer zone (Fig. 1c). These included members of the lipocalin (LCN15) and sterol carrier protein (SCP; SCP2D1 and SCP2) families that have particularly small pockets (Supplementary Tables 7a and 8). After excluding these three outliers, the volumes occupied by ligands in vitro were comparable to those in cellulo ($P = 0.132$) (Extended Data Fig. 2c,d).

For the functional benchmark, we assessed the cellular relevance of newly identified LTP–lipid pairs, particularly those from the in vitro assay. To this end, we used LC–MS/MS-based lipidomics to measure how overexpression of each LTP affected the lipidome of HEK293 cells (Fig. 1a and Supplementary Table 7b,c; for gel source data see Supplementary Fig. 1c; Supplementary Methods). We postulated that a gain in LTP function (perturbing the fluxes of its cargoes) would affect the abundance of the respective cargoes or their metabolic products. Overall, 24 individual lipid subclasses were measured consistently across all samples, defining a matrix of 1,032 LTP–lipid pairs, of which 225 (22%) were significantly affected by a gain of LTP function (Fig. 1d). Remarkably, for the subset of LTP–ligand pairs that were identified in both screens, 44% of ligands were affected by overexpression of their respective LTP, which is significantly higher than what is observed when all possible LTP–lipid pairs are considered ($P = 1.05 \times 10^{-5}$; Fig. 1d). The overexpression of the corresponding LTPs affected the abundance of newly discovered ligands as frequently as that of previously known cargoes (47% versus 40%, respectively), indicating comparable functional relevance of both ligands sets (Fig. 1d). In addition, a significant fraction of the ligands identified in the in vitro assay were supported by overexpression data (36%; $P = 2.35 \times 10^{-2}$; Fig. 1d), indicating that the in vitro approach yields functionally relevant LTP–ligand pairs. This is somewhat lower than for the in cellulo assay (58%; $P = 5.23 \times 10^{-7}$), suggesting that some LTP–ligand pairs identified in vitro may not assemble in HEK293 cells. Among them, we identified lipids abundant in *E. coli*, including odd-chained species[27], and phosphatidylglycerol (Extended Data Fig. 2b), suggesting that some may have been mobilized during expression in *E. coli*[16]. For some LTPs, this pre-loading with bacterial lipids may have limited their capacity to mobilize other lipids in the in vitro assay. Nevertheless, these data indicate that lipids of bacterial origin can be mobilized, whether in a heterologous expression system or in an in vitro binding assay, thus providing relevant information on the specificity and structural nature of LTP binding pockets. To facilitate follow-up studies, we provide a detailed documentation of the results of this benchmark in Supplementary Table 8.

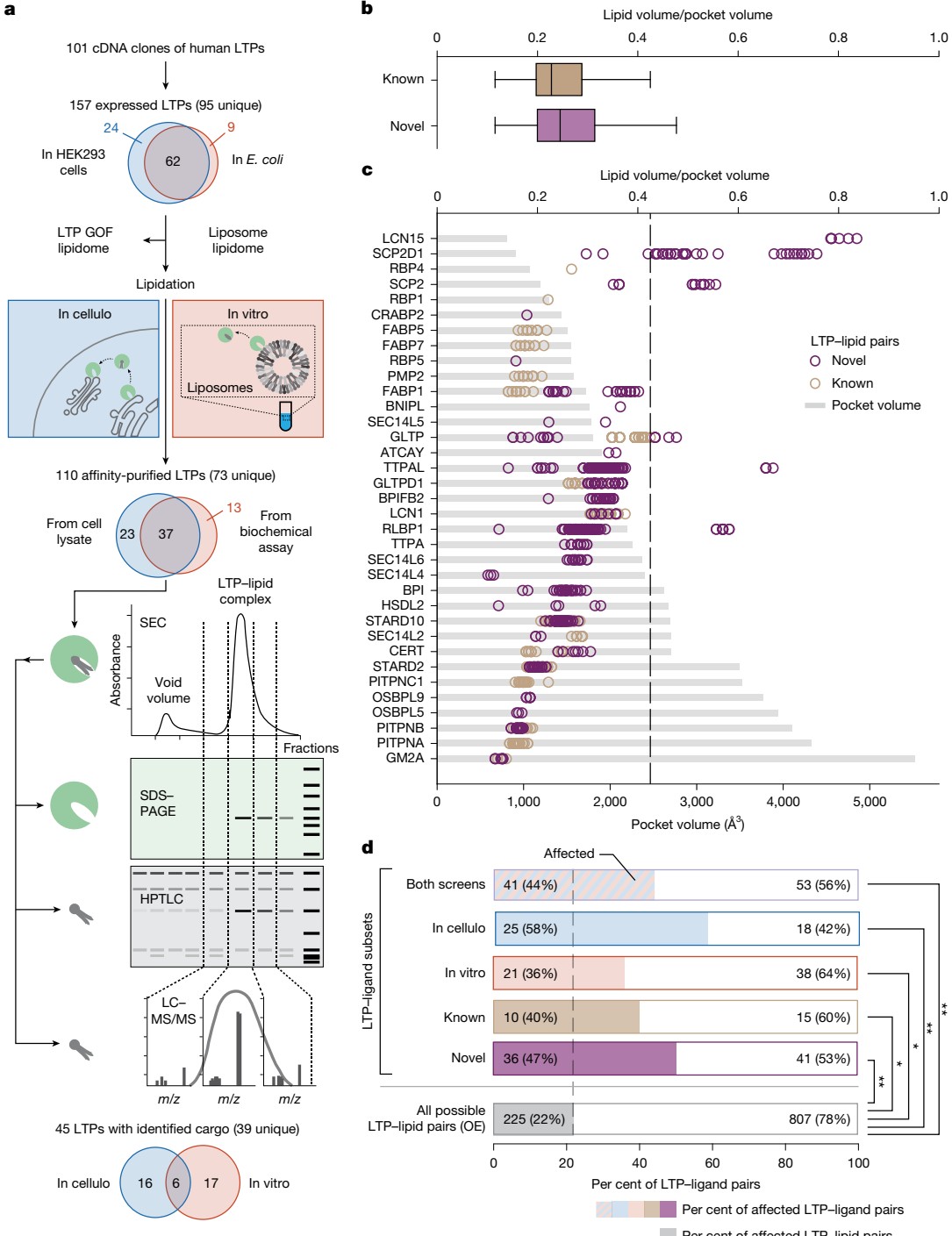

**Fig. 1 | A resource on human LTPs. a**, Overview of the experimental approaches. GOF, gain of function. Created in BioRender; Gavin, A. C. https://biorender. com/0mht878 (2025). **b,c**, Structural benchmark of 35 LTPs for which ligands were identified by LC–MS/MS. OPSBP2, OSBP, OSBPL1A and OSBPL2 are absent: their ligands (sterols and phosphatidylinositol phosphates) were identified only by HPTLC, providing no information on the molecular species. **b**, Subset of LTPs with both known (213 pairs) and novel ligands (173 pairs). The x axis shows the ratio of lipid species volume and pocket volume. The two distributions overlap; Welch's two-sided t-test, P = 0.252 (no statistical difference). The centre line is the median, box limits denote first and third quartiles and whiskers extend to the furthest data point within 1.5 times the interquartile range. **c**, Distribution of the ratio of lipid species volume and LTP pocket volume (top x axis) for all LTP–ligand pairs analysed (n = 756). The bottom axis represents the LTP pocket volume. The dashed line indicates the maximal ratio (0.425) of lipid species to pocket volumes observed for the known LTP–ligand pairs. **d**, Functional benchmark. The x axis represents the fraction of LTP–ligand pairs for which overexpression (OE) of the LTP in HEK293 cells led to a significant change in the corresponding ligand (lipid subclass) (Welch's two-sided t-tests, Bonferroni correction for multiple testing). n = 3 biological replicates. For all possible LTP–lipid subclass pairs, tests account for variations in lipid subclasses across all overexpression experiments, downweighting those affected by multiple LTPs. A lipid subclass is considered affected if at least one species in that subclass is affected. The y axis represents the LTP–ligand subsets that were analysed. The grey bar at the bottom shows the affected fraction of all possible combinations of LTP–lipid pairs (from the overexpression dataset—that is, all subclasses of lipids seen (24) by all overexpressed LTPs (43)). Fisher's exact test. *P ≤ 0.05, **P ≤ 10⁻⁵; both screens, P = 1.05 × 10⁻⁵; in cellulo, P = 5.23 × 10⁻⁷; in vitro, P = 2.35 × 10⁻²; known, P = 4.74 × 10⁻²; novel, P = 4.39 × 10⁻⁶.

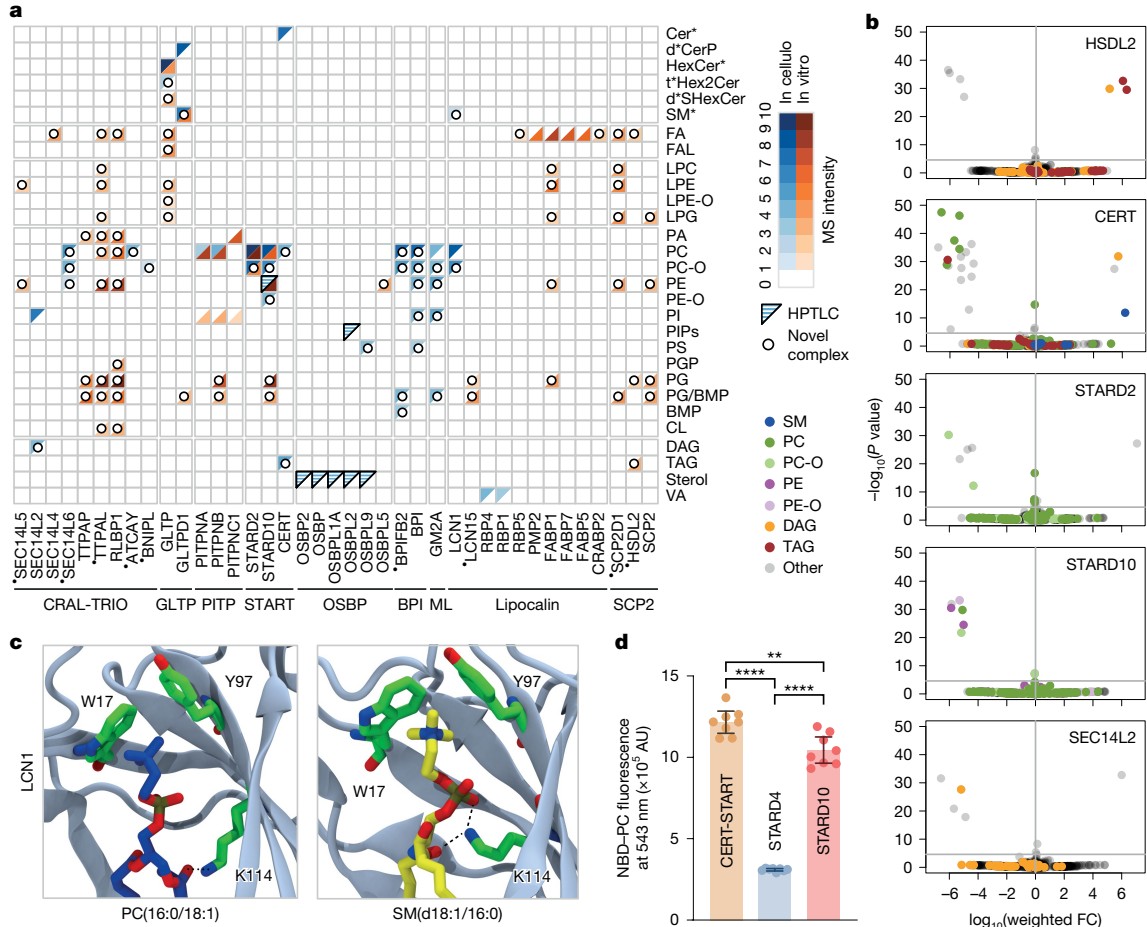

**Fig. 2 | Novel ligands for most LTP families. a**, Normalized mass spectrometry (MS) intensity (minimum–maximum scaling) of all observed LTP–lipid pairs (lipid subclass level). Ionization efficiencies vary between lipid classes, allowing only qualitative comparisons. Previously orphaned LTPs are indicated with dots. HPTLC-based observations are represented when mass spectrometry data were unavailable. LTPs on the *x* axis have been seriated (Extended Data Fig. 3a). **b**, Effect of overexpression of selected LTPs on the total HEK293 lipidome. *n* = 3 biological replicates. Statistics define weighted fold change (FC) in the lipid species abundance across all overexpression data and associated *P* values. Welch's two-sided *t*-test corrected for multiple testing (Bonferroni). Horizontal grey lines denote *P* = 0.05. Each dot represents an individual lipid species. **c**, Snapshots from molecular dynamics simulations showing phosphatidylcholine (PC(16:0/18:1); left) or sphingomyelin (SM(d18:1/16:0); right) bound to LCN1 and highlighting the aromatic residues involved in cation–π interactions with the choline moieties (W17 and Y97). Hydrogen bonds between lipid and LCN1 are represented with dashed black lines. *n* = 3 independent simulations. **d**, Fluorescence-based binding assay of CERT lipid transfer domain (CERT-START), STARD4 and STARD10 to NBD–phosphatidylcholine (NBD–PC), presented (in arbitrary units (AU)) as the

emission (at 543 nm) of fluorescence of the NBD group excited at 470 nm. STARD10 is a known phosphatidylcholine binder; STARD4 binds cholesterol and not phosphatidylcholine. *n* = 8 independent experiments. Data are mean ± 95% confidence interval. Welch's ANOVA followed by two-sided Dunnett's post hoc multiple comparison test. **\*\****P* = 0.053, **\*\*\*\****P* ≤ 0.0001. BMP, bis(monoacylglycero)phosphate; Cer, ceramide; CerP, ceramide 1-phosphate; CL, cardiolipin; DAG, diacylglycerol; FA, fatty acid; FAL, fatty acid aldehyde or alcohol; Hex2Cer, dihexosylceramide; HexCer, hexosylceramide; LPC, lyso-phosphatidylcholine; LPE, lyso-phosphatidylethanolamine; LPE-O, ether-linked lyso-phosphatidylethanolamine; LPG, lyso-phosphatidylglycerol; PA, phosphatidic acid; PC, phosphatidylcholine; PC-O, ether-linked phosphatidylcholine; PE, phosphatidylethanolamine; PE-O, ether-linked phosphatidylethanolamine; PG, phosphatidylglycerol; PGP, phosphatidylglycerol-phosphate; PI, phosphatidylinositol; PIPs, phosphatidylinositol phosphates; PS, phosphatidylserine; SHexCer, sulfated hexosylceramide; SM, sphingomyelin; TAG, triacylglycerol; VA, vitamin A. Sphingolipid species notation (**a**, top right): d*, dihydrosphingolipid or sphingolipid; t*, phytosphingolipid, dihydrosphingolipid with OH on fatty acid or sphingolipid with OH on fatty acid; *, combination of all species.

## The LTPs interactome reveals new ligands

We identified new ligands for 72% of the LTPs that we analysed, including 9 LTPs with previously unknown cargoes or auxiliary lipids, and ligands that were not known to be part of the LTP system (Fig. 2a). Notably, in the gain-of-function experiments (assessed at the lipid subclass level), we observed a change in the cellular abundance of the corresponding ligands for 36 out of the 77 new LTP–lipid pairs that were detectable, providing further evidence of a physiological role (Fig. 1d and Supplementary Table 8). We conducted additional experiments and structure-based analyses to provide new insights into LTP biochemistry.

HSDL2 is a mitochondrial and peroxisomal orphan LTP whose dysfunction has been implicated in human disorders, leading to impaired neutral lipid storage through mechanisms that remain poorly understood[28]. In vitro, we found that HSDL2 is capable of mobilizing triacylglycerol, a novel ligand in the context of LTP-mediated lipid transport (Fig. 2a and Supplementary Table 5). We confirmed the relevance of this interaction in a cellular context, in which overexpression of HSDL2 resulted in significant increases in the abundance of several species of triacylglycerol and diacylglycerol (a metabolite of triacylglycerol) (Fig. 2b and Supplementary Table 7c). As HSDL2 is part of a protein network consisting of proteins involved in mitochondrial and peroxisomal β-oxidation (Extended Data Fig. 3b,c),

it is likely that HSDL2–triacylglycerol complexes have a role in this process.

Diacylglycerol is another example of a previously unrecognized ligand in the LTP system. In cellulo, we found it in complex with SEC14L2, a lipid-presenting chaperone for several lipid kinases, such as the phosphatidylinositol 4-kinase[14], the phosphatidylinositol 3-kinase or an as yet unidentified alpha-tocopherol (vitamin E) kinase[15]. In cellulo, SECL14L2 formed complexes with the expected substrate, phosphatidylinositol, but also with some diacylglycerol species (Fig. 2a). Supporting this interaction, overexpression of SEC14L2 in HEK293 cells resulted in a significant decrease in levels of cellular diacylglycerol species (Fig. 2b and Supplementary Table 7c). Of note, diacylglycerol is the substrate for a family of ten diacylglycerol kinases (which produce phosphatidic acid). These kinases may represent downstream effectors or potential new clients for SEC14L2 lipid chaperone activity.

Some LTPs that are known to mobilize phosphatidylcholine and phosphatidylethanolamine also mobilized the corresponding phospholipid with ether-linked fatty acids, both in vitro and in cellulo (Fig. 2a). Although structurally similar to their ester-linked counterparts, ether lipids are synthesized via a distinct pathway, exhibit unique biological functions and are predominantly trafficked through non-vesicular mechanisms, probably involving an as yet uncharacterized LTP system[29,30]. Two STARD proteins, STARD2 and STARD10, bind phosphatidylcholine, and phosphatidylcholine and phosphatidylethanolamine, respectively, as well as their ether species (Fig. 2a). Molecular simulations supported the notion that STARD2 can bind to both ether- and ester-phosphatidylcholine (Extended Data Fig. 4a), with both ligands reducing the flexibility of the STARD2 gate region (Extended Data Fig. 4b and Supplementary Fig. 4), as observed for ligands in other STARD proteins[17,31]. Consistently, overexpression of STARD2 and STARD10 in HEK293 cells significantly affected not only phosphatidylcholine and phosphatidylethanolamine ester but also ether species (Fig. 2b and Supplementary Table 7C). Our analyses revealed six additional LTPs that were capable of mobilizing ether lipids (Fig. 2a), providing valuable insights for future studies investigating ether lipid fluxes and the organelles involved in their transport.

Next, we used this resource as a starting point for molecular dynamics simulations to define the molecular determinants of lipid specificity. For example, three LTPs—LCN1 (a lipocalin), STARD2 and STARD10—have distinct specificities for lipids with a choline headgroup. LCN1, the major lipid-binding protein in tears (a fluid with phosphatidylcholine and sphingomyelin[32]), bound to both sphingomyelin (another novel lipid in LTP system) and phosphatidylcholine (the previously known cargo) in cellulo (Fig. 2a). By contrast, STARD2 and STARD10 were unable to mobilize sphingomyelin, either in cellulo or in vitro, but bound to phosphatidylcholine[33] (Fig. 2a). Using molecular dynamics simulation, we identified two aromatic amino acids in the hydrophobic cavity of LCN1 that form the binding site for the choline headgroup, where phosphatidylcholine and sphingomyelin were housed in an elongated conformation (Fig. 2c and Extended Data Fig. 5a). Mutation of these residues to alanine in simulations prevented phosphatidylcholine from establishing enthalpically favourable interactions (important for specificity) within the pocket (Extended Data Fig. 5b). By contrast, the STARD2 binding site[34] imposed a bend of the phosphatidylcholine headgroup, with the conserved phosphate-binding arginine (Arg78) positioned deeper in the pocket than the choline-binding aromatics (Extended Data Fig. 4a), a conformation that is unlikely to be adopted by the sphingosine backbone of sphingomyelin. The shape of the binding site defined specificities for lipid ligands that share chemical similarities but differ in structural flexibility.

## LTPs bind to several classes of lipids

Some members of the STARD, the oxysterol-binding protein (OSBP), PITP or CRAL-TRIO families are known to bind several lipid classes representing cargoes, but also auxiliary lipids[8–13]. Our analyses, covering 9 evolutionarily and structurally distinct families revealed that 25 out of the 39 LTPs that we studied could form complexes with more than one class of lipids (Fig. 3a,b).

To reinforce this observation, we have integrated relevant and systematic lipidomics datasets (Extended Data Fig. 6). We postulated that lipid pairs—that is, those co-mobilized by the same LTPs—if relevant, should share biological relationships. Therefore, we evaluated the links of these pairs in independent lipidomics datasets—specifically, whether they exhibit co-regulation and co-localization more frequently than random lipid pairs. To this end, we integrated lipidomics data obtained after systematic knockdown of enzymes involved in sphingolipid metabolism[6] (Extended Data Fig. 6a) and scored all lipid pairs according to their co-regulation. In this scoring system, '+1.0' denotes perfect co-regulation (lipids always change abundance in the same way) and '−1.0' denotes mutual exclusion (lipids always change in an opposite manner). We also exploited a large spatial metabolite database, METASPACE, which consists of thousands of tissue sections analysed by imaging mass spectrometry (Extended Data Fig. 6b). Each pixel in these images corresponds to one mass spectrometry analysis[35] (Supplementary Methods). Finally, we also included a dataset capturing lipid co-localization at subcellular levels, based on the lipidome of affinity-purified organelles[36] (Extended Data Fig. 6c and Supplementary Table 9a). For all lipid pairs in these datasets, we determined their co-occurrence by the Manders' overlap coefficient (Extended Data Fig. 6b,c and Supplementary Methods). In this system, '+1.0' and '0.0' represent lipid pairs that always or never co-localize, respectively. Of note, we observed that the lipid pairs mobilized by the same LTPs are significantly more co-regulated after metabolic perturbation (Extended Data Fig. 6a) and co-localized more (Extended Data Fig. 6b,c) than expected from random sets of lipid pairs (Supplementary Table 9b). These trends remained significant when only lipid classes were considered—that is, when closely related lipid species or subclasses were excluded from the analysis. Overall, this shows that lipids that are co-mobilized by the same LTP are also often biologically linked.

Among the LTPs that can mobilize more than one class of ligands, the sphingolipid transporters ceramide transporter (CERT) and glycolipid transfer protein domain containing 1 (GLTPD1) are noteworthy. Even for these widely studied LTPs, we have identified new ligands. CERT is known to transfer ceramides from the endoplasmic reticulum to the Golgi[37]. In in silico simulations, phosphatidylcholine formed stable complexes with the steroidogenic acute regulatory transfer (START) domain of CERT, acting as a cofactor facilitating the release of ceramide[13]. In the in cellulo assay, we detected CERT complexes not only with ceramide, but also with phosphatidylcholine and triacylglycerol (Fig. 3a and Supplementary Table 4). Using a fluorescence emission shift assay, we confirmed that CERT can bind to phosphatidylcholine in vitro (Fig. 2d and Supplementary Table 7e). Furthermore, overexpression of CERT resulted in a significant increase in sphingomyelin and diacylglycerol (both produced by the transfer of the phosphocholine headgroup from phosphatidylcholine to ceramide) and a significant decrease in phosphatidylcholine and triacylglycerol species (Fig. 2b and Supplementary Table 7c). Mobilization of phosphatidylcholine and triacylglycerol by CERT does not necessarily imply their transport, but may instead facilitate ceramide uptake and/or release[13]. These metabolites are involved in the further conversion of ceramide[38,39]. Their mobilization by CERT could couple the transport of toxic ceramide with its conversion into sphingomyelin[38]—or perhaps into a form of ceramide storage in lipid droplets (acylceramide)[39].

GLTPD1 is known to transfer ceramide 1-phosphate from the Golgi to the plasma membrane[40]. We found that GLTPD1 formed complexes not only with its known cargo, ceramide 1-phosphate, but also with sphingomyelin both in vitro and in cellulo (Fig. 3a). Similar complexes have been observed in vitro between a plant orthologue of GLTPD1 (ACD11) and sphingomyelin[41], and here we report the assembly of

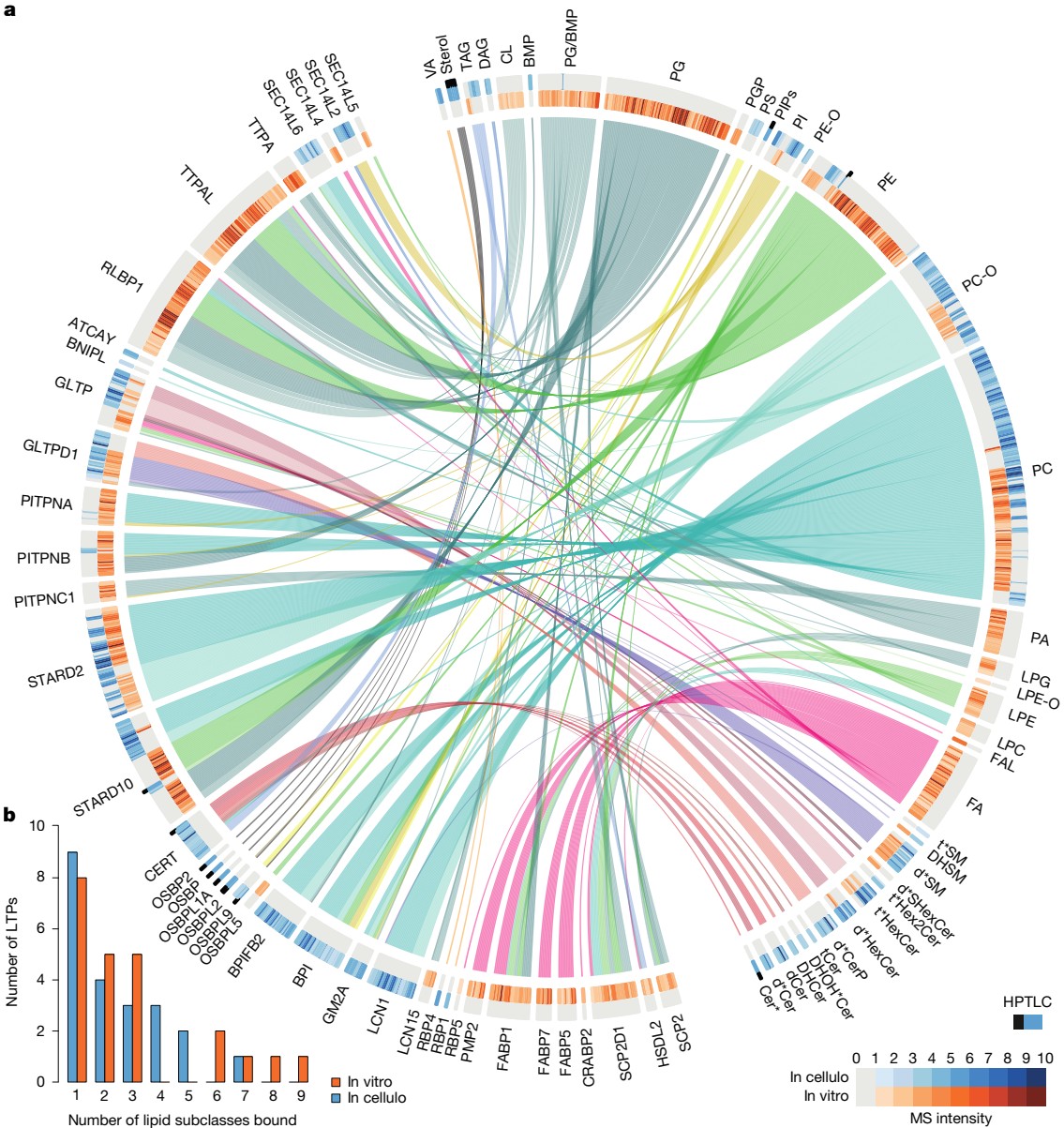

**Fig. 3 | Most LTPs mobilize multiple lipid subclasses, and most lipid subclasses are mobilized by multiple LTPs. a**, Lipids co-mobilized by the 39 LTPs analysed in the in cellulo and in vitro screens. All LTPs (left half) are linked to individual lipid species they mobilized (right half). The LTPs have been seriated as in Fig. 2a (Extended Data Fig. 3a). Individual lipid species are grouped into subclasses. HPTLC-based observations are only represented when no mass spectrometry data was available (black squares on top of blue squares).

Heat maps at the periphery indicate the normalized intensity (minimum–maximum scaling) of lipid species observed. **b**, Distribution of LTP multiple lipid-binding capacity in cellulo and in vitro. Sphingolipid species notation (along perimeter in **a**): DH, dihydrosphingolipid; t, phytosphingolipid; d, sphingolipid; DHOH*, phytosphingolipid or dihydrosphingolipid with OH on fatty acid.

GLTPD1–sphingomyelin complexes in cellular context. Our results have motivated recent analyses, based on complementary cell-based methods, showing that a loss of GLTPD1 function affected retrograde sphingomyelin transport[19].

Overall, this confirms the idea that multi-lipid binding is a common feature of many LTPs, revealing new functional links between lipids and possible regulatory mechanisms linking transport to metabolism.

## Properties of the LTP-mobilized lipidome

Although it is well known that LTPs can recognize specific lipid headgroups, the importance of acyl chain size and saturation in defining binding specificity has remained largely understudied. This is owing to the difficulty of testing large numbers of lipid species in classical biochemical assays. Here we have studied the lipid-binding properties of many LTPs, testing an unprecedented variety of lipid species, encompassing whole lipidomes. We investigated whether the lipids recognized by the analysed LTPs differed from the total lipidome, and specifically, whether the LTP system showed preferences for certain lipid attributes. To achieve this, we profiled the total lipidome of the liposomes (artificial membranes) used in the biochemical assays and the lipidome of HEK293 cells grown under the same conditions as for the AP–MS experiments (Supplementary Table 10a,b). We then compared these reference lipidomes with the lipidome mobilized by LTPs in cellulo or in vitro, respectively, to identify patterns of selective enrichment or divergence.

The LTPs studied in both screens preferentially mobilized glycerophospholipids with shorter fatty acids, whereas mobilized

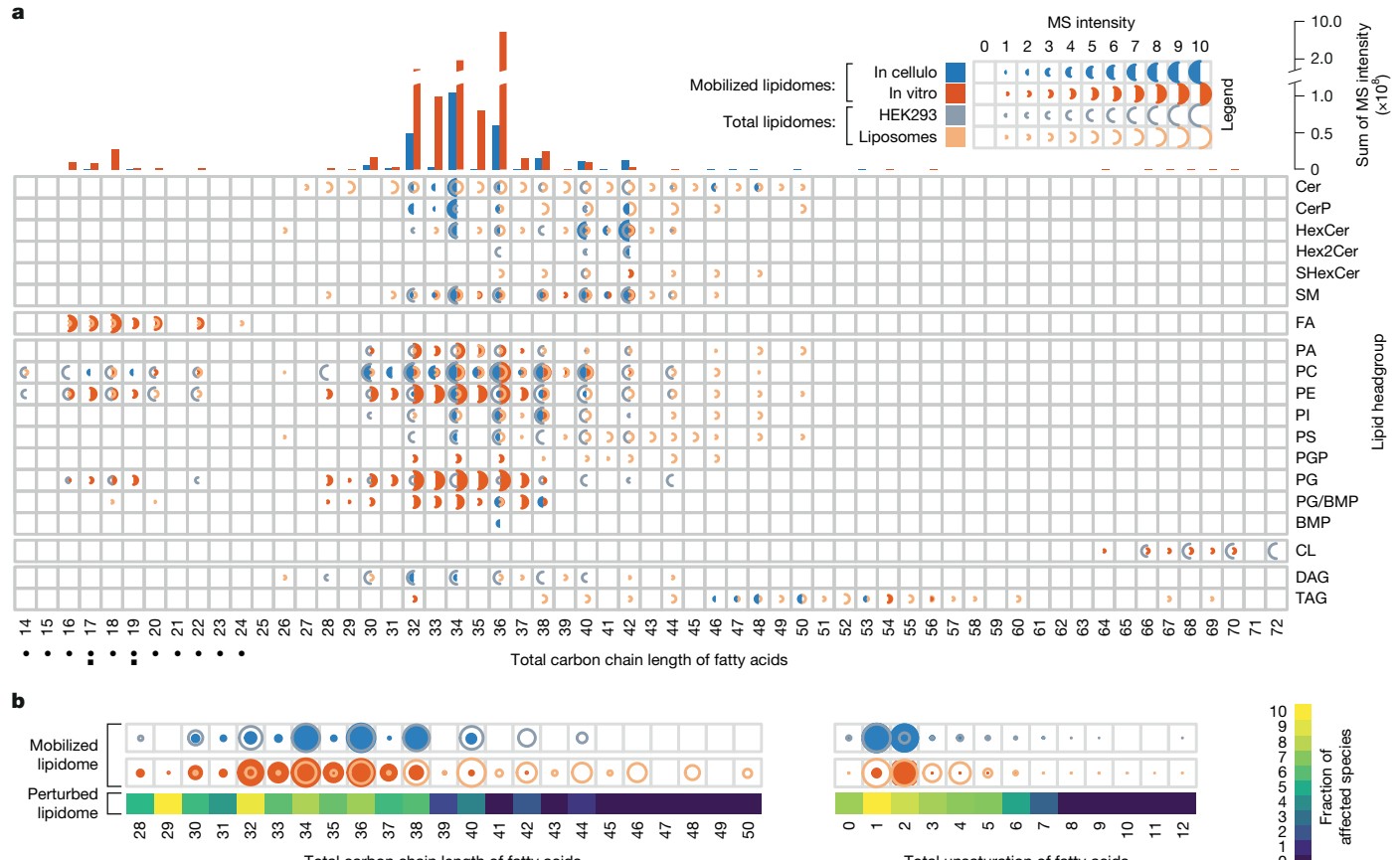

**Fig. 4 | The LTP system preferentially mobilizes lipids with shorter fatty acids bearing one or two unsaturated carbons. a**, Properties of the LTP-mobilized lipidome. Top, distribution of lipid carbon chain lengths. The *y* axis shows the sum of the mass spectrometry intensities (minimum–maximum scaling) of all lipids with these total chain lengths. Bottom, distribution of normalized intensities for observed combinations of head groups and total carbon chain lengths. Comparison of lipid species distributions mobilized in cellulo (blue filled half-circles) and in vitro (orange filled half-circles) with those of the HEK293 cell (grey empty half-circles) or liposomes (orange empty half-circles) lipidomes. Glycerophospholipid species are represented (bottom left) as black circles (lyso-glycerophospholipids) or squares

(glycerophospholipids). **b**, Glycerophospholipids with shorter and mono-saturated or bi-saturated fatty acids are predominantly mobilized by LTPs and affected by LTP overexpression in HEK293 cells. Distribution of glycerophospholipid species according to the total carbon chain length (left) or unsaturation (right) of their fatty acids. Top two rows, glycerophospholipids in the LTP-mobilized lipidomes (filled circles) and the total lipidomes (empty circles). Normalized intensities and legend as in **a**. Bottom row, normalized fraction of species (minimum–maximum scaling) with indicated chain length (left) or number of unsaturated carbons (right) significantly affected by overexpression of LTPs in HEK293 cells (based on paired two-sided *t*-test of induced versus non-induced samples) (Supplementary Table 7d).

sphingolipids had more complex selectivity patterns (Fig. 4a). The shorter fatty acids might facilitate extraction from membranes because of their reduced lateral hydrophobic interactions. LTPs also showed a preference for glycerophospholipids and sphingolipids bearing one to two sites of unsaturation in their fatty acids (that is, carrying one or two double bounds on the fatty acid) (Fig. 4b). These lipid species can cause deep membrane defects, a phenomenon that can contribute to the membrane lipid uptake[42]. By contrast, polyunsaturated fatty acids or fully saturated fatty acids cause only superficial or no defects, respectively, and may be more difficult to extract from membranes[42]. In line with these observations, we found that lipids affected by LTP overexpression showed similar trends (Fig. 4b and Supplementary Table 7d). Changes were enriched in glycerophospholipids species with shorter and mono-saturated or bi-saturated fatty acids. This supports the notion that aliphatic chain lengths and saturation define lipid pools that are differentially accessible to the LTP system.

## Discrete specificities for acyl chains

We also observed examples of LTPs whose lipid preferences deviated from the general trend described above, notably CERT and class I PITPs. The presence of specific fatty acids in their cargoes, ceramide

and phosphatidylinositol, is known to define pools with distinct functions[22,43,44]. The enzymes involved in the metabolism of these lipids can exhibit acyl chain specificity, thus contributing to the formation and maintenance of these pools[42,45], but whether LTPs share similar attributes remains largely understudied.

In mammals, ceramides are synthesized by six ceramide synthases, each producing species with specific fatty acids[46] that are subject to distinct metabolic fates[44]. In vitro, CERT is known to be specific for ceramides containing 14–20 carbon fatty acids[47], defining a pool of ceramide for sphingomyelin synthesis, whereas ceramides containing 22–26-carbon fatty acids are not transported by CERT and are destined for hexosylceramide synthesis[38,48]. Our data showed that CERT–ceramide complexes assembled in cellulo had a similar selectivity for short- and medium-chain ceramides, but not for long-chain ceramides (Fig. 5a and Supplementary Table 4). Sphingomyelin and ceramide 1-phosphate species associated with GLTPD1 mostly had 14–20 carbon fatty acids, while GLTP-associated hexosylceramides had mainly 22–26-carbon fatty acid (Fig. 5a). We observed that a gain of CERT function led to an increase in sphingomyelin levels in HEK293 and in HeLa cells, showing that this effect is conserved in different cell types (Fig. 2b, Extended Data Fig. 7a and Supplementary Tables 7b and 11). Although anticipated, these findings validated the capacity of our approach to

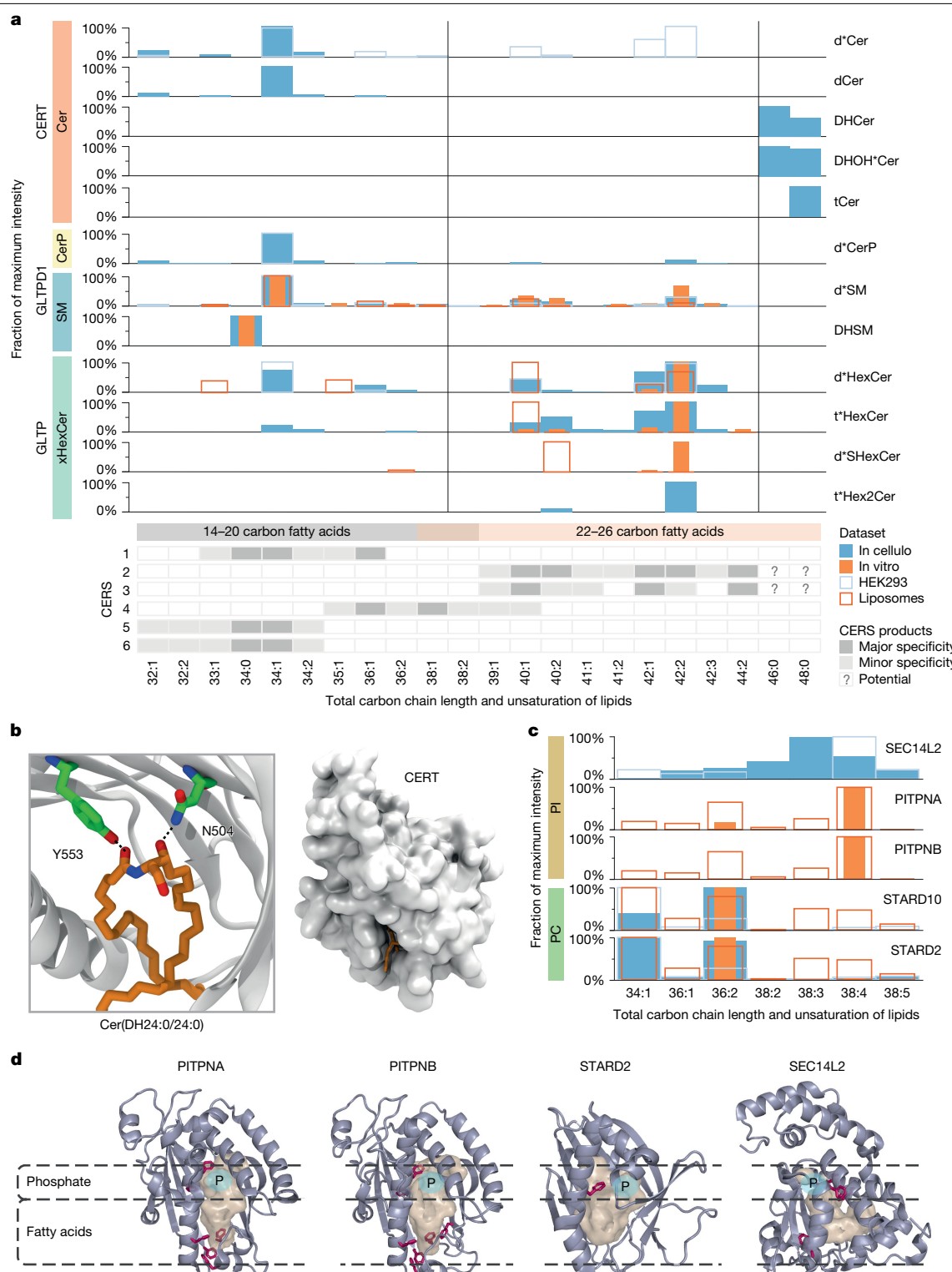

**Fig. 5 | CERT and Class I PITPs have preferences for discrete lipid species.**
**a**, CERT mobilized very long saturated dihydroceramides and phytoceramides.
Top, in cellulo and in vitro distribution of sphingolipid species mobilized by
CERT, GLTPD1 and GLTP, and comparison with those present in the whole
HEK293 cells or liposomes lipidomes. The intensities of each individual species
(within a subclass) were normalized to that of the most abundant species
measured for that subclass and for the corresponding LTP. The data for all
hexosyl-containing sphingolipids were integrated into xHexCer. Bottom,
known fatty acid specificity of ceramide synthase (CERS)[46]. **b**, Right, snapshot
from molecular simulations showing that a very long dihydroceramide,
Cer(DH24:0/24:0) (orange sticks) is fully buried in the hydrophobic cavity of

the START domain of CERT (surface representation). Left, amino acids N504
and Y553 (green sticks) of the START domain of CERT (in cartoons) interact with
the lipid headgroup via hydrogen bonds (close up view). $n = 3$ independent
simulations. **c**, Comparison of the lipid species bound to PITPNA and PITPNB
with those bound to SEC14L2, STARD2 and STARD10. The intensities of individual
species were normalized as in **a**. Colour scheme as in **a**. **d**, Visualization of
the structures of human PITPNA (Protein Data Bank (PDB): 1UW5), PITPNB
(AlphaFold model), STARD2 (PDB: 7U9D) and SEC14L2 (PDB: 4OMJ) highlighting
the presence of a phenylalanine signature (in magenta) in the fatty acid-binding
region of PITPNA and PITPNB but not in STARD2 and SEC14L2. The position of
the phospholipid phosphate group (P) is highlighted in cyan.

recover the characteristic lipid species associated with each system. Notably, we also observed CERT in complex with saturated and very long dihydroceramides and phytoceramides with 46 and 48 carbons (Fig. 5a and Supplementary Table 4), each representing a mixture of species with the same total chain length and number of unsaturated carbons, but differently distributed over their fatty acids (22-, 24- and 26-carbon sphingoid bases and 26-, 24- and 22-carbon fatty acids) (Extended Data Fig. 7b and Supplementary Fig. 2). These species were not observable in total HEK293 lipidomes and were not often recorded in spectral libraries, suggesting that they are rare low abundant species (Extended Data Fig. 7c). Molecular dynamics simulation of the START domain of CERT in aqueous environments, in the presence of very long dihydro- or phyto-ceramide, demonstrated that the hydrophobic cavity of this domain can accommodate the long lipid tails of both types of cargo while maintaining the positioning of their headgroups within the known ceramide-binding site (Fig. 5b and Extended Data Fig. 7d). These findings support a role for CERT in the sorting of very long-chain dihydroceramides, which are thought to contribute structurally to the maintenance and stability of ordered membrane microdomains[49].

The phosphatidylinositol cycle replenishes the plasma membrane with phosphatidylinositol after phospholipase C-mediated PtdIns(4,5)$P_2$ hydrolysis. The phosphatidylinositol species involved in this cycle contain a stearoyl (C18:0) and a polyunsaturated arachidonoyl (C20:4) chain[22,43]. We observed that members of the class I PITP family—PITPNA and PITPNB—preferentially bound arachidonoyl-containing phosphatidylinositols, PtdIns(38:4), in vitro (Fig. 5c and Extended Data Fig. 8a). A smaller fraction of PtdIns(36:2) present in the total lipidome was mobilized by PITPNA, but not seen bound to PITPNB. By structure-based analyses, we identified a conserved cluster of several aromatic amino acids (mainly phenylalanines) located at the bottom of the PITPs binding pocket, close to the gate and well positioned to form an interaction site with arachidonic acid unsaturated carbons (Fig. 5d and Extended Data Fig. 8b). Notably, the selectivity of a cytosolic phospholipase A2 for arachidonoyl-containing lipids is known to involve a similar group of phenylalanines interacting with the four double bonds of arachidonic acid[50]. Such a cluster was not found in the lipid-binding site of SEC14L2 (Fig. 5d), another phosphatidylinositol-binding protein[51], which binds a wide range of different species (Fig. 5c and Extended Data Fig. 8a). It was also absent from STARD2 (Fig. 5d) and STARD10 (not shown), which are structurally related to PITPs and also bound—although marginally—to arachidonoyl-containing phosphatidylcholine, but also many other species (Fig. 5c). Several phosphatidylinositol cycle enzymes, known to exhibit C20:4 preference, contribute to the maintenance of lipids with this acyl chain in the cycle[45]. Our data show that this may also apply to the LTP system.

## Discussion

Understanding how LTPs function in cells and how their activities can adapt to the state of membranes and lipidomes remains a challenge that requires, among other things, understanding how LTPs operate at the molecular level. The capacity of LTPs to mobilize membrane lipids is essential to their cellular function, as this involves specific cargoes, as well as regulatory auxiliary lipids. The widespread ability to bind to multiple classes of lipids suggests that these lipid-induced regulatory mechanisms are common. However, how a single LTP selectively mobilizes lipids of diverse structure remains poorly understood, as do the consequences of these interactions on the metabolic fate of the cargo. By defining individual LTP–lipid and lipid–lipid pairings, our work provides a foundation for future biophysical, molecular dynamics simulation and cell biology studies aimed at addressing these questions[13,18] and should motivate the extension of these approaches, for example, to different cell types or states.

We demonstrate the feasibility of systematic studies of human LTPs, illustrate how we can integrate large-scale data and adapt concepts from systems biology to LTPs. We have just begun to study the consequences of these interactions in a cellular context, by analysing the effect of LTP gain-of-function on cellular lipidomes. The study of the biological functions of LTPs, through systematic exploration of the consequences of their perturbation on organelle function, lipid trafficking and metabolic fate, will be the next challenge to be addressed, and we believe that this resource will serve as a basis for such studies.

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

## Reporting summary

Further information on research design is available in the Nature Portfolio Reporting Summary linked to this article.

## Data availability

The lipidomics data can be downloaded from https://www.ebi.ac.uk/metabolights/MTBLS9567. Organelle lipidomics data are available from ref. 36. The Source Data are organized as followed: molecular biology (Supplementary Table 1), LTP expression (Supplementary Table 2), mass spectrometry fragmentation (Supplementary Table 3), lipid species bound to LTP (Supplementary Table 4), lipid subclasses bound to LTP (Supplementary Table 5), LTPs lipidated in only one of the assays (Supplementary Table 6), results of the structural and functional benchmarks (Supplementary Tables 7 and 8), functional relationship of lipids co-mobilized by the same LTP (Supplementary Table 9), lipidomic data of bovine liver and brain extracts and HEK293 cells (Supplementary Table 10) and lipidomics of CERT-overexpressing HeLa cells (Supplementary Table 11). Supplementary Tables used to produce figures are summarized in Supplementary Table 14. Gels, SEC profiles and western blots are provided in Supplementary Fig. 1. External datasets analysed (but not generated) in this work are summarized in Supplementary Table 15.

## Code availability

The codes generated are accessible at https://github.com/saezlab/lipyd and https://github.com/krtiteca/ScriptsAssociatedWithLTP-Article. The voronota-pocket script is available at https://github.com/kliment-olechnovic/voronota/blob/master/voronota-pocket. The in house code based on the MDAnalysis package is available at https://github.com/reuter-group/MD-contacts-analysis.

**Acknowledgements** We thank E. Mila Vilalta, who ran the in vitro lipid binding assay; L. Alvarez Aguilar and M. Varga for generation of in house generated LTP cDNA libraries and subsequent cloning in mammalian vectors; the EMBL Metabolomics core facility, the Proteomics platform of the University of Geneva, Lipidomics Core Facility of the Danish Cancer Institute and the READS platform of the University of Geneva, for sharing expertise; Lipotype for their expertise in lipidomic analysis; V. Rybin for help with protein expression and biophysical characterization; B. Klaus and M. Rogon for sharing expertise in bioinformatics and statistical data analysis; M. Beck and his group, in particular A. Andres-Pons, for sharing expression vectors and know-how on human cell culture and inducible systems; A. De Matteis and M. Fico for insightful discussions; A. Lathuilière for constructive feedback on the manuscript; H. Riezman, E. Varesio, B. Snijder, G. D'Angelo, L. M. Ahmad, and T. Hornemann for inspiring discussions; and members and former members of the A.-CG groups at the Department of Cell Physiology and Metabolism, University of Geneva and European Molecular Biology Laboratory, EMBL, especially, E. Galster, as well as members of the A. Lathuilière group for continuous discussions and support. A.-C.G. acknowledges financial support from the Louis-Jeantet foundation and the Swiss National Science Foundation (grant 320030-227988). K.T. was supported by the EU Marie Skłodowska-Curie Actions project 843407, LipTransProMet. N.R., F.E., S.G., C.C., M.M., R.T. and A.-C.G. acknowledge financial support from the Research Council of Norway (grant 335772). K.M. acknowledges financial support from the Independent Research Fund Denmark (6108-00542B) and the Danish Cancer Society (R124-A7929). Simulations were performed on resources provided by Sigma2—the National Infrastructure for High-Performance Computing and Data Storage in Norway (grant nn4700k). We also acknowledge Sigma2 for awarding this project access to the LUMI supercomputer, owned by the EuroHPC Joint Undertaking, hosted by CSC (Finland) and the LUMI consortium through Sigma2/Norway.

**Author contributions** A.-C.G., A.C. and M.L.H. designed the research. K.T., D.T., M.L.H, A.-C.G., T.A. and J.S.-R. designed the data integration and analysis strategy. A.C., J.Z., L.v.E., C.G. and C.C. conducted the molecular biology, cell biology and biochemistry experiments. L.v.E. and A.-C.G. designed and conducted LTP overexpression experiments with the help of K.T. M.L.H. established the lipidomics platform with the help of A.C. M.L.H., A.C., I.Ø.N., M.M.F. and K.M. conducted the mass spectrometry analyses and K.T. analysed their data. K.T., D.T. and S.T. performed the bioinformatic analysis. N.R., M.M. and R.T. performed and analysed the molecular dynamics simulations. F.E., N.R., K.O. and S.G. designed and performed the structure-based analyses of the lipid binding pockets. A.-C.G., KT and M.L.H. wrote, and all authors reviewed, the manuscript.

**Competing interests** A.-C.G. reports funding from Lipotype GmbH, Dresden Germany. J.S.-R. reports funding in the last three years from GSK and Pfizer and fees or honoraria from Travere Therapeutics, Stadapharm, Astex, Owkin, Pfizer, Vera, Grunenthal, Tempus and Moderna. M.L.H. is currently employed by Absea Biotechnology GmbH, Berlin, Germany. I.Ø.N. is currently employed by Novo Nordisk, Bagsværd, Denmark and A.C. is currently employed by AB Sciex Germany GmbH. T.A. is a co-founder and shareholder of DeepCyte Inc. and Metacloud Inc. M.M.F. is currently the owner of MM Foged Consulting. The other authors declare no competing interests.

**Additional information**
**Correspondence and requests for materials** should be addressed to Marco L. Hennrich or Anne-Claude Gavin.

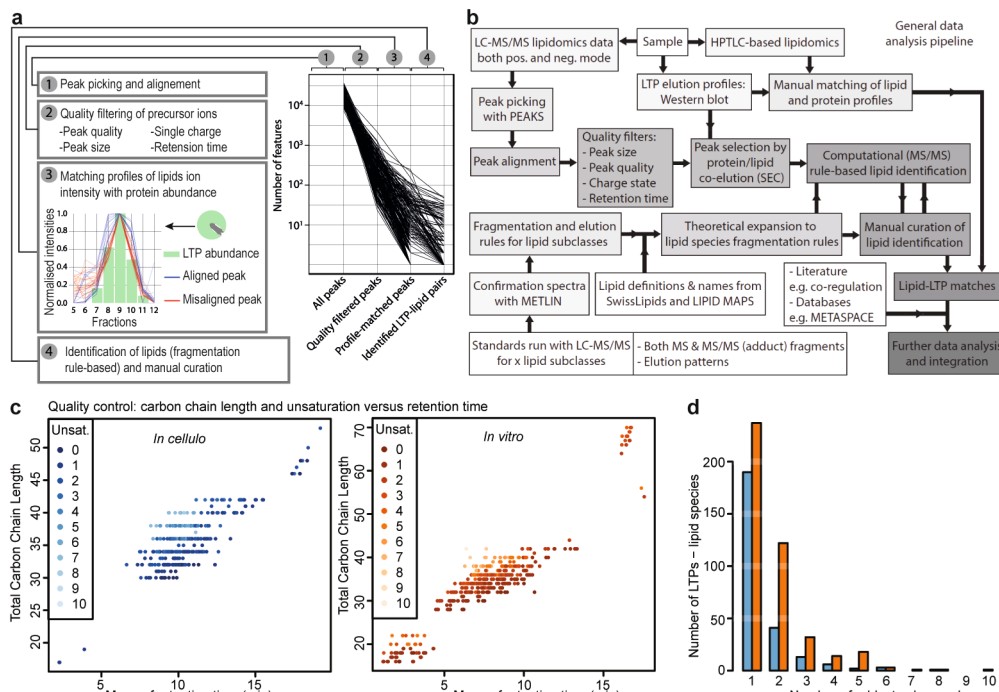

**Extended Data Fig. 1 | (a)** Main steps in the analysis of lipidomic data after LC-MS/MS, ensuring filtering of non-specific background. **(b)** Detailed workflow for the LC-MS/MS data analysis pipeline. **(c)** Lipid species with long acyl chains (y-axis) and few unsaturations (color gradient) have longer retention times (x-axis) compared to short acyl chains and acyl chains with many unsaturations. The retention times (x-axis) of lipid ligands identified in the in cellulo (left panel) and in vitro (right panel) screens are plotted against the summed number of carbons in the chains (y-axis). Unsat., unsaturation. **(d)** Number of LTP ligands observed with different adducts.

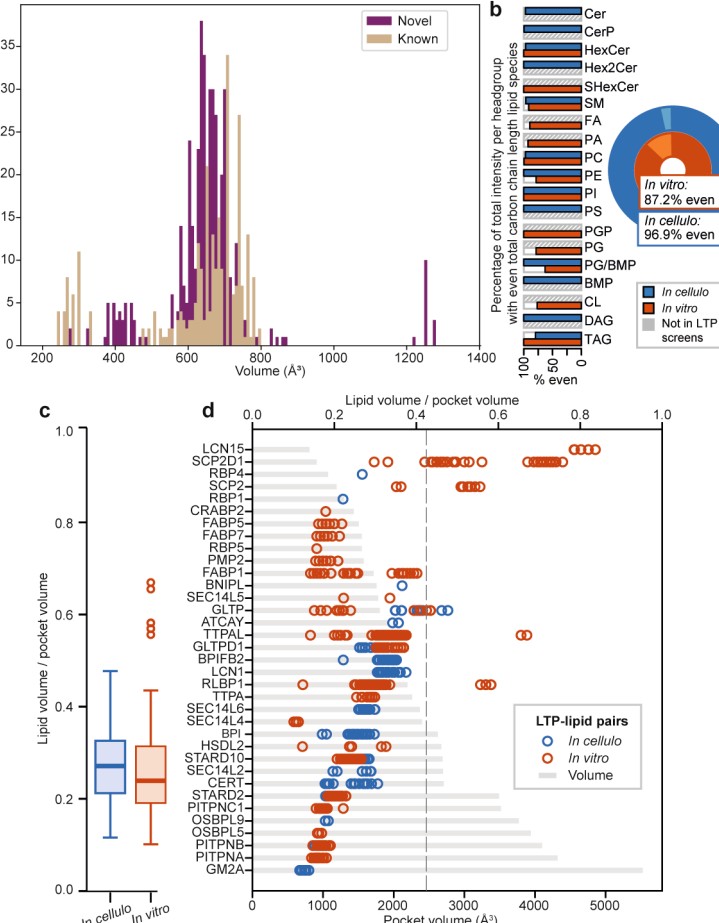

**Extended Data Fig. 2 | Structural and functional benchmarks. (a)** Distribution of volumes (Å³) of all lipid species (known or novel) mobilized by LTPs, calculated as described in the Supplementary Methods section. **(b)** Left panel: Overview of the distribution of total odd- and even-numbered fatty acyls in lipid classes mobilized by LTPs. The total intensity of all lipid species of a category was set to 100%. The percentage of intensity from lipids with an even-numbered total carbon chain length is displayed as a bar. Right panel: Fraction of all mobilized (in cellulo or in vitro) lipid species with an even-numbered total carbon chain length. **(c, d)** Comparison of the in cellulo and in vitro ligands in the structural benchmark. **(c)** LTPs excluding LCN15, SCP2D1 and SCP2, the three outliers with particularly small lipid-binding pockets. The whisker plot shows the ratio between lipid species and pocket volumes (y-axis) for in cellulo (n = 262) and

in vitro (n = 451) LTP-lipid pairs. Center line, median; box limits, first and third quartiles; the farthest data point within 1.5 times the interquartile range; circle-points, outliers. The two distributions overlap, meaning that the distributions are not statistically different (Welch's two-sided t-test, p-value = 0,132). Similar results are obtained when the entire lipocalin and SCP2 families are excluded from the comparison (n = 235 in cellulo; n = 451 in vitro; Welch's two-sided t-test, p-values = 0.710; data not shown). **(d)** Distribution of the ratios between lipid species volumes and LTP pocket volumes (upper x-axis) for all LTP-ligands pairs analyzed. The pocket volume is also represented with grey bars for each protein (lower x-axis). The vertical striped line indicates the maximal ratio (0.425) of lipid species volume to pocket volume observed for the known LTP-ligands pairs.

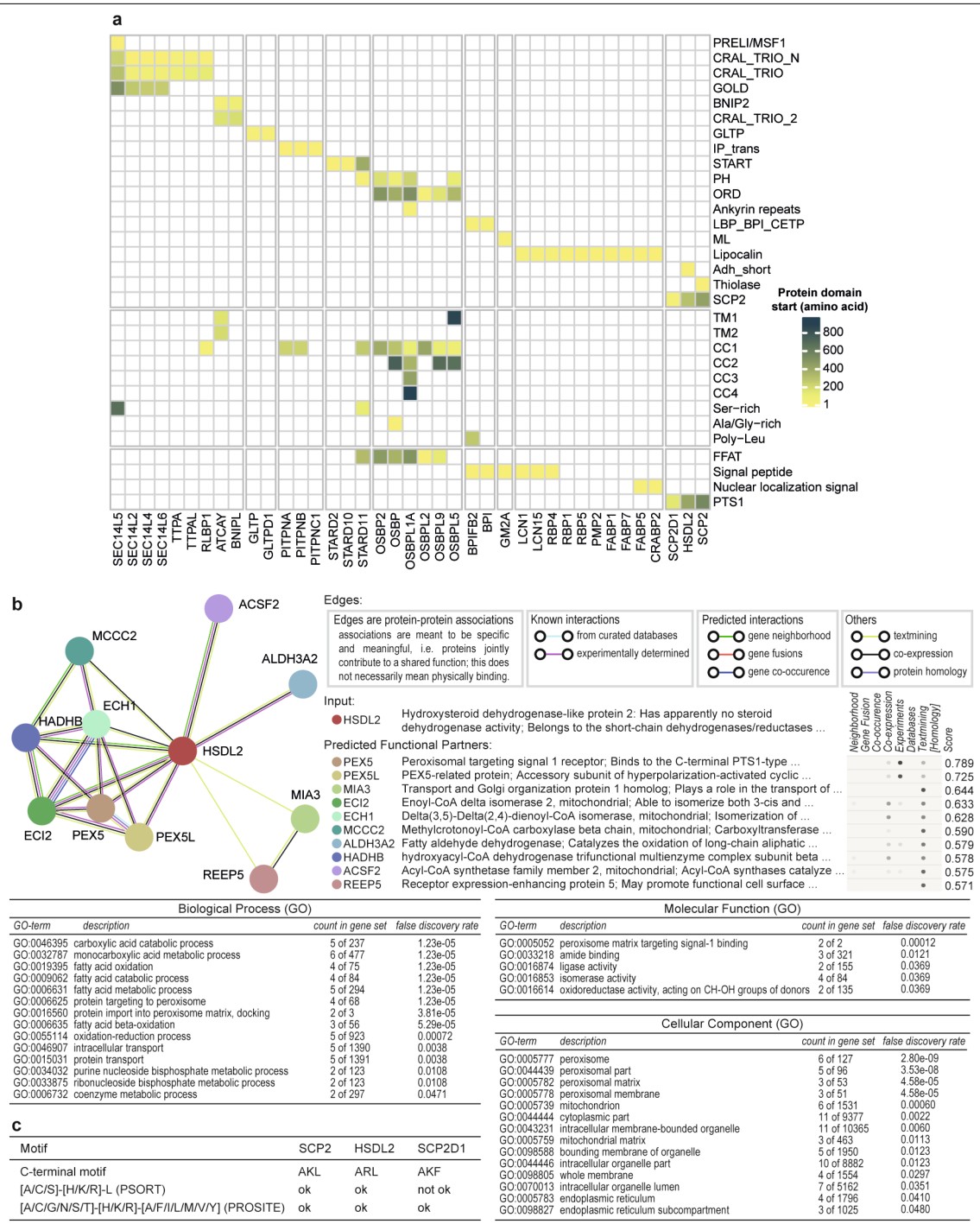

**Extended Data Fig. 3 | Results of LTP seriation according to domains, regions and motif position on their sequence.** These results have been used to order LTPs in Figs. 2a and 3a. The scale represents the position of the start of the domain on the protein sequence according to the database Pfam (Supplementary Methods). The seriation of the LTPs was done according to the travelling salesman algorithm (Supplementary Methods). (**b**) Analysis of the HSDL2 interactome and associated enrichment of gene ontology (GO) terms of its nodes (results from STRING database search, version 11.0 on 05/2020, Supplementary Methods). (**c**) Analysis of the peroxisomal targeting signal (PTS1) for HSDL2 and the other LTPs from the SCP2 family which mobilized lipids in both screens (Supplementary Methods).

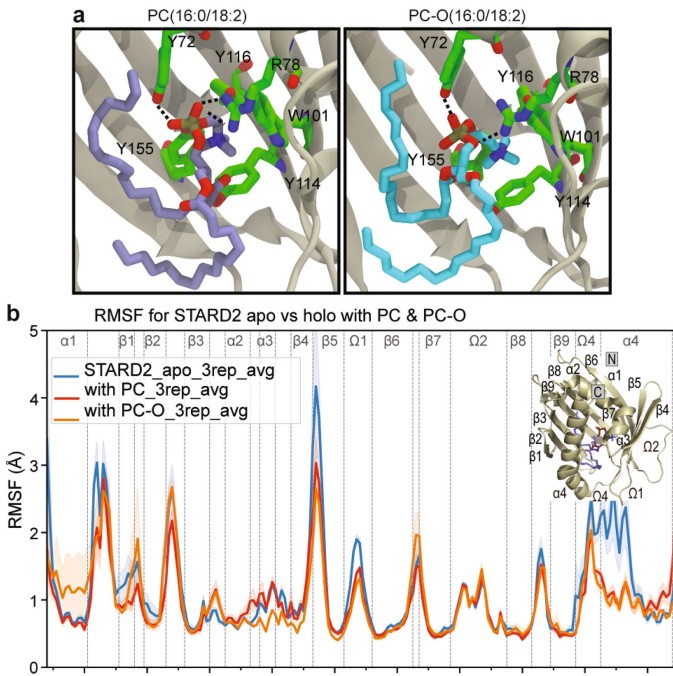

**a**

PC(16:0/18:2)

PC-O(16:0/18:2)

**b** RMSF for STARD2 apo vs holo with PC & PC-O

STARD2_apo_3rep_avg
with PC_3rep_avg
with PC-O_3rep_avg

RMSF (Å)

Residue number

**Extended Data Fig. 4 | STARD2 binds phosphatidylcholine and ether-phosphatidylcholine.** (**a**) Representative snapshot from triplicated independent 500 ns-long molecular dynamics simulations of STARD2 (cartoon representation) in complex with phosphatidylcholine (PC(16:0/18:2)) (left panel, purple) and ether-phosphatidylcholine (PC-O(16:0/18:2)) (right panel, cyan). A stick representation is used to highlight key aromatic amino acids (W101, Y114, Y116, and Y155) interacting with the choline moiety of the phosphatidylcholines and polar amino acids (Y72, R78) interacting with the phosphate group. Hydrogen bonds are represented with black dashed lines. (**b**) Root-mean-square fluctuation (RMSF) profiles of STARD2 in the apo state (blue) compared with holo states bound to phosphatidylcholine (PC, red) and ether-phosphatidylcholine (PC-O, orange). RMSF values are averages over three independent simulations. Shaded regions represent the standard error of the mean. Secondary structure elements are shown above the plots and labelled on the holo structures displayed as insets.

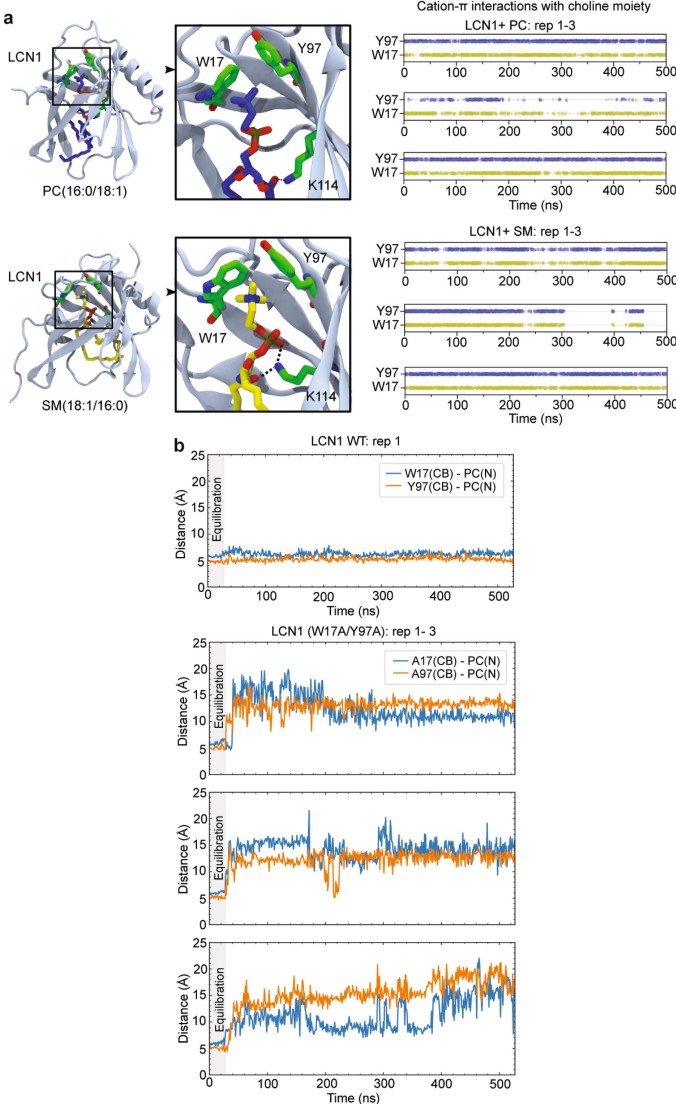

**Extended Data Fig. 5 | LCN1 sphingomyelin and phosphatidylcholine binding site.** (**a**) Left panels: snapshots from molecular dynamics simulations showing phosphatidylcholine (PC(16:0/18:1), dark blue) or sphingomyelin (SM(18:1/16:0), yellow) bound to LCN1 and highlighting the aromatic residues involved in cation-π interactions with the choline moiety of PC (W17 and Y97). Hydrogen bonds between lipid and protein are represented with dashed black lines. Right panels: time-series analysis of cation-π interactions between W17 and Y97 and the choline headgroup of the phosphatidylcholine (PC) or sphingomyelin (SM). n = 3 independent simulations. (**b**) Effect of W17 and Y97 mutations on the position of phosphatidylcholine (PC) in the LCN1 binding site. Distance between the choline headgroup of phosphatidylcholine and the suggested LCN1 binding site for choline plotted along simulation time: (upper panel) LCN1[WT] and (lower panels) LCN1[W17A/Y97A]. For each system, LCN1 is loaded with phosphatidylcholine. The distance is calculated between the choline nitrogen (PC(N)) and the beta carbon (Cβ) of residues 17 and 97. The time series show both the equilibration and production time windows. n = 3 independent experiments for both simulations.

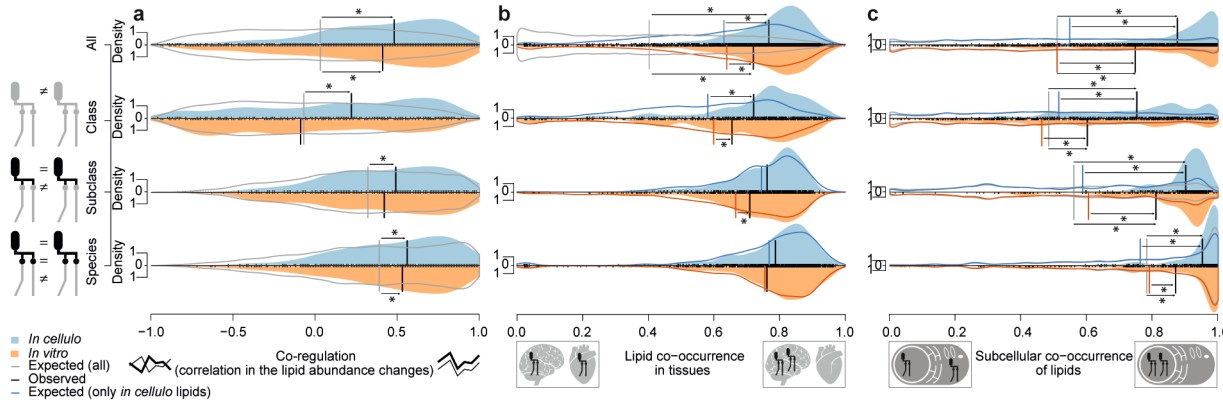

**Extended Data Fig. 6 | Lipid pairs mobilized by the same LTP share biological relationships: they are co-regulated and co-localized. (a-c)** Distribution of all correlation scores (co-regulation and co-occurrences) calculated for ligand pairs in in cellulo or in vitro screens. Y-axis, data densities (with Gaussian kernels) for different LTP-ligand pairs: species pairs (with different chains, but the same headgroup and linkage), subclass pairs (with different linkage, but the same headgroup) and class pairs (with different headgroups). An overall density of 1 would correspond to a uniform distribution. All densities have the same surface area for all comparisons within each subpanel. Large vertical lines in the graphs represent medians. Small vertical lines show the distribution of individual ligand pairs. Significant shifts of the observed medians are indicated by an asterisk (see Supplementary Table 9B, Fisher's exact tests). The expected distribution lines in grey are always based on all entries in the reference database, while the expected distribution lines in color are only based on the observed lipid subclasses from the reference database by either the in cellulo (blue) or in vitro (orange) screens. (**a**) Comparison of lipid co-mobilization by the LTPs with the co-regulation of these lipids following cellular perturbations. (**b**) Comparison of lipid co-mobilization by LTPs with the Manders' co-occurrence of these lipids in tissue sections from the METASPACE database[35]. (**c**) Comparison of lipid co-mobilization by the LTPs with the Manders' co-occurrence of these lipids in sub-cellular fractions[36].

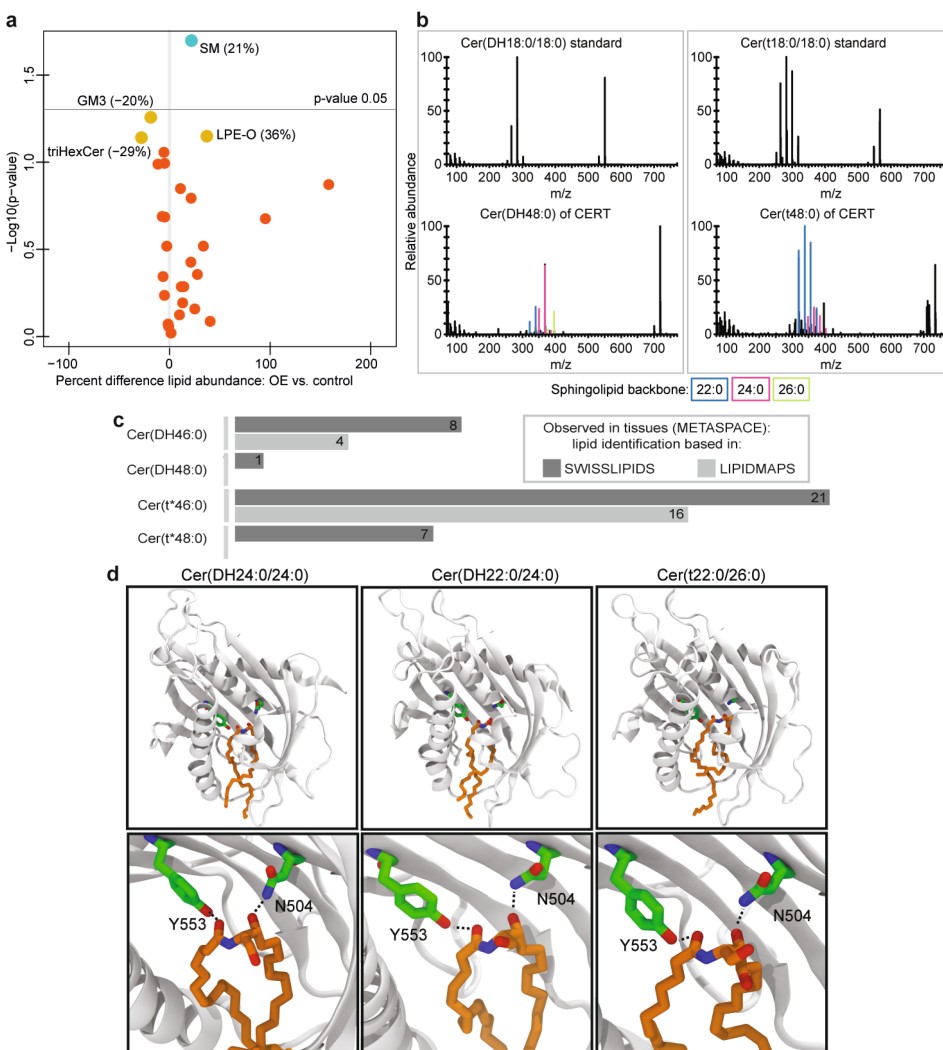

**Extended Data Fig. 7 | (a)** The effect of CERT overexpression (OE) on the lipidome of HeLa cells. Horizontal grey line, p-value = 0.05. Non-paired two-sided t-test. Each dot represents an individual lipid species. n = 3 biological replicates. **(b)** Fragmentation spectra for Cer(DH48:0) and Cer(t48:0), and those of Cer(DH18:0/18:0) and Cer(t18:0/18:0) standards. The patterns of the ceramides with carbon lengths of 48 are similar to our ceramide standards with the expected mass shifts. From the spectra it is obvious that different combinations of sphingoid base and N-acyl chain coelute and are co-fragmented. The m/z for the 22:0, 24:0 and 26:0 sphingoid backbones are colored in blue, pink and green, respectively. **(c)** Number of public datasets in METASPACE[35] that report the observation of Cer(DH46:0), Cer(DH48:0), Cer(t*46:0) and Cer(t*48:0). **(d)** Molecular dynamics simulations show that the START domain of CERT (shown as cartoon) can accommodate very long dihydroceramides (represented as sticks) in its hydrophobic cavity: Cer(DH24:0/24:0) and Cer(DH22:0/24:0); and also, phytoceramides, Cer(t22:0/26:0). Selected residues at the lipid binding site (N504, Y553) are shown as sticks. n = 3 independent simulations.

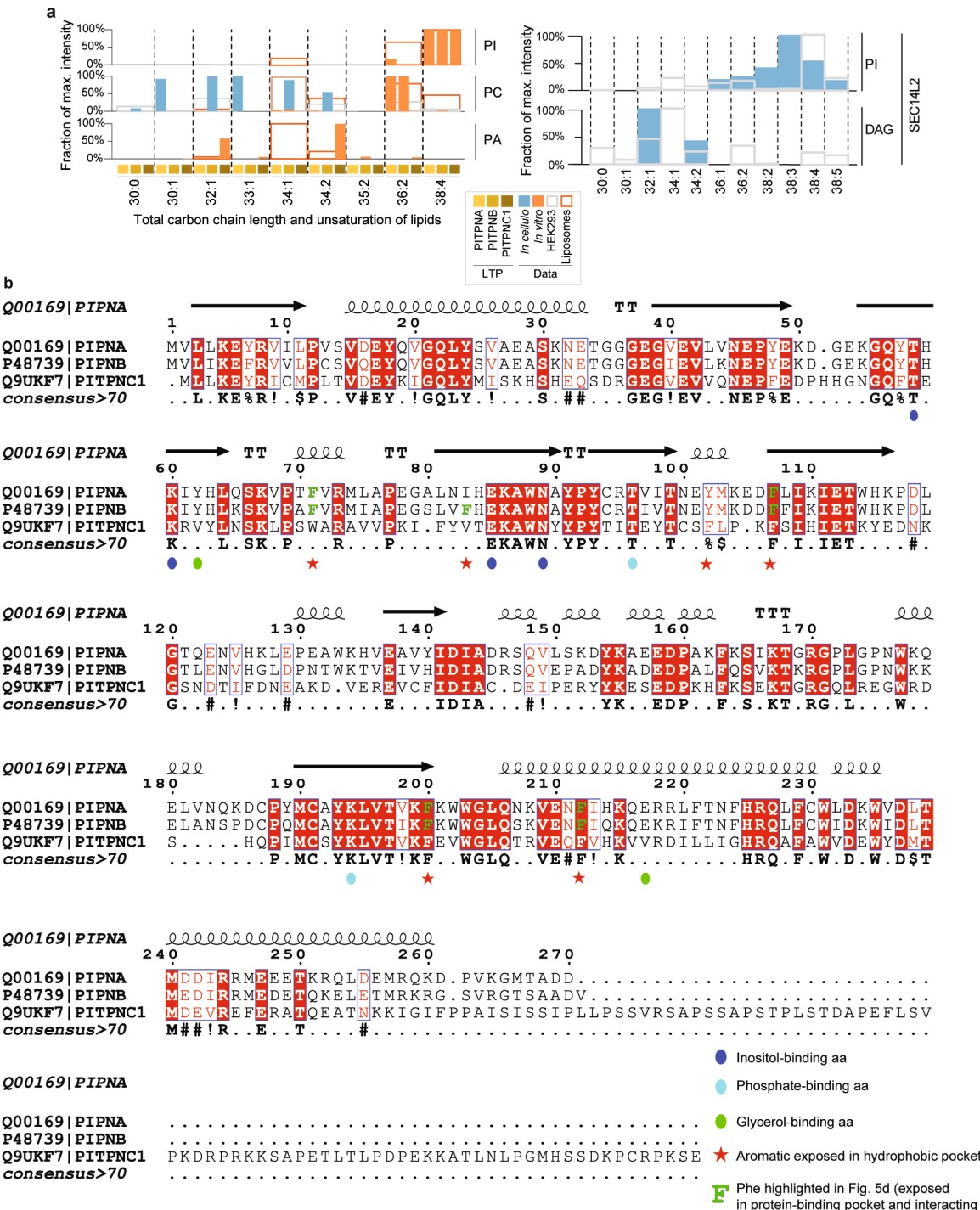

**Extended Data Fig. 8 | PITPs with preferences for distinct PI species. (a)** Left panel: Distribution of species of phosphatidylinositol (PI), phosphatidylcholine (PC) and phosphatidic acid (PA) bound to the PITP-family of LTPs. The observed intensities of the individual species are normalized to the maximum observed for the specific LTP-lipid subclass pair. The in cellulo (filled blue) and in vitro (filled orange) distributions for the LTPs are compared with distribution of the lipids present in the whole HEK293 cells (light blue border) and liposomes (orange borders). Right panel: observed lipid species of phosphatidylinositol (PI) and diacylglycerol (DAG) for SEC14L2 (filled blue boxes) in comparison with the observed species for these lipid classes in the full HEK293 cells (empty grey boxes). For completeness, the results for PITPNC1 are also mentioned, although in this case the identification of arachidonoyl-containing phosphatidylinositol is based on low abundant ions. **(b)** Sequence alignments highlighting several aromatic amino acids (mainly phenylalanines) conserved in human PITPNA, PITPNB (and PITPNC1). The phenylalanines shown in magenta in Fig. 5d are written in green.

# Reporting Summary

## Statistics

For all statistical analyses, confirm that the following items are present in the figure legend, table legend, main text, or Methods section.

| n/a | Confirmed | |
|---|---|---|
| ☐ | ☒ | The exact sample size (*n*) for each experimental group/condition, given as a discrete number and unit of measurement |
| ☐ | ☒ | A statement on whether measurements were taken from distinct samples or whether the same sample was measured repeatedly |
| ☐ | ☒ | The statistical test(s) used AND whether they are one- or two-sided<br>*Only common tests should be described solely by name; describe more complex techniques in the Methods section.* |
| ☒ | ☐ | A description of all covariates tested |
| ☐ | ☒ | A description of any assumptions or corrections, such as tests of normality and adjustment for multiple comparisons |
| ☐ | ☒ | A full description of the statistical parameters including central tendency (e.g. means) or other basic estimates (e.g. regression coefficient) AND variation (e.g. standard deviation) or associated estimates of uncertainty (e.g. confidence intervals) |
| ☐ | ☒ | For null hypothesis testing, the test statistic (e.g. *F*, *t*, *r*) with confidence intervals, effect sizes, degrees of freedom and *P* value noted<br>*Give P values as exact values whenever suitable.* |
| ☒ | ☐ | For Bayesian analysis, information on the choice of priors and Markov chain Monte Carlo settings |
| ☒ | ☐ | For hierarchical and complex designs, identification of the appropriate level for tests and full reporting of outcomes |
| ☐ | ☒ | Estimates of effect sizes (e.g. Cohen's *d*, Pearson's *r*), indicating how they were calculated |

*Our web collection on statistics for biologists contains articles on many of the points above.*

## Software and code

Policy information about availability of computer code

| | |
|---|---|
| Data collection | Lidomics data for LTP were acquired on an Agilent 1260 HPLC system coupled to a Q-Exactive Plus (Thermo) operated by a Xcalibur software versions 2.5.0.2042 or 2.8.1.2806. Lipidomics data for LTP gain-of-function was performed by Lipotype GmbH (Dresden, Germany) on a QExactive mass spectrometer (Thermo Scientific) equipped with a TriVersa NanoMate ion source (Advion Biosciences). SEC experiments were performed on an Ettan LC (GE Healthcare) or an Akta Pure (Cytiva) systems. Imaging of WB and HTPLC was done on a ChemiDoc (Bio-Rad) and a Pharos FX Plus molecular imager (Bio-Rad) imaging systems. Fluorescent-based binding assay was performed on a Spectramax Paradigm plate reader Molecular Devices piloted by SoftMax Pro 7 software. |
| Data analysis | Raw LC-MS/MS data were converted to mzML format with MS Convert from the ProteoWizard package (version 3.0.679). Peak picking, feature detection and alignment were done with PEAKS Studio 7.0 (Bioinformatics Solutions Inc.). HPTLC data were analysed using ImageJ 1.48m. Analysis of the fluorescent-based lipid binding data were performed using Graphpad Prism 10 version 10.6.1. The data analyses were done with R version 3.5.0 on a x86_64-w64-mingw32 platform. Scripts were tested with R version 4.2.2 on a x86_64-w64-mingw32/x64 (64-bit) platform. The following packages (algorithms) were used: for the visualization of beanplots, we employed the R package beanplot (version 1.2); for the circular visualization and heatmaps, we employed the packages circlize (version 0.4.12) and ComplexHeatmap (Bioconductor, version 2.3.2); for various string modifications, we employed the stringr package (version 1.4.0); for coloring schemes and gradients, we employed the RColorBrewer package (version 1.1-2); to extract input from Uniprot, we used the UniprotR package (version 1.2.4.); for automatic structural chemical entity classification we used ClassyFire (http://classyfire.wishartlab.com/). Identification and calculations of the volumes of the LTP pockets was done using the voronota-pocket script in the Voronota software package (version 1.28.4083) available at https://github.com/kliment-olechnovic/voronota/releases/tag/v1.28.4083. The volume of the lipids required an automated calculation of the number and nature of atoms and bonds, which was done with the RDKit: Open-Source Cheminformatics Software. Systems for molecular dynamics simulations were prepared using the CHARMM-GUI web server, the EDock web server, and the CHARMM General Force Field |

(CGenFF) program version 2.5.1. The CGenFF force field v. 4.6 and the CHARMM36m force field with its extension for WYF-choline cation-π interactions were used. Molecular dynamics simulations were conducted using the NAMD3 simulation package. The analyses were conducted using several software: Charmm (version 49a1), an in-house code based on the MDAnalysis package (https://github.com/reuter-group/MD-contacts-analysis), and VMD. PyMol (version 3.0.0) and VMD (version 1.9.3) were used to prepare figures of protein structures.
The codes are accessible via in https://github.com/saezlab/lipyd and https://github.com/krtiteca/ScriptsAssociatedWithLTPArticle. The voronota-pocket script is available at https://github.com/kliment-olechnovic/voronota/blob/master/voronota-pocket. The in-house code based on the MDAnalysis package is available at https://github.com/reuter-group/MD-contacts-analysis.

For manuscripts utilizing custom algorithms or software that are central to the research but not yet described in published literature, software must be made available to editors and reviewers. We strongly encourage code deposition in a community repository (e.g. GitHub). See the Nature Portfolio guidelines for submitting code & software for further information.

## Data

Policy information about availability of data

All manuscripts must include a data availability statement. This statement should provide the following information, where applicable:
- Accession codes, unique identifiers, or web links for publicly available datasets
- A description of any restrictions on data availability
- For clinical datasets or third party data, please ensure that the statement adheres to our policy

The lipidomics data can be downloaded from https://www.ebi.ac.uk/metabolights/MTBLS9567. Organelle lipidomics data are available at https://doi.org/10.1101/2025.10.05.680593.
The source data are organized as followed: molecular biology and LTP expression, MS fragmentation behaviour Supplementary Tables 1,2 and 3, respectively; lipid species or subclasses bound to LTP, Supplementary Tables 4 and 5, respectively; LTPs lipidated in only one of the assays, Supplementary Table 6; Results of the structural and functional benchmarks, Supplementary Tables 7 and 8; Functional relationship of lipids co-mobilized by the same LTP, Supplementary Table 9; Lipidomic data of bovine liver / brain extracts and HEK293 cells, Supplementary Table 10; Lipidomics CERT-over-expressing HeLa cells, Supplementary Table 11. Supplementary Tables used to produce the panels of the figures are summarized in Supplementary Table 14. Gels, SEC profiles and western blots are provided in Supplementary Figure 1.
External datasets analysed (but not generated) in this work are: sequences of human LTPs, UniProtKB (https://www.uniprot.org/uniprotkb) and SMART (https://smart.embl.de/smart/change_mode.cgi); definition of domains and motifs, InterPro (https://www.ebi.ac.uk/interpro/) and PFAM (http://pfam.xfam.org/); lipid identification rule-sets, SwissLipids (https://swisslipids.org/) and METLIN (https://metlin.scripps.edu/); SMILES for 1D lipid representations, LIPID MAPS (https://www.lipidmaps.org/); analyses of HSDL2 protein interactome, STRING (https://string-db.org/); analyses of organelle targeting sequences, PSORT (https://psort.org/) and PROSITE (https://prosite.expasy.org/); coregulated lipids, https://doi.org:10.1016/j.cell.2015.05.051; lipid colocalization, METASPACE (https://metaspace2020.org/); lipid subcellular localisation, https://doi.org:10.1101/2025.10.05.680593; protein structures, PDB (https://www.rcsb.org/); simulated protein structures, AlphaFold DB (https://alphafold.ebi.ac.uk/). External datasets analysed (but not generated) in this work are summarized in Supplementary Table 15.

## Research involving human participants, their data, or biological material

Policy information about studies with human participants or human data. See also policy information about sex, gender (identity/presentation), and sexual orientation and race, ethnicity and racism.

| | |
|---|---|
| Reporting on sex and gender | We have not performed research involving human participants |
| Reporting on race, ethnicity, or other socially relevant groupings | We have not performed research involving human participants |
| Population characteristics | We have not performed research involving human participants |
| Recruitment | We have not performed research involving human participants |
| Ethics oversight | We have not performed research involving human participants |

Note that full information on the approval of the study protocol must also be provided in the manuscript.

## Field-specific reporting

Please select the one below that is the best fit for your research. If you are not sure, read the appropriate sections before making your selection.

☒ Life sciences    ☐ Behavioural & social sciences    ☐ Ecological, evolutionary & environmental sciences

For a reference copy of the document with all sections, see nature.com/documents/nr-reporting-summary-flat.pdf

## Life sciences study design

All studies must disclose on these points even when the disclosure is negative.

| | |
|---|---|
| Sample size | No calculations were performed to determine the sample size. The sample size was limited by the number and availability of human LTP cDNAs and our success in expressing LTP in HEK293 and E. coli, respectively. |

| Data exclusions | No data were excluded. |
|---|---|
| Replication | All AP-MS analyses were performed once with lipidomic analyses of at least three adjacent SEC fractions in positive and negative ionization modes. Lipidomic analyses of HEK293 and HeLa cells overexpressing or not overexpressing LTP were performed in three independent biological replicates. The fluorescence emission shift binding assay was independently repeated eight times. All simulations were performed in triplicate, using a different velocity distribution for each replicate, and yielded similar results. |
| Randomization | Samples were not allocated to experimental groups. The HEK293-LTP cells that were cultured in parallel (batches) were selected so that they belonged to different LTP families (different names) and so that the molecular weight of the overexpressed LTPs was different in order to avoid any exchange of cell lines and to make any errors visible. For AP-MS experiments, samples were processed for LC-MS/MS as soon as they became available. For lipidomic analyses of LTP-overexpressing lines, they were randomized using the Rand() function in Microsoft Excel. |
| Blinding | Differences in the growth rates between the different cell lines and the need to verify that HEK293 and E. coli overexpressed LTP with the correct molecular weight prevented investigator blinding. Investigators were blinded to sample identity for the lipidomics analyses (AP-MS and HEK293 cells over-expressing LTP). |

# Reporting for specific materials, systems and methods

We require information from authors about some types of materials, experimental systems and methods used in many studies. Here, indicate whether each material, system or method listed is relevant to your study. If you are not sure if a list item applies to your research, read the appropriate section before selecting a response.

### Materials & experimental systems

| n/a | Involved in the study |
|---|---|
| ☐ | ☒ Antibodies |
| ☐ | ☒ Eukaryotic cell lines |
| ☒ | ☐ Palaeontology and archaeology |
| ☒ | ☐ Animals and other organisms |
| ☒ | ☐ Clinical data |
| ☒ | ☐ Dual use research of concern |
| ☒ | ☐ Plants |

### Methods

| n/a | Involved in the study |
|---|---|
| ☒ | ☐ ChIP-seq |
| ☒ | ☐ Flow cytometry |
| ☒ | ☐ MRI-based neuroimaging |

## Antibodies

| Antibodies used | anti-HA, 12CA5 mouse hybridoma, EMBL PEPCF (from InVivo BioTech Services, GmbH a Bruker company, clone name 12CA5; batch AK2055/01B.1); dilution 1:1.000); Anti-Mouse IgG HRP,  cat. nr GENA931-1ML, Sigma-Aldrich (Cytiva)(dilution 1:10.000) . |
|---|---|
| Validation | All antibodies were validated by the  manufacturer for their suitability for use in western blot applications.<br>Anti-HA, 12CA5 -> Applications: dot blots, immunochemistry[, immunoprecipitation, western blotting<br>Anti-Mouse IgG HRP,  GENA931 -> Applications: mouse IgG HRP linked whole Ab has been used in immunoblotting |

## Eukaryotic cell lines

Policy information about cell lines and Sex and Gender in Research

| Cell line source(s) | Flp-In T-REx-293 (Cat.Nr R78007) and Flp-In T-REx-HeLa (Cat.Nr R71407) were sourced from Thermofisher. |
|---|---|
| Authentication | No further authentication was performed. |
| Mycoplasma contamination | These cells were regularly checked and tested negative for mycoplasma. |
| Commonly misidentified lines (See ICLAC register) | No commonly misidentified cell lines were used in this study. |

## Plants

Seed stocks

We did not use seed stockes

Novel plant genotypes

We do not report novel plant genotypes

Authentication

We have not done plant authentication

