## [Peer Review File · Nature]

Systematic analyses of lipid mobilization by human lipid transfer proteins

Corresponding Author: Professor Anne-Claude Gavin

Version 0:

Reviewer comments:

Referee #1

(Remarks to the Author)

This study systematically determines the lipids bound by the majority of known human lipid transport proteins (LTPs). Given how challenging it can be to find which lipids are bound by an LTP and that the lipids bound by many LTPs were unknown, this study is a technical tour de force. The findings will be an important resource for those investigating LTPs, lipid metabolism, and lipid signaling. The authors identify two broad conceptual advances provided by their work: many LTPs can recognize both the head groups and acyl chains of the lipids they bind and many LTPs bind more than one class of lipids. While these are notable conclusions, they are not terribly surprising since they already been shown to be true of several LTPs families, such as the ORPs and PITPs.

The work is well done. While it is likely to an important resource, it does not have a big enough conceptual advance for it to be appropriate for a high impact journal. The study would be strong if it more definitively showed how the new binding data allows novel biological insights. It comes closest it its examination of CERT function, but the interesting new ideas about the roles of CERT in complex sphingolipid metabolism are not verified by in cell lipid trafficking measurements and lipid flux analyses.

The study also makes claims about lipid movement in cells that are not supported by experimental evidence. Lipid mobility in cells, contrary to the claim of the title, is not measured. The results section has interesting speculation about routes of lipid transfer in cells, but they are not tested.

(Remarks on code availability)

Referee #2

(Remarks to the Author)

In this manuscript, the group of Anne-Claude Gavin and collaborators have carried out a systematic brute force effort to identify lipid cargoes for human lipid transfer proteins (LTPs). Two complementary approaches were applied: 1) inducible overexpression of affinity-tagged LTPs in HEK293 cells followed by purification and analysis of lipids stably associated with them, and 2) expression and purification of LTPs from *E. coli* and analysis of lipids extracted by them from 400 nm liposomes made of DOPC, bovine liver and porcine brain lipid extracts. From a total of 101 LTPs cloned and 95 expressed, 73 unique protein-lipid complexes were purified (of these, 37 proteins were from both cell lysates and LTP-liposome mixtures, the rest from either cells or liposome mixtures) and finally, lipid cargoes were identified for 39 unique proteins.

Overall, this work has an ambitious goal. It aims to generate a resource that would capture the general principles of non-vesicular lipid transport in humans. Yet, the number of novel lipid-protein interactions identified remained rather modest, with the majority being based solely on the in vitro approach, leaving reservations on the physiological relevance. Lipid ligands for nine LTPs with no previously known ligands were identified. In principle, the two approaches should strengthen each

other. However, only six proteins extracted lipids in both approaches. It seems that only one of these represents a novel lipid-protein interaction: STARD2, a.k.a. PCTP (phosphatidylcholine transfer protein) was found to complex not only with PC, but also with ether-PC.

Furthermore, some of the general principles highlighted, e.g. that LTPs form complexes with more than one lipid class, are already well known, while some of the more exciting ideas proposed remain insufficiently documented and validated. Please see specific comments below.

Major points:

1. Validation of in vitro lipid binding data in cells: The in vitro approach has considerable limitations. The human LTPs can already bind E.coli lipids and LTPs in vitro are devoid of cellular spatiotemporal control mechanisms, as also discussed by the authors. Therefore, it would be important to strengthen at least some of the novel lipid-protein interactions identified solely from the liposome approach, by additional functional cell-based data.

As an example, HSDL2 was identified to bind TAG in vitro. The authors discuss that HSDL2 is a peroxisomal LTP and according to GeneCards, it is located in mitochondria and peroxisome. TAG would be a novel ligand in the LTP system and neither mitochondria nor peroxisomes are canonical sites of TAG transport or metabolism. Thus, it is not easy to understand what the role of this TAG-binding LTP should be. Additional experimental evidence from cellular systems supporting the in vitro binding would be very helpful.

2. One of the interesting ideas proposed relate to new principles of sphingolipid transport, with three pools of ceramide with different metabolic fates pending on chain-length and co-operation of transfer proteins. However, the overall concept and its implications are not clearly enough articulated and the experimental evidence in support is rather preliminary. The authors did carry out a small pilot experiment where they overexpressed CERT and analyzed its effects on the HEK293 cell lipidome, then reporting on increased total cellular sphingomyelin and ceramide 1-phosphate and decreased hexosylceramides (Fig. 6b). It would be important to elucidate the lipidome changes in more detail, e.g. the chain compositions, other lipids including other identified CERT ligands, whether the levels of other LTPs are adjusted in response to CERT overexpression, and if the effects of CERT overexpression are specific to the HEK293 model or more general. It seems that the CERT simulation data (Fig. 6c) provides more questions than answers, e.g. how does the presence of POPC affect the affinity for ceramide of different chain lengths, how strong are these interactions etc.

3. Another interesting suggestion made is the potential organization of LTPs into functional networks linking distinct organelles and contributing to the coordination of metabolism between different cellular compartments. This is based on the observation that lipid pairs mobilized by the same LTPs should be co-regulated and co-localized. By scoring such co-occurrence from available datasets, the authors indeed find evidence for this. The problem is that the subcellular co-localization data – that turned out to be the most significant for the conclusion – is not openly available. In the methods, the authors state that this is a dataset containing lipidomes of organelles affinity-purified with antibodies in a manuscript in preparation. It would be important to make this dataset publicly available in the context of the present manuscript, as it is critical for the conclusions and would align with the authors' idea of a resource article.

Additional points:

4. Are the very short chain PC and PE species of HEK293 cells (e.g. C14, C16 in Fig. 2b) lysolipids? The authors identified 14 potential new lysolipid interacting proteins using the in vitro assay. How does a single vs. two acyl chains affect the extractability from membranes?

5. In Figs 3c and 6a, grey borders indicate the relative distribution of lipids present in HEK293 cells. Comparing the observed distribution of lipids in vitro to cellular lipids does not seem to make sense. Should these rather be compared to the relative distribution of lipids in the liposomes?

6. The authors found that in HEK293 cells, PI did not associate with PITPs but with SEC14L2. They discuss that this is probably because SEC14L2-PI complexes contribute to important functions in dividing cells that need to duplicate their organelles, whereas phospholipase signaling is probably not active in this context. What would these important functions be? Also, it seems unlikely that PLC signaling is not operating in these cells, see e.g. PMID 15741171.

(Remarks on code availability)

Referee #3

(Remarks to the Author)

This study provides the first systematic analysis of the lipid binding properties of human lipid transfer proteins (LTPs), a large and structurally diverse class of soluble lipid transporters that coordinate spatially separated lipid metabolic pathways, maintain organelle-specific lipid compositions and govern lipid-mediated cell signaling. By coupling high-throughput affinity purification methods with LC-MS/MS-based lipidomics, the authors identified lipid cargoes for 39 unique human LTPs using two complementary approaches. The study as a whole is a tour-the-force, yielding not only many novel lipid-binding partners but also establishing an important methodological pipeline for identifying protein-bound lipids and exploring protein-lipid interactions in different cell types and under distinct physiological conditions. Interestingly, most LTPs analyzed were found to bind more than one class of lipids. Data base mining and correlation analysis revealed that lipid pairs mobilized by the

same LTPs are significantly more co-regulated upon metabolic perturbation and co-localized than would be expected from random sets of lipid pairs. Based on these findings, the authors postulate that this lipid co-transport represents an important, but hitherto underestimated mechanism for the organization and integration of spatially segregated lipid metabolic pathways. While experimental validation of the latter concept would have been very welcome, the reported LTP-lipid interaction map and how it was established provide an important resource for future work aimed at unravelling the architecture and regulation of the highly interconnected metabolic lipid network.

However, the work also has some shortcomings that the authors need to comment on. To begin with, their study focuses exclusively on box-type LTPs and excludes bridge-like LTPs. While box-type LTPs contain an internal cavity that can accommodate one lipid molecule at the same time and shuttle between two membranes, bridge-like LTPs have an opening that extends along its entire length. The extended opening forms a seam that allows lipids to slide while the protein remains stationary positioned between two membranes. As bridge-like LTPs are more promiscuous regarding their lipid binding partners and often form large multimers, mapping their lipid binding properties using the approach described in this study will unlikely yield meaningful results. Nevertheless, it would appropriate that the authors at least mention their existence in the context of their work.

For mapping the lipid-LTP interaction network, the authors used two complementary approaches. In one they measured the ability of LTPs expressed in Hek293 cells to associate stably with lipids in a physiological context of an intact human cell. In addition, they analyzed the ability of LTPs expressed in *E. coli* to extract lipids from artificial membranes prepared with total liver or brain lipid extracts. Even though the majority of the 39 LTPs analyzed were successfully expressed and purified in both expression systems (HEK293 and *E. coli*), only six extracted lipids in both assays. This is somewhat surprising. In addition to commenting on this, it would be useful if the authors explicitly document which LTP expressed in both systems extracted lipids in only one. This may be particularly relevant given that, as noted by the authors, many of the LTP-lipid complexes formed in vitro contain lipids that are abundant in bacteria (FA, PA, CL, PG, PE and their lyso-forms), suggesting that their assembly may in fact have occurred in the *E. coli* expression host. This notion is consistent with the finding that the bound lipids often have odd numbered acyl chains, which are more frequent in bacteria. As LTPs pre-loaded with bacterial lipids may not be able to sample the full range of potential lipid cargoes present in membranes composed of lipids extracted from liver and brain, the lipid binding profiles of LTPs assessed using the in vitro biochemical assay may be skewed.

The group of LTPs whose lipid binding profiles were characterized under physiological conditions inside Hek293 cells includes several representatives of lipocalins, a large but poorly characterized family of secretory proteins that can bind a range of hydrophobic substances. As the expression constructs used to produce these proteins in Hek293 cells contain an N-terminal affinity tag, they would fail to enter the secretory pathway and accumulate in the cytosol. Of note is that membranes of secretory organelles typically display asymmetric lipid distributions. Consequently, lipocalins mistargeted to the cytosol may fail to encounter their preferred lipid binding partners. For instance, the authors find a strong signal for PC and only a weak signal for SM associated with the lipocalin LCN1 purified from Hek293 cells (Fig. 3a). As SM almost exclusively populates the luminal leaflet of secretory organelles, a preferential binding of SM by LCN1 would be obscured by failure of the latter to enter the secretory pathway. The latter scenario could be experimentally validated using the in vitro approach with LCN1 produced in *E. coli*. However, this analysis is missing from the study.

(Remarks on code availability)

Referee #4

(Remarks to the Author)

The manuscript by Gavin et al. describes the largest functional characterization of 39 unique lipid transfer proteins in human cells. To achieve that, the authors successfully overexpressed 86 LTPs in HEK293 cells (for the in cellulo characterization) and 71 LTPs in *E. coli* (for the in vitro characterization). They then employed affinity purification and size exclusion chromatography (SEC) to purify fractions where the LTPs (bound to their cargo) elute. Afterward, they performed HPTLC followed by LC-MS lipidomics to identify the lipid bound to the specific LTPs purified. Using their computational pipeline, the authors analyzed the lipidomics datasets and identified co-elution patterns between SEC and LC-MS. Finally, they infer general properties of LTP-associated lipidome, suggest new routes of lipid transfer, generalize their properties to lipid classes, and claim new principles of sphingolipid transport. Overall, the manuscript represents a significant leap in the characterization of LTPs in human cells and offers an invaluable resource to the lipid trafficking and cell biology community. However, several cell biological, and analytical concerns must be addressed to make some of the very important claims warranted. More specifically,

Biologically:

- The manuscript rightfully acknowledges, in several instances, that lipid transport depends on the local lipid environment of the respective LTPs. However, it utilizes the in vitro datasets in combination with the HEK 293 datasets to suggest a bona fide function of the purifiable 39 LTPs. This approach carries the inherent limitations that both in cellulo and in vitro functions, with that definition, will most likely not mimic the natural environment where these proteins are expressed and localized. This can lead to false positive and negative functional annotations, of which the rates are unknown. It is unclear how much of the reported cargo lipid for the novel LTPs will translate into a more native lipid environment where LTPs are expressed in the right organelle with the right lipid environment. A more systematic approach to report on the false positive/negative rate or the extent of the cargo dependence on the local lipid environment might be needed to address this. For example, i) the effect of subcellular localization of the LTPs, and thereby their local lipid environment, on their cargo, ii) the relationship of protein sequence or structure homology to function and propagating that knowledge to non-characterized LTPs, orthogonal

validation, etc.

- A striking consequence of ignoring the inherent false positive/negative assumption underlying the whole experimental design is the manuscript's statement, "Not all lipid species can be easily manipulated by LTP systems." This statement ignores that 1) not all relevant lipids are detected and ii) not all LTPs are characterized (less than 50% of the initially cloned LTPs could be characterized in the manuscript). Both facts leave room for invalidating the statement and similar generalizations across the manuscript.

- The manuscript relies on the gain of function (overexpression) as the sole mode of assigning function. A complementary approach with loss of function (knockdown or knockout of at least certain LTPs) can shed light on function in a more biological context in combination with gain of function experiments.

Analytically;

- The manuscript hints that establishing fragmentation rules for lipids is a contribution (lines 301-302). The lipid fragmentation rules have been established for a long time for all the classes described (for some examples from recent references that used the same set of rules)

o <https://doi.org/10.1016/j.xcrm.2021.100407> with all the rules and fragmentation pathways published in the GitHub

<https://github.com/SysMedOs/AdipoAtlasScripts/tree/main/LipidIdentification>

o LipidMatch <https://doi.org/10.1186/s12859-017-1744-3>

o Lipid Blast (<https://doi.org/10.1038/nmeth.2551>)

o MSDIAL <https://doi.org/10.1038/s41587-020-0531-2>

o Lipids Standards Initiative published a list of the diagnostic fragments (neutral losses): <https://lipidomicstandards.org/lipid-class-specific-fragments/>

As the rules are not novel and have been used in mass spectrometry-based lipidomics since its start more than 20 years ago, the manuscript should refer to the established rules and highlight the novelty they provide. It is unclear from the supplementary methods nor from GitHub which part is novel; therefore, this should be better highlighted.

- Unusual lipid species are reported, such as ceramides with carbon chain lengths of 46 and 48 and triglycerides with a total carbon chain length of 32. The manuscript should show the MS1 extracted ion chromatograms (EICs) and MS2 spectra in both cases and the rules used to verify these unusual species. For example, what are the sphingoid base and N-acyl chain lengths for that very long ceramide? No ceramide of that length has been reported in HEK 293 cells, and a citation is missing for its previous report. Moreover, the very short triglyceride with 32 carbons in the chains must contain medium-chain fatty acids, which were not reported earlier (nor cited by the current manuscript) in HEK 293 cells under normal growth conditions.

- The manuscript refers mainly to the sum composition level annotation of lipids. However, based on the fragmentation rules and the LC-MS method used, the authors should be able to identify lipids at the species level with the individual chain composition. It is not clear why the manuscript ignores that level of structural granularity. Especially in the case of sphingolipids, the manuscript seems to ignore the diversity on the sphingoid base, N-acyl combinations missing an opportunity to discover novel specificities (or proving the lack thereof).

- Normalization (min-max or sum) is performed, ignoring the fact that ionization efficiency is different among different lipid classes and follows certain trends in long vs. shorter acyl chain lengths. This leads to a situation where the same number of molecules of different lipid classes (or species) will generate significantly different numbers of ions (reflected in intensity levels or areas under the peaks for the same amount of different lipids). Considering this, it is unclear how the manuscript addresses this to comment on specific lipid cargo of different classes bound to the same LTP. This might directly affect some interpretations as it penalized lipids with low ionization efficiency or abundance. A possible suggestion is to use ionization efficiency factors (calibration curve slopes) from pure standards spiked to a common matrix, normalizing to protein abundance and showing the intensity distribution of all lipids detected in the same fraction detected for some LTPs.

(Remarks on code availability)

Version 1:

Reviewer comments:

Referee #1

(Remarks to the Author)

While this study has been improved, my concerns remain the same. The revised manuscript now more clearly explains what the major advances of the study are and more carefully describes the scope and limits of the results. However, the study still does not provide the major conceptual insights expected of paper in Nature. The rebuttal says, "... the conceptual advances are at molecular and biochemical levels, and not so much at the cellular ones." This is reasonable but the most important questions about LTPs are how they function in cells. Understanding which lipids are bound by most of the LTPs in humans is certainly important, but, after years of work on LTPs, the primary challenge remains understanding how LTPs function in cells and integrate with cellular lipid homeostasis. Knowledge of which lipids they bind is critical but, by itself, is not a game changer. The study remains an important resource for those who want to understand how LTPs function.

(Remarks on code availability)

Referee #2

(Remarks to the Author)

In the revised manuscript by Titeca, Chiapparino, Hennrich and coworkers, the authors have added considerable new experimental and simulation data to support their claims.

They have now provided additional experimental evidence from a cell-based system as a functional benchmark. Specifically, they measured how the lipidome of HEK293 cells is affected by overexpression of the individual LTPs studied. They found that the abundance of almost half (44%) of the lipid ligands identified was affected by overexpression of their respective LTP. This is significantly higher than what is observed when all possible LTP-lipid pairs are considered. Yet, in some cases, LTP overexpression increased and in other cases decreased the amount of the ligand identified. This is specifically mentioned in the few exemplary cases discussed more thoroughly. For instance, overexpression of SEC14L2 resulted in a decrease in the identified substrate diacylglycerol, whereas overexpression of HSDL2 resulted in an increase in the identified substrate triacylglycerol. The authors have provided a new Extended Data Table 8, which summarizes in which cases significant changes in the abundance of the expected ligand were observed. This Table should also include information on whether this change was an increase or a decrease. In addition, it remains unclear if significant changes in the levels of other lipids than the expected ligands were observed upon LTP overexpressions at the whole lipidome level. If such changes were observed, it would be helpful to include the information which additional lipid species were up- or downregulated, to facilitate follow-up studies.

According to the authors, the subcellular colocalization data that they use and refer to, cannot be made publicly available at this stage. Instead, they have listed the colocalization data for all lipid pairs in a new Extended Data Table 9A. This is unfortunately not very useful. However, according to the authors another manuscript describing these data in a more comprehensible manner will apparently be made public soon, and should remedy this.

The experimental work is now also supported by atomic-level simulation results. Their role in the manuscript is not decisive, but they provide some additional insight into how LTPs bind certain lipids. The basis of these simulations is the same force field (CHARMM36m), which has been appropriately used in all cases studied, so the results are comparable. However, there are four key problems and weaknesses in the simulation section, as follows:

- 1) The description of the methods gives the impression that simulations have been performed primarily only once for each system (STARD2 being an exception with 3 repeats). This does not meet quality standards (for comparison, would it be acceptable for a wet-lab experiment to be performed only once). Each system should be simulated at least three times over a sufficient time scale, producing independent samples.
- 2) The manuscript implies that the LTPs studied in the simulations (e.g., STARD2) mobilize lipids and that mobilization affects both ester and ether lipids. However, mobilization is not studied in the simulations. The simulations focus only on lipid-protein interactions within the LTP cavity. Does lipid movement occur between the membrane and the LTP? To do this, one would need to carry out a massive set of simulations to observe spontaneous loading/unloading events, but doing this is possibly not feasible. Alternatively, one would need to determine the free energy profiles of loading/unloading by first pulling a lipid out of the LTP into the membrane, which would also generate a transition path, and using it, implement the corresponding loading process from the membrane to the LTP. These simulations would reveal whether the free energy barrier for lipid loading to the carrier protein is low enough in relation to the thermal energy that the mobilization is possible; and whether the ester vs. ether bond affects this process, and in other cases whether the head group/acyl chain affects the mobilization. Without this information, the simulations described in the manuscript do not tell us anything other than the ability of certain lipids to remain inside the LTP for (less than) a microsecond.
- 3) Continuing from the previous point, how do lipids stabilize LTPs? If the observed lipids are ligands for these LTPs, then the presence of lipids inside the LTP stabilizes the protein structure. What does the RMSD/RMSF data tell?
- 4) The simulation data provide valuable information about how LTPs could specifically bind a particular lipid head group. For instance, the manuscript reports that in LCN1 the residues W17 and Y97 would be critical for this binding with the PC head group. The immediate question is, if these residues are mutated (in simulations), will the specific binding be lost?

(Remarks on code availability)

Referee #3

(Remarks to the Author)

In the revised manuscript, the authors introduced structural and functional benchmarks to better assess the quality and functional relevance of their system-wide analysis of LTPs to mobilize lipids. As structural benchmark, they determined to what extent the sizes of lipid ligands identified in their assays fitted in the volumes of lipid-binding pockets estimated from known or predicted LTP structures. This led them to uncover the existence of a non-occupied buffer zone, analogous to that previously reported for the ligand-binding cavity of enzymes. Additionally, they found that the volumes occupied by newly identified LTP ligands was similar to those of known ligands. As functional benchmark, they used LC-MS/MS to measure how the lipidome in HEK293 cells was affected by overexpression of individual LTPs. This revealed that cellular levels of newly identified lipid ligands were as frequently affected as those of previously known lipid ligands. These benchmarks are valuable additions to the study as they underscore the quality and functional relevance of the reported systematic analysis of lipid-binding properties of 39 LTPs. As outlined below, the revised manuscript also raises some questions and concerns, which I believe the authors should be able to address w/o additional experimental work.

1) For the functional benchmark, the authors found that a significant fraction of the ligands identified in the in vitro assay were supported by overexpression data (36%). This is lower than for the in cellulo assay (58%) but still significantly higher than what is observed when all possible LTP-lipid pairs are considered (22%; Fig. 1d), indicating that both the in vitro and in cellulo approaches yield functionally relevant LTP-ligand interactions. It would be relevant to know how the in cellulo and in vitro data sets compare with respect to the structural benchmark. This comparison is lacking from the revised manuscript.

2) PC was identified as putative novel lipid cargo of CERT in the in cellulo assay. Using a fluorescence emission assay, the authors provide experimental evidence that CERT can bind PC in vitro. Previous in silico simulations showed that PC may act as a cofactor to facilitate ceramide release (Moqadam et al., 2024; Ref. 10), providing additional proof that PC is an authentic CERT binding partner. However, whether CERT mediates PC transfer remains to be established. The observed decrease in cellular PC levels upon CERT overexpression (Fig. 2b) does not necessarily reflect the physiological relevance of the observed PC-CERT interaction (as suggested by the authors on p. 10 lines 257-260), but rather a consequence of increased consumption of PC as headgroup donor in sphingomyelin production.

3) Identification of TAG as putative novel lipid cargo of CERT is striking. This finding raises a series of questions, for instance how TAG would be accommodated in a lipid-binding pocket tuned for ceramide, a lipid with a small hydrophilic headgroup. Also, how would CERT gain access to and extract TAG from cellular membranes and how can a drop in cellular TAG levels upon CERT overexpression be explained in the context of an accelerated sphingomyelin production. I do not expect the authors to answer these questions. However, it would be helpful if they add at least a bit of context to their identification of TAG as novel lipid binding partner of CERT given that they explicitly refer to this discovery multiple times in the manuscript.

4) In the Introduction (p. 2, line 51) and Discussion (p. 13, line 352) the authors refer to the crucial but largely elusive membrane-induced mechanisms that ensure vectorial transport of lipids. What mechanisms are they referring to? And how do they believe their system-wide analysis of LTP-lipid pairs will help advance our understanding of how lipids are mobilized inside cells. The manuscript leaves some room for improvement to stipulate this in more detail.

(Remarks on code availability)

Referee #4

(Remarks to the Author)

The authors have performed a major revision of the original manuscript, adding several experiments, MD simulations, and revising the text accordingly. In this revision, they have satisfactorily addressed my concerns by incorporating a substantial amount of orthogonal data, such as HEK overexpression lipidomics data and MD simulations, which adds another dimension to the novelty and generalizability of some of their findings. They also addressed concerns regarding the mass spectrometry method, highlighting unusual species (e.g., very long-chain spingoid bases containing ceramides). They also addressed several concrete examples that showcase the potential impact of the manuscript as a resource. Although the concern regarding the cellular lipid microenvironment and its effect on LTP functions has not been addressed, the BioRxiv manuscript they refer to in the rebuttal probably complements this aspect. Reading the revised manuscript was a pleasure, and I am confident the communities of lipid cell biology, lipid metabolism and trafficking, and lipidomics will read the manuscript with great interest and will build several projects on its findings.

(Remarks on code availability)

Referee #5

(Remarks to the Author)

I co-reviewed this manuscript with one of the reviewers who provided the listed reports.

(Remarks on code availability)

The simulations were performed using the NAMD software package (<https://pubmed.ncbi.nlm.nih.gov/32752662/>), developed and maintained by the Tajkhorshid group at the University of Illinois at Urbana-Champaign. It is an open-source, free-of-charge simulation code focused on biological molecular systems. The current version was published in 2020 and has been cited approximately 2600 times (Google Scholar), indicating its widespread use.

Version 2:

Reviewer comments:

Referee #2

(Remarks to the Author)

The specific points made regarding the presentation of the experimental data provided have been adequately addressed.

Regarding the four points raised on the simulation data added during the manuscript revision, the authors have provided

additional data as requested on points 1, 3 and 4. Point 2 that specifically queried the energetic requirements related to the claimed lipid mobilization(s), the authors responded by toning down the text by replacing "mobilization" by "binding". The authors argued that the approaches required are not tractable or suitable for validating several new protein-ligand pairs at the scale done in this study. Admittedly, these approaches are resource intensive and require complex collective variables and are therefore not feasible to probe for all the new protein-lipid binding pairs identified. This reviewer's request was made with the hope that achieving this for even some of the novel LTP action(s) claimed should provide new mechanistic insights into lipid transfer by LTPs. Knowing how significant added value simulations can bring to experimental research, the simulation results reported in the manuscript unfortunately fall far short of what simulations could have demonstrated.

(Remarks on code availability)

Referee #3

(Remarks to the Author)

In the revised manuscript, the authors thoughtfully and carefully addressed my previous concerns. The work provides compelling evidence that multi-lipid binding is a common feature among LTPs and points at previously unrecognised mechanisms for metabolic cross-regulation of different lipid classes. The work already served as an important resource for a recent influential study aimed at imaging transport routes of individual lipids in cells and mapping their metabolism (Iglesias-Artola et al. 2025, Nature 646, 474-482). I have no doubt it will inspire other systematic efforts to unravel the entire cellular network of LTP-dependent lipid fluxes.

(Remarks on code availability)

Referee #5

(Remarks to the Author)

I co-reviewed this manuscript with one of the reviewers who provided the listed reports.

(Remarks on code availability)

June 5th, 2025

Dear _____,

We thank you for your interest in our work and for the reviewers' comments, which we find valuable and constructive. We have addressed them, while remaining within the scope of this manuscript, with substantial new data that support the new concepts and insights:

- We introduced two systematic benchmarks to assess the quality and functional relevance of the resource. A structural benchmark, in which we determined how ligand sizes fit into LTP lipid-binding pockets. We report the existence of a non-ligand occupied “buffer zone” similar to that documented in the ligand-binding cavity of enzymes, and that this zone is similar for both newly discovered and previously known LTP-ligand pairs. A functional benchmark to test the extent to which a gain in LTP function affects the cellular abundance of their respective cargoes (~260 lipidomic experiments). Cellular levels of newly identified ligands were as frequently affected as those of known cargoes. The *in vitro* and *in cellulo* data scored similarly in this benchmark assay. This reinforces the conclusion that the two datasets are complementary and that systematic biochemical approaches to human LTPs are feasible, with the data obtained being relevant to the lipid transfer function of LTPs. This is a new section “*Quality and functional relevance of datasets*” pages 5-6, new Figs. 1b-1d, Extended Data Fig. 2a and Extended Data Tables 7A-7D and 8.
- We have demonstrated that lipid species with shorter, mono- or bi-saturated fatty acids were the species primarily affected by gains in LTP function in HEK293 cells. These results reinforce our initial finding on the general properties of the lipidome mobilized LTP and support the conclusion that not all species are equally mobilizable by the LTPs studied (new Fig. 5b).

We have described LTPs, CERT and PITPs, whose specificity for acyl chains deviated from the trends described above. In our assays, they preferred ceramide and phosphatidylinositol species respectively, whose fatty acids were known to define pools with distinct functions (Figs. 6a and 6c). We have acquired further evidence in support of these observations and proposed molecular mechanisms for this selectivity (new Figs. 6b and 6d, and Extended Data Figs. 5d and 6b). This can be found in a new section, “Discrete specificities for aliphatic chains” on pages 11-13.

- We have carried out experiments to support the examples that illustrate that the resources are useful sources of new insights. Ligands for orphan LTPs, new lipids in the LTP system and new players in sphingolipid transports are now supported by: cell-based LTP gain-of-function assays (HSDL2, CERT, STARD2, STARD10, SEC14L2)(new Fig. 2b), molecular dynamics simulations (LCN1, CERT, STARD2, STARD10,)(new Figs. 2c and new Extended Data Figs. 4a, 4b and 5d), bioinformatics analyses (PITPNA, PITPNB) (new Fig. 6d and Extended Data Fig. 6b) and new biochemical assays (CERT)(new Fig. 2d). These new data are described on pages 7-8, 10 and 11-13.

In the revised version of the main text, we have highlighted the changes in grey. Below is a detailed, point-by-point response to how we have addressed the reviewers' concerns and incorporated the new data into the revised manuscript.

Yours sincerely,

Anne-Claude Gavin, on behalf of the authors

Referee #1

This study systematically determines the lipids bound by the majority of known human lipid transport proteins (LTPs). Given how challenging it can be to find which lipids are bound by an LTP and that the lipids bound by many LTPs were unknown, this study is a technical tour de force. The findings will be an important resource for those investigating LTPs, lipid metabolism, and lipid signaling. The authors identify two broad conceptual advances provided by their work: many LTPs can recognize both the head groups and acyl chains of the lipids they bind and many LTPs bind more than one class of lipids. While these are notable conclusions, they are not terribly surprising since they already been shown to be true of several LTPs families, such as the ORPs and PITPs.

We thank the reviewer for considering the work a technical tour de force and the data an important resource. We would like to follow on from the reviewer's statement *"Given how challenging it can be to find which lipids are bound by an LTP and that the lipids bound by many LTPs were unknown, this study is a technical tour de force"* and propose that another conceptual advance is that such systematic biochemical approaches to human LTPs are feasible. In the revised version, we have reinforced this notion with a comprehensive benchmark that supports the quality and relevance of LTP-lipid pairs in cellular transport (see below). We believe that the approaches should inspire/facilitate future systematic cell biology or structural work, and analyses in different biological contexts. An idea also expressed by reviewer 3 *"The study as a whole is a tour-the-force, yielding not only many novel lipid-binding partners but also establishing an important methodological pipeline for identifying protein-bound lipids and exploring protein-lipid interactions in different cell types and under distinct physiological conditions."*

While we agree with the reviewer that it is indeed known that LTPs can recognize specific lipid head groups, to our knowledge, the importance of acyl chain size and saturation in defining binding specificity has remained largely understudied. It is mainly due to the difficulty to test a large number of lipid species in classical biochemical assays, which remain limited to a small number of purified or synthetic lipids. The data provided capture the lipid species preference at the scale of entire lipidomes of a large number of LTPs (covering nine classes of LTP). This is to our knowledge an unprecedented scale. The fact that the LTP system has preferences for lipids with shorter acyl chains and 1-2 unsaturation is novel and shows for that not all species are equally mobilizable by LTPs. For the revised version, we have acquired new experimental evidences that support this new concept. We demonstrate that species with shorter and mono- or bi-saturated acyl chains are the species primarily affected by gains in LTP function in HEK293 cells (see below).

A few LTPs were indeed known to bind to more than one lipid, i.e. cargoes but also cofactors, chaperones or exchange currencies for the vectorial transport (of the cargoes). Our analysis is on a different scale, covering nine evolutionarily distinct families that have different structures/folds. Based on a body of experimental evidence, we find that most of them share this same attribute. The formation of complexes with several lipids - cargoes, cofactors, chaperones or currencies - seems to be at the heart of their biochemical function. We believe that this to be an important discovery, and our data, describing how LTPs can mobilize/manipulate lipids at membranes, will inspire future work.

The work is well done. While it is likely to an important resource, it does not have a big enough conceptual advance for it to be appropriate for a high impact journal. The study would be strong if it more definitively showed how the new binding data allows novel biological insights. It comes closest it its examination of CERT function, but the interesting new ideas about the roles of CERT in complex sphingolipid metabolism are not verified by in cell lipid trafficking measurements and lipid flux analyses.

We realize our lack of clarity in the first version regarding the scope of the work. Due to the approaches taken, the conceptual advances are at molecular and biochemical levels, and not so much at the cellular ones. We address LTPs biochemical function *in cellulo* and *in vitro*, which is highly relevant. The ability of LTPs to selectively extract lipids from membranes and assemble stable, soluble LTP-lipid complexes is an important biochemical property. This is the first step in a series of membrane-induced mechanisms, still poorly understood, which ensure the vectorial transport of lipids. For a few LTPs, we know that they involve interactions with other lipids entering their lipid-binding cavity, i.e. cofactors or exchange currencies. At the biochemical/molecular level, the work brings new concepts that we find interesting and that could not have arisen from one-at-a-time studies (see above). On these molecular aspects, we followed the reviewer's suggestions and have extended the structure-based analyses, molecular dynamic simulations, cell-based and biochemical assays to benchmark, validate and demonstrate that the dataset represents a valuable source of information (see details below).

While we agree that using the LTP-ligand map as a blueprint for measuring cell lipid trafficking and lipid fluxes is an interesting route to elucidate LTP function, the organelles involved and the metabolic fates of cargoes, it is currently beyond the scope of this manuscript. New approaches based on chemical engineering of lipids are emerging (as in André Nadler's group, doi.org/10.1101/2024.05.14.594078) and would be ideal to follow up on these questions in a systematic manner, also based on the resource described here. Nevertheless, we thought we would extend our data in the direction suggested by the reviewer, and measured the effect of gain of function of individual LTPs, by overexpression, on the total HEK293 lipidome (~260 individual experiments). We expect that a gain in LTP function (i.e. perturbing the fluxes of its cargoes) will affect the abundance of the respective cargoes or their metabolic products. This allowed us to address (and validate) the functional relevance of newly identified LTP-ligand pairs (especially those seen *in vitro*) in a cellular context (see details below).

We realize that we did not clearly delimit the scope and aspect of the novelty in the original version. The revised version has been considerably expanded and reorganized to remedy these points. We now explicitly define the scope of this work on page 2 (last paragraph) and page 3 (1st-2nd paragraph), and the novelty aspect throughout the result section. Here are the details of the actions we have taken to demonstrate more definitively that both approaches lead to new insights:

1) Overall assessment of the quality and relevance of the new binding data to the LTP transfer function. This benchmark is based on the integration of both systematic structural and functional data. The idea is to compare the *in vitro* assay with the *in cellulo* assay, and the newly identified ligands with the previously known ligands, and determine whether their quality and relevance are similar. This is described in a new section entitled "*Quality and functional relevance of datasets*" (pages 5-6).

- The structural benchmark is based on known or predicted structures used to estimate the volume of lipid-binding pockets in LTPs and determine how the sizes of lipid ligands observed in our assays fitted in these volumes (new Figs. 1b and 1c, new Extended Data Fig. 2a and new Extended Data Table 7A). We show that the volumes occupied by previously known and newly discovered LTP ligands were very similar (p-value: 0.25), both occupying less than 42.5% of the pocket, revealing the existence of a "buffer zone", similar to that documented in the ligand-binding cavity of enzymes.

- For the functional benchmark we measured how the lipidome of HEK293 cells is affected by a gain in LTP function (new Fig. 1d and new Extended Data Tables 7B and 7C). We postulated that a gain in LTP function (i.e. perturbing the fluxes of its cargoes) would affect the abundance of the respective cargoes or their metabolic products. And indeed, the abundance of 44% of the ligands we identified was affected by overexpression of their respective LTP, which is significantly higher than what is observed

when all overexpression data are considered (22%; p-value: 1.05×10^{-5}). Importantly, the set of newly discovered ligands was as often affected by overexpression of the corresponding LTP (47%) as the set of previously known cargoes (40%), supporting the idea that the two sets are of similar quality. Also, a significant fraction of the ligands identified in the *in vitro* assay were supported by overexpression data (36%; p-value: 2.35×10^{-2}). This figure is lower than for the *in cellulo* assay (58%; p-value: 5.23×10^{-7}) and may suggest that some of these complexes do not assemble in cells. To facilitate follow-up studies, benchmark results are documented in a new Extended Data Table 8.

These benchmarks show that the set of newly discovered LTP-ligand pairs scored similarly to those in the set of previously known pairs. This supports the new concept that such systematic biochemical approaches to human LTPs are feasible, and the data obtained are pertinent to the lipid transfer function of LTP.

2) Additional experiments to support new biological insights. The original manuscript included several examples illustrating how binding data contribute to new biological insights. The revised version includes new validation experiments: LTP gain-of-function assay in cells (new Fig. 2b), molecular dynamics simulations (new Figs. 2c and 6b and new Extended Data Figs. 4a, 4b and 5d), structural bioinformatics analyses (new Fig. 6d and Extended Data Fig. 6b) and new biochemical assays (new Fig. 2d). In details:

- HSDL2 is a previously orphan LTP for which we identified ligands, HSDL2-triacylglycerol (TAG): This is now further supported by the observation that the overexpression of HSDL2 in HEK293 led to significant increases in the abundance of TAG and diacylglycerol (DAG) species (a metabolite of TAG) (new Fig. 2b).

- Lipids that were not known to be part of the LTP system:

SEC14L2/DAG: now supported by the observation that overexpression of SEC14L2 in HEK293 cells led to a significant decrease in the cellular levels of DAG species (new Fig. 2b and Extended Data Tables 7C and 8).

LCN1/sphingomyelin (SM): LCN1 has specificity for choline-containing lipids, phosphatidylcholine (PC) (known ligand) and SM (new ligand). In this case, we validated this observation using molecular dynamics simulation and could identify the binding site for the choline headgroups in the hydrophobic cavity of LCN1. We also compared it to the binding pockets of STARD2 and STARD10, which also bind PC, but not SM. Our model supports the notion that the shape of the binding site accounts for this difference in specificity. LCN1 implies binding of its cargoes in an elongated conformation. In contrast, STARD2 imposes a bend of the PC headgroup, with the conserved phosphate-binding arginine (Arg78) being placed deeper in the pocket than the choline-binding aromatics, a conformation that is unlikely to be adopted by the sphingosine backbone of sphingomyelin (Fig. 2c and new Extended Data Figs. 4a and 4b).

We have included a new example, ether-linked lipids, which synthesis is known to involve a distinct pathway (than ester-linked ones) and their intracellular transport is mainly by non-vesicular routes, but the transport system is elusive. We observed that STARD2 and STARD10 mobilized ester PC and PC/phosphatidylethanolamine (PE), respectively, as well as their ether species (Fig. 2a). We performed molecular dynamics simulation that supported the notion that STARD2 can mobilize ether- and ester-PC alike (Extended Data Fig. 4a). We have also confirmed the relevance of these interactions in a cellular context where overexpression of STARD2 or STARD10 significantly affected not only PC and PE ester but also ether species (Fig. 2b and Extended Data Tables 7C and 8).

- LTPs bind to more than one class of lipids. We now illustrate this concept with LTPs in sphingolipid transport. These transports imply well-studied LTPs mobilizing well known cargoes, but we identify

new ligands. CERT is known to transfer ceramides, and in *in silico* simulations, PC seems to act as a cofactor entering the lipid-binding cavity and facilitating ceramide release (DOI: 10.1021/acs.jpcb.4c02398). We observed *in cellulo* complexes with ceramide as well as PC and TAG. We could confirm binding to PC using an *in vitro*, fluorescence emission shift assay (new Fig. 2d). In addition, *in cellulo*, we show that CERT gain-of-function resulted in a significant increase in a ceramide metabolite, SM, as well as a decrease in PC and TAG species (new Fig. 2b, and new Extended Data Tables 7C and 8).

3) LTP recognition of lipids with specific acyl chain length and/or unsaturation.

- General properties of the LTP-mobilized lipidome. This analysis has led to the concept, that not all species are equally mobilizable by LTPs, which show general preferences for glycerophospholipids with shorter acyl chains and 1-2 unsaturation. We have acquired new experimental evidences supporting this new concept. We demonstrate that species with shorter and mono- or bi-saturated acyl chains are the species primarily affected by gains in LTP function in HEK293 cells. These new data are in new Fig. 5b (previous Fig. 2), and described on page 11 (2nd paragraph).

- We have added a new section, “Discrete specificities for aliphatic chains” starting page 11. It concerns examples of LTPs (CERT/ceramide and PITPs/phosphatidylinositol (PI)) with acyl chain preference deviating from the general trends described above. We have rewritten this section (on pages 11-13) to define how the binding data contribute to current knowledge, and we have carried out additional validation experiments. Within these two lipid classes, ceramides and PI, specific acyl chains are known to define pools with distinct functions. The enzymes involved in the metabolism of these lipids often exhibit acyl chain specificity, which contribute to define the pools. The question of whether LTPs share similar attributes remains largely open, and we thought that this resource, because of its scope encompassing an unprecedented variety of lipid species, might help answer that question.

In vitro, CERT is known to be specific for ceramides containing 14-20 carbon fatty acids, defining a pool of ceramide for sphingomyelin synthesis, whereas, ceramides containing 22-26 carbon fatty acids are not transported by CERT and are destined to hexosylceramide synthesis. Our data showed that CERT-ceramide complexes assembled *in cellulo* had a similar selectivity for short-and medium-chain ceramide but not long-chain ceramides (Fig. 6a and Extended Data Table 4). Sphingomyelin and ceramide 1-phosphate species associated with GLTPD1 predominantly contained 14-20 carbon fatty acids, whereas GLTP-associated hexosylceramides are predominated by 22-26 carbon fatty acid species (Fig. 6a). We observed that a gain of CERT function led to an increase in sphingomyelin levels in HEK293 and in HeLa cells, showing that this effect is conserved in different cell types (Fig. 2b, Extended Data Fig. 5a and Extended Data Tables 7B and 11). Although anticipated, these findings validated the capacity of our approach to recover the characteristic lipid species associated with each system. More intriguing, CERT also assembled *in cellulo* with rare, saturated and very long dihydroceramides and phytoceramides with 46 and 48 carbons (24-, 26-carbon FA and 22-, 24-carbon spingoid base)(Fig. 6a and Extended Data Fig. 5b). These species were not observable in the total lipidome and we were unable to test the impact of CERT overexpression on their abundance. They were not often recorded in spectral libraries, suggesting that they are rare low abundant species (Extended Data Fig. 5c). However, with molecular dynamics simulations we were able to show that the CERT-START domain can accommodate the long lipid tails of these large cargoes while maintaining the positioning of their headgroups within the known ceramide-binding site (new Fig. 6b and Extended Data Fig. 5d). This suggests that CERT has broader specificity, and may define a third ceramide pool.

The 2nd example is PI, where species containing arachidonoyl (C20:4) define a pool dedicated to the PI cycle (reloads the plasma membrane with PI after phospholipase C-mediated PI(4,5)P2 hydrolysis). We observed that members of the class I PITP family – PITPNA and PITPNB – preferentially bound

arachidonoyl-containing phosphatidylinositols, PI(38:4), *in vitro* (Fig. 6c and Extended Data Fig. 6a). A smaller fraction of phosphatidylinositol (36:2) present in the total lipidome was mobilized by PITPNA, but not seen bound to PITPNB. By aligning their amino acid sequences, we identified a conserved cluster of several aromatic amino acids (mainly phenylalanines) located at the bottom of the PITPs binding pocket, close to the gate and well positioned to form an interaction site with arachidonic acid unsaturations (Fig. 6d and Extended Data Fig. 6b). Interestingly, the selectivity of a cytosolic phospholipase A2 for arachidonoyl-containing lipids is known to involve a similar group of phenylalanines interacting with the four double bonds of arachidonic acid (DOI:10.1021/jacs.7b12045). Such a cluster was not found in the lipid-binding site of SEC14L2 (Fig. 6d and Extended Data Fig. 6b), another phosphatidylinositol-binding protein, which binds a wide range of different species (Fig. 6c and Extended Data Fig. 6a). It was also absent from STARD2 and STARD10 (Fig. 6d and Extended Data Fig. 6b), which are structurally related to PITPs and also bound – even though marginally - to arachidonoyl-containing phosphatidylcholine, but also many other species (Figs. 6c and Extended Data Fig. 6a). Several PI cycle enzymes, known to exhibit C20:4 preference, contribute to the maintenance of lipids with this acyl chain in the cycle. Our data show that this may also apply to transport systems.

The study also makes claims about lipid movement in cells that are not supported by experimental evidence. Lipid mobility in cells, contrary to the claim of the title, is not measured. The results section has interesting speculation about routes of lipid transfer in cells, but they are not tested.

We realize that we overstated the scope of this work and mentioned claims about lipid movements that we did not directly measure. To address this point, we now precisely define the scope of this work on page 2 (last paragraph) and page 3 (1st and 2nd paragraph). We removed all overstatements concerning lipid movement, transport or transfer and replace them by lipid mobilization or lipid binding. In the title, we replaced “mobility” by “mobilization”. We did the same for the term cargo, which was replaced by ligands or lipid binders. All these changes are highlighted in grey in the new version.

Referee #2:

In this manuscript, the group of Anne-Claude Gavin and collaborators have carried out a systematic brute force effort to identify lipid cargoes for human lipid transfer proteins (LTPs). Two complementary approaches were applied: 1) inducible overexpression of affinity-tagged LTPs in HEK293 cells followed by purification and analysis of lipids stably associated with them, and 2) expression and purification of LTPs from *E. coli* and analysis of lipids extracted by them from 400 nm liposomes made of DOPC, bovine liver and porcine brain lipid extracts. From a total of 101 LTPs cloned and 95 expressed, 73 unique protein-lipid complexes were purified (of these, 37 proteins were from both cell lysates and LTP-liposome mixtures, the rest from either cells or liposome mixtures) and finally, lipid cargoes were identified for 39 unique proteins.

Overall, this work has an ambitious goal. It aims to generate a resource that would capture the general principles of non-vesicular lipid transport in humans. Yet, the number of novel lipid-protein interactions identified remained rather modest, with the majority being based solely on the *in vitro* approach, leaving reservations on the physiological relevance. Lipid ligands for nine LTPs with no previously known ligands were identified. In principle, the two approaches should strengthen each other. However, only six proteins extracted lipids in both approaches. It seems that only one of these represents a novel lipid-protein interaction: STARD2, a.k.a. PCTP (phosphatidylcholine transfer protein) was found to complex not only with PC, but also with ether-PC.

Furthermore, some of the general principles highlighted, e.g. that LTPs form complexes with more than one lipid class, are already well known, while some of the more exciting ideas proposed remain insufficiently documented and validated. Please see specific comments below.

The two approaches were not necessarily intended to strengthen/validate each other, but also to complement each other. Certain interactions missed *in cellulo* (expression levels, cargo abundance/presence, controlled LTP activity, etc) can be captured *in vitro*, and conversely, interactions missed *in vitro* (lipid loading requiring a cellular machinery, LTP activity (PTM), etc) can be captured *in cellulo*. In the revised version, we have addressed this point by including an overall benchmark in which we assess the quality of the new binding data, and also their relevance (particularly for those observed *in vitro*) to the transfer function of LTPs in a cellular context (see below). For the latter, we measured the effect of gain-of-function of individual LTPs, by overexpression, on the total lipidome of HEK293 (~260 individual experiments). We expect that a gain in LTP function (i.e. perturbing the fluxes of its cargoes) will affect the abundance of the respective cargoes or their metabolic products (see details below). One of the conclusions of this benchmark is that the *in cellulo* and the *in vitro* approaches are complementary (see below). This means that not only the intersection, but also the union of the two data sets must be considered.

Importantly, the numbers mentioned by the reviewer relates to LTP specificity for lipid sub-classes (i.e. head groups or linkages). However, beyond head groups, the datasets provide important information on the binding preference of LTPs for a large number of lipid species, covering entire lipidomes and representing > 750 LTP-lipid species pairs. To our knowledge, this scale is unprecedented, as only a defined number of lipid species could be tested in conventional biochemical assays. We observed that the LTP system has preferences for lipids with shorter fatty acids and 1-2 unsaturation, which is novel and shows that not all species are equally mobilizable by LTPs. For the revised version, we have acquired new experimental evidences that now support this new concept. We demonstrate that species with shorter and mono- or bi-saturated fatty acids are the species primarily affected by gains in LTP function in HEK293 cells (see below).

Following the reviewer's suggestions, we have considerably extended the revised version, with additional structure-based analyses, molecular dynamic simulations and biochemical assays to

benchmark, validate and demonstrate that the dataset represents a valuable resource (see below).

Major points:

1. Validation of *in vitro* lipid binding data in cells: The *in vitro* approach has considerable limitations. The human LTPs can already bind E.coli lipids and LTPs *in vitro* are devoid of cellular spatiotemporal control mechanisms, as also discussed by the authors. Therefore, it would be important to strengthen at least some of the novel lipid-protein interactions identified solely from the liposome approach, by additional functional cell-based data. As an example, HSDL2 was identified to bind TAG *in vitro*. The authors discuss that HSDL2 is a peroxisomal LTP and according to GeneCards, it is located in mitochondria and peroxisome. TAG would be a novel ligand in the LTP system and neither mitochondria nor peroxisomes are canonical sites of TAG transport or metabolism. Thus, it is not easy to understand what the role of this TAG-binding LTP should be. Additional experimental evidence from cellular systems supporting the *in vitro* binding would be very helpful.

Due to the approaches taken, we address LTPs biochemical function *in cellulo* and *in vitro*, which is highly relevant. The ability of LTPs to selectively extract lipids from membranes and assemble stable, soluble LTP-lipid complexes is an important biochemical property. This is the first step in a series of membrane-induced mechanisms, still poorly understood, which ensure the vectorial transport of lipids. While we agree that it is important to address the cellular function of LTPs, i.e. how they contribute to cellular lipid trafficking, fluxes, metabolism or organelle function, it is beyond the scope of this manuscript. New approaches based on chemical engineering of lipids are emerging and would be ideal to follow up on these questions in a systematic manner, also based on the resource described here.

Nevertheless, we thought we would extend our data in the direction suggested by the reviewer, and measured the effect of gain of function of individual LTPs, by overexpression, on the total HEK293 lipidome (~260 individual experiments). We have also strengthened both LTP-lipid binding datasets with additional experiments in order to: 1) broadly assess the performance of both approaches in terms of quality and relevance to the transfer function of LTPs in a cellular context, and 2) provide additional validation and evidence in support of new biological insights. In details:

1) Overall assessment of the quality and relevance of the new binding data to the LTP transfer function. This benchmark is based on the integration of both systematic structural and functional data. The idea is to compare the *in vitro* assay with the *in cellulo* assay, and the newly identified ligands with the previously known ligands, and determine whether their quality and relevance are similar. This is described in a new section entitled “*Quality and functional relevance of datasets*” (on pages 5-6).

- The structural benchmark is based on known or predicted structures used to estimate the volume of lipid-binding pockets in LTPs and determine how the sizes of lipid ligands observed in our assays fitted in these volumes (new Figs. 1b and 1c, new Extended Fig. 2a and new Extended Data Table 7A). We show that the volumes occupied by previously known and newly discovered LTP ligands were very similar (p-value: 0.25), both occupying less than 42.5% of the pocket, revealing the existence of a “buffer zone”, similar to that documented in the ligand-binding cavity of enzymes.

- For the functional benchmark we measured how the lipidome of HEK293 cells is affected by a gain in LTP function (new Fig. 1d and new Extended Data Tables 7B-7C). We postulated that a gain in LTP function (i.e. perturbing the fluxes of its cargoes) would affect the abundance of the respective cargoes or their metabolic products. And indeed, the abundance of 44% of the ligands we identified was affected by overexpression of their respective LTP, which is significantly higher than what is observed when all overexpression data are considered (22%; p-value: 1.05×10^{-5}). The set of newly discovered ligands was as often affected by overexpression of the corresponding LTP (47%) as the set of

previously known cargoes (40%), supporting the idea that the two sets are of similar quality. Importantly, a significant fraction of the ligands identified in the *in vitro* assay were supported by overexpression data (36%; p-value: 2.35×10^{-2}), showing that this approach also produces relevant LTP-ligand pairs and does not appear to be plagued with “considerable limitations”. This figure is lower than for the *in cellulo* assay (58%; p-value: 5.23×10^{-7}) and may suggest that some of these complexes do not assemble in cells. To facilitate follow-up studies, benchmark results are documented in a new Extended Data Table 8.

Overall, the benchmark brings important notions/concepts:

- The two approaches are complementary: both bring sets of high confidence LTP-ligand pairs (scoring similarly to the previously known ligands).

- Such systematic biochemical approaches to human LTPs are feasible, and the data obtained are pertinent to the lipid transfer function of LTPs. In this functional benchmark, the set of newly discovered LTP-ligand pairs performed similarly to those previously known.

2) Additional experiments to support new biological insights. The original manuscript included several examples illustrating how binding data contribute to new biological insights. The revised version includes new validation experiments: LTP gain-of-function assay in cells (new Fig. 2b), molecular dynamics simulations (new Figs. 2c and 6b and new Extended Data Figs. 4a, 4b and 5d), structural bioinformatics analyses (new Fig. 6d and Extended Data Fig. 6b) and new biochemical assays (new Fig. 2d). In details:

- HSDL2 is a previously orphan LTP for which we have identified an *in vitro* triacylglycerol (TAG) ligand: We confirmed the relevance of this interactions in HEK293 cells where overexpression of HSDL2 led to significant increases in the abundance of its novel ligand, TAG, and one of its metabolites, diacylglycerol (DAG) (new Fig. 2b). To answer the reviewer’s question, HSDL2 is part of a protein network consisting of proteins involved in mitochondrial/peroxisomal β -oxidation (Extended Data Fig. 3), and we propose on page 7 (second paragraph) that HSDL2-TAG complexes play a role in this process. As discussed above, addressing the organelles at play and the cellular function is beyond the scope of this manuscript.

- Lipids that were not known to be part of the LTP system:

SEC14L2/DAG: now supported by the observation that overexpression of SEC14L2 in HEK293 cells led to a significant decrease in the cellular levels of DAG species (new Fig. 2b).

LCN1/sphingomyelin (SM): LCN1 has specificity for choline-containing lipids, phosphatidylcholine (PC) (known ligand) and SM (new ligand). In this case, we validated this observation using molecular dynamics simulation and could identify the binding site for the choline headgroups in the hydrophobic cavity of LCN1. We also compared it to the binding pockets of STARD2 and STARD10, which also bind PC, but not SM. Our model supports the notion that the shape of the binding site accounts for this difference in specificity. LCN1 implies binding of its cargoes in an elongated conformation. In contrast, STARD2 imposes a bend of the PC headgroup, with the conserved phosphate-binding arginine (Arg78) being placed deeper in the pocket than the choline-binding aromatics, a conformation that is unlikely to be adopted by the sphingosine backbone of sphingomyelin (Fig. 2c and new Extended Data Figs. 4a and 4b).

We include a new example, ether-linked lipids, which synthesis is known to involve a distinct pathway (than ester-linked ones) and their intracellular transport is mainly by non-vesicular route, but the transport system is elusive. We observed that STARD2 and STARD10 mobilized ester PC and PC/phosphatidylethanolamine (PE) respectively, as well as their ether species (Fig. 2a). We performed

molecular dynamics simulation that supported the notion that STARD2 can mobilize ether- and ester-PC alike (Extended Data Fig. 4a). We have also confirmed the relevance of these interactions in a cellular context where overexpression of STARD2 or STARD10 significantly affected not only PC and PE ester but also ether species (Fig. 2b and Extended Data Tables 7C and 8).

- LTPs bind to more than one class of lipids. We now illustrate this concept with LTPs in sphingolipid transport. These transports imply well-studied LTPs mobilizing well known cargoes, but we identify new ligands. CERT is known to transfer ceramides, and in *in silico* simulations, PC seems to act as a cofactor entering the lipid-binding cavity and facilitating ceramide release (DOI: 10.1021/acs.jpcc.4c02398). We observed *in cellulo* complexes with ceramide as well as PC and TAG. We could confirm binding to PC using an *in vitro*, fluorescence emission shift assay (new Fig. 2d). In addition, *in cellulo*, we show that CERT gain-of-function resulted in a significant increase in a ceramide metabolite, SM, as well as a decrease in PC and TAG species (new Fig. 2b, and new Extended Data Tables 7C and 8).

- General properties of the LTP-mobilized lipidome. This analysis has led to the concept, that not all species are equally mobilizable by LTPs, which show general preferences for glycerophospholipids with shorter fatty acids and 1-2 unsaturation. We have acquired new experimental evidences supporting this new concept. We demonstrate that species with shorter and mono- or bi-saturated fatty acids are the species primarily affected by gains in LTP function in HEK293 cells. These new data are in new Fig. 5b (previous Fig. 2), and described on page 11 (second paragraph).

2. One of the interesting ideas proposed relate to new principles of sphingolipid transport, with three pools of ceramide with different metabolic fates pending on chain-length and co-operation of transfer proteins. However, the overall concept and its implications are not clearly enough articulated and the experimental evidence in support is rather preliminary. The authors did carry out a small pilot experiment where they overexpressed CERT and analyzed its effects on the HEK293 cell lipidome, then reporting on increased total cellular sphingomyelin and ceramide 1-phosphate and decreased hexosylceramides (Fig. 6b). It would be important to elucidate the lipidome changes in more detail, e.g. the chain compositions, other lipids including other identified CERT ligands, whether the levels of other LTPs are adjusted in response to CERT overexpression, and if the effects of CERT overexpression are specific to the HEK293 model or more general. It seems that the CERT simulation data (Fig. 6c) provides more questions than answers, e.g. how does the presence of POPC affect the affinity for ceramide of different chain lengths, how strong are these interactions etc.

CERT is now part of a new section, “Discrete specificities for aliphatic chains” starting page 11. It concerns examples of LTPs (CERT/ceramide and PITPs/phosphatidylinositol (PI)) with acyl chain specificity deviating from the general preferences observed for other LTPs (which generally prefer lipids with shorter and mono- or bi-unsaturated fatty acids). Within these two lipid classes, ceramides and PI, specific fatty acids are known to define pools with distinct functions. The enzymes involved in the metabolism of these lipids often exhibit acyl chain specificity, which contribute to define the pools. The question of whether LTPs share similar attributes remains largely open, and we thought that this resource, because of its scope encompassing an unprecedented variety of lipid species, might help answer that question.

We have rewritten this section on pages 11-13 to define how the binding data contribute to current knowledge, and we have carried out additional validation experiments. We also removed the CERT-phosphatidylcholine simulation data, old Fig. 6c.

In vitro, CERT is known to be specific for ceramides containing 14-20 carbon fatty acids, defining a pool of ceramide for sphingomyelin synthesis, whereas, ceramides containing 22-26 carbon fatty acids are

not transported by CERT and are destined to hexosylceramide synthesis. Our data showed that CERT-ceramide complexes assembled *in cellulo* had a similar selectivity for short- and medium-chain ceramide but not long-chain ceramides (Fig. 6a and Extended Data Table 4). Sphingomyelin and ceramide 1-phosphate species associated with GLTPD1 predominantly contained 14-20 carbon fatty acids, whereas GLTP-associated hexosylceramides are predominated by 22-26 carbon fatty acid species (Fig. 6a). We observed that a gain of CERT function led to an increase in sphingomyelin levels in HEK293 and in HeLa cells, showing that this effect is conserved in different cell types (Fig. 2b, Extended Data Fig. 5a and Extended Data Tables 7B and 11). Although anticipated, these findings validated the capacity of our approach to recover the characteristic lipid species associated with each system. More intriguing, CERT also assembled *in cellulo* with rare, saturated and very long dihydroceramides and phytoceramides with 46 and 48 carbons (24-, 26-carbon FA and 22-, 24-carbon spingoid base)(Fig. 6a and Extended Data Fig. 5b). These species were not observable in the total lipidome and we were unable to test the impact of CERT overexpression on their abundance. They were not often recorded in spectral libraries, suggesting that they are rare low abundant species (Extended Data Fig. 5c). However, with molecular dynamics simulations we were able to show that the CERT-START domain can accommodate the long lipid tails of these large cargoes while maintaining the positioning of their headgroups within the known ceramide-binding site (new Fig. 6b and Extended Data Fig. 5d). This suggests that CERT has broader specificity, and may define a third ceramide pool.

The 2nd example is PI, where species containing arachidonoyl (C20:4) define a pool dedicated to the PI cycle (reloads the plasma membrane with PI after phospholipase C-mediated PI(4,5)P₂ hydrolysis). We observed that members of the class I PIP family – PITPNA and PITPNB – preferentially bound arachidonoyl-containing phosphatidylinositols, PI(38:4), *in vitro* (Fig. 6c and Extended Data Fig. 6a). A smaller fraction of phosphatidylinositol (36:2) present in the total lipidome was mobilized by PITPNA, but not seen bound to PITPNB. By aligning their amino acid sequences, we identified a conserved cluster of several aromatic amino acids (mainly phenylalanines) located at the bottom of the PITPs binding pocket, close to the gate and well positioned to form an interaction site with arachidonic acid unsaturations (Fig. 6d and Extended Data Fig. 6b). Interestingly, the selectivity of a cytosolic phospholipase A2 for arachidonoyl-containing lipids is known to involve a similar group of phenylalanines interacting with the four double bonds of arachidonic acid (DOI:10.1021/jacs.7b12045). Such a cluster was not found in the lipid-binding site of SEC14L2 (Fig. 6d and Extended Data Fig. 6b), another phosphatidylinositol-binding protein, which binds a wide range of different species (Fig. 6c and Extended Data Fig. 6a). It was also absent from STARD2 and STARD10 (Fig. 6d and Extended Data Fig. 6b), which are structurally related to PITPs and also bound – even though marginally - to arachidonoyl-containing phosphatidylcholine, but also many other species (Figs. 6c and Extended Data Fig. 6a). Several PI cycle enzymes, known to exhibit C20:4 preference, contribute to the maintenance of lipids with this acyl chain in the cycle. Our data show that this may also apply to transport systems.

3. Another interesting suggestion made is the potential organization of LTPs into functional networks linking distinct organelles and contributing to the coordination of metabolism between different cellular compartments. This is based on the observation that lipid pairs mobilized by the same LTPs should be co-regulated and co-localized. By scoring such co-occurrence from available datasets, the authors indeed find evidence for this. The problem is that the subcellular co-localization data – that turned out to be the most significant for the conclusion – is not openly available. In the methods, the authors state that this is a dataset containing lipidomes of organelles affinity-purified with antibodies in a manuscript in preparation. It would be important to make this dataset publicly available in the context of the present manuscript, as it is critical for the conclusions and would align with the authors' idea of a resource article.

We apologize for not making this data available. It is difficult to synchronize articles between different laboratories. The colocalization data for all lipid pairs used in Fig. 4c is now listed in a new Extended Data Table 9A. Kenji Maeda's group will submit their manuscript with the lipidomic datasets to journals and BioRxiv in the coming weeks. We will update our manuscript with the BioRxiv DOI as soon as they become available.

Additional points:

4. Are the very short chain PC and PE species of HEK293 cells (e.g. C14, C16 in Fig. 2b) lysolipids? The authors identified 14 potential new lysolipid interacting proteins using the *in vitro* assay. How does a single vs. two acyl chains affect the extractability from membranes?

The C14 and C16 lipids in Fig. 2b (now Fig. 5a) are indeed mainly lyso-lipids. Among them we also have very short chained PC and PE. They were found bound *in cellulo* to BPI and BPIFB2 (Extended Data Table 4). This information is now added to the new Fig. 5a, where we flag these short-chain lipids and explain their nature in the legend.

We do not really know how the distribution of carbons between a lyso-lipid and a lipid (with the same total number of carbons) would affect extractability. However, this had no impact on our analyses of the general properties of the lipidome mobilized by LTP, as we focused on glycerophospholipids with two acyl chains. That said, we do note that some of these species appear to be differentially represented in the total lipidome compared to the mobilized lipidome. It's a point we haven't followed up.

5. In Figs 3c and 6a, grey borders indicate the relative distribution of lipids present in HEK293 cells. Comparing the observed distribution of lipids *in vitro* to cellular lipids does not seem to make sense. Should these rather be compared to the relative distribution of lipids in the liposomes?

We have now measured the lipidome present in the liposomes used in the biochemical assays (Extended Data Table 10). This is now described on page 11 (first paragraph) of the revised manuscript. Previous Figs. 3c and 6a, now Figs. 6a and 6c, have been updated accordingly. The conclusions and main messages remain unaffected.

6. The authors found that in HEK293 cells, PI did not associate with PITPs but with SEC14L2. They discuss that this is probably because SEC14L2-PI complexes contribute to important functions in dividing cells that need to duplicate their organelles, whereas phospholipase signaling is probably not active in this context. What would these important functions be? Also, it seems unlikely that PLC signaling is not operating in these cells, see e.g. PMID 15741171.

The cells we studied are HEK293 exponentially growing and dividing cells. We thought that the duplication of, for example, the Golgi membranes requires the transport of PI to this organelle for phosphorylation to PI4P in the Golgi. We meant that SEC14L2 activity is housekeeping and should be present in cultured cells. This is in contrast to the PI cycle which implies PLC activation downstream of GPCR activation. Research on the PI cycle are based on HEK293 cells stably expressing the rat angiotensin receptor AT1 and exposed to an agonist, angiotensin II, which selectively triggers the hydrolysis of the PM pool of PI(4,5)P2 (<https://doi.org/10.15252/embr.202154532>). This probably explains why we did not see PI with PITPs *in cellulo*; cells were not stimulated.

This part has been rewritten and simplified on page 5 (first paragraph) of the revised manuscript.

Referee #3:

This study provides the first systematic analysis of the lipid binding properties of human lipid transfer proteins (LTPs), a large and structurally diverse class of soluble lipid transporters that coordinate spatially separated lipid metabolic pathways, maintain organelle-specific lipid compositions and govern lipid-mediated cell signaling. By coupling high-throughput affinity purification methods with LC-MS/MS-based lipidomics, the authors identified lipid cargoes for 39 unique human LTPs using two complementary approaches. The study as a whole is a tour-the-force, yielding not only many novel lipid-binding partners but also establishing an important methodological pipeline for identifying protein-bound lipids and exploring protein-lipid interactions in different cell types and under distinct physiological conditions. Interestingly, most LTPs analyzed were found to bind more than one class of lipids. Data base mining and correlation analysis revealed that lipid pairs mobilized by the same LTPs are significantly more co-regulated upon metabolic perturbation and co-localized than would be expected from random sets of lipid pairs. Based on these findings, the authors postulate that this lipid co-transport represents an important, but hitherto underestimated mechanism for the organization and integration of spatially segregated lipid metabolic pathways. While experimental validation of the latter concept would have been very welcome, the reported LTP-lipid interaction map and how it was established provide an important resource for future work aimed at unravelling the architecture and regulation of the highly interconnected metabolic lipid network.

We thank the reviewer for the supporting statements, in particular the fact that the approach and resulting datasets represent an important resource for future work, and for the constructive comments.

In the revised version, we have responded to the reviewer's remark about validation with additional experiments to benchmark the two datasets (see below), and to validate the new concepts and insights. The latter includes additional experiments in support of:

- the individual examples included in the first version of the manuscript: LTP gain-of-function assay on cells (new Fig. 2b), molecular dynamics simulations (new Figs. 2c and 6b and new Extended Data Figs. 4a, 4b and 5d), structural bioinformatics analyses (new Fig. 6d and new Extended Data Fig. 6b) and new biochemical assays (new Fig. 2d). These new data are described through-out the result section (pages 7-8, 10 and 11-13).
- the general properties of the LTP-mobilized lipidome. This analysis has led to the new concept that not all species are equally mobilizable by LTPs, which show general preferences for glycerophospholipids with shorter acyl chains and 1-2 unsaturation (new Fig. 5). We have acquired new experimental evidence in support of this new concept. We demonstrate that species with shorter and mono- or bi-saturated acyl chains are the species primarily affected by gains in LTP function in HEK293 cells. These new data appear in a new Fig. 5b and are described on page 11 (second paragraph).

However, the work also has some shortcomings that the authors need to comment on. To begin with, their study focuses exclusively on box-type LTPs and excludes bridge-like LTPs. While box-type LTPs contain an internal cavity that can accommodate one lipid molecule at the same time and shuttle between two membranes, bridge-like LTPs have an opening that extends along its entire length. The extended opening forms a seam that allows lipids to slide while the protein remains stationary positioned between two membranes. As bridge-like LTPs are more promiscuous regarding their lipid binding partners and often form large multimers, mapping their lipid binding properties using the approach described in this study will unlikely yield meaningful results. Nevertheless, it would be appropriate that the authors at least mention their existence in the context of their work.

According to the reviewer suggestion, we now mention that we focused on non-transmembrane box-like LTPs and excluded the few bridge-like LTPs, considering their bulk lipid transport via long hydrophobic grooves, on page 4 (first paragraph). We also quote the excellent review of Neuman et al. (<https://doi.org:10.1016/j.tcb.2022.03.011>).

For mapping the lipid-LTP interaction network, the authors used two complementary approaches. In one they measured the ability of LTPs expressed in Hek293 cells to associate stably with lipids in a physiological context of an intact human cell. In addition, they analyzed the ability of LTPs expressed in *E. coli* to extract lipids from artificial membranes prepared with total liver or brain lipid extracts. Even though the majority of the 39 LTPs analyzed were successfully expressed and purified in both expression systems (HEK293 and *E. coli*), only six extracted lipids in both assays. This is somewhat surprising. In addition to commenting on this, it would be useful if the authors explicitly document which LTP expressed in both systems extracted lipids in only one. This may be particularly relevant given that, as noted by the authors, many of the LTP-lipid complexes formed *in vitro* contain lipids that are abundant in bacteria (FA, PA, CL, PG, PE and their lyso-forms), suggesting that their assembly may in fact have occurred in the *E. coli* expression host. This notion is consistent with the finding that the bound lipids often have odd numbered acyl chains, which are more frequent in bacteria. As LTPs pre-loaded with bacterial lipids may not be able to sample the full range of potential lipid cargoes present in membranes composed of lipids extracted from liver and brain, the lipid binding profiles of LTPs assessed using the *in vitro* biochemical assay may be skewed.

The two approaches were not necessarily intended to strengthen/validate each other, but also to complement each other. Certain interactions missed *in cellulo* (expression levels, cargo abundance/presence, controlled LTP activity, etc) can be captured *in vitro*, and conversely, interactions missed *in vitro* (lipid loading requiring a cellular machinery, LTP activity (PTM), etc) can be captured *in cellulo*.

In the revised version, we have addressed this point by including an overall benchmark in which we assess the quality of the new binding data, and also their relevance (particularly for those observed *in vitro*) to the transfer function of LTP in a cellular context. The idea was to compare the *in vitro* assay with the *in cellulo* assay, and the newly identified ligands with previously known ligands, and determine whether the *in vitro* data shows limitation. This is described in a new section entitled “Quality, limitation and functional relevance of datasets” (pages 5-6). The benchmark is based on both systematic structural and functional data.

- The structural benchmark is based on known or predicted structures used to estimate the volume of lipid-binding pockets in LTPs and determine how the sizes of lipid ligands observed in our assays fitted in these volumes (new Figs 1b and 1c, new Extended Fig. 2a and new Extended Data Table 7A). We show that the volumes occupied by previously known and newly discovered LTP ligands were very similar (p-value: 0.25), both occupying less than 42.5% of the pocket, revealing the existence of a “buffer zone”, similar to that documented in the ligand-binding cavity of enzymes.

- For the functional benchmark we measured how the lipidome of HEK293 cells is affected by a gain in LTP function (new Fig. 1d and new Extended Data Tables 7B and 7C). We postulated that a gain in LTP function (i.e. perturbing the fluxes of its cargoes) would affect the abundance of the respective cargoes or their metabolic products. And indeed, the abundance of 44% of the ligands we identified was affected by overexpression of their respective LTP, which is significantly higher than what is observed when all overexpression data are considered (22%; p-value: 1.05×10^{-5}). The set of newly discovered ligands was as often affected by overexpression of the corresponding LTP (47%) as the set of previously known cargoes (40%), supporting the idea that the two sets are of similar quality. Importantly, a significant fraction of the ligands identified in the *in vitro* assay were supported by

overexpression data (36%; p-value: 2.35×10^{-2}), showing that this approach also produces relevant LTP-ligand pairs. This figure is lower than for the *in cellulo* assay (58%; p-value: 5.23×10^{-7}) and may suggest that some of these complexes do not assemble in cells. To facilitate follow-up studies, benchmark results are documented in a new Extended Data Table 8. We have also followed the reviewer's suggestion and added a new Extended Data Table 6, explicitly documenting the LTP expressed in the two systems that extracted lipids in only one of them.

The benchmark shows that both the *in vitro* and *in cellulo* data bring sets of relevant LTP-ligand pairs, the two datasets are complementary. This means that not only the intersection, but also the union of the two data sets must be considered. In addition, the set of newly discovered LTP-ligand pairs performed similarly to those in the set of previously known pairs. This confirms that such systematic biochemical approaches to human LTPs are feasible, and the data obtained are pertinent to the lipid transfer function of LTP.

The idea that the pre-loading of LTPs with bacterial lipids could affect the loading of cognate cargoes in subsequent biochemical assays is indeed valid. In our experience, the binding of these bacterial lipids is generally reversible and they are (partially) replaced by the mammalian cargoes (if available in liposomes). But of course, these observations are based only on the few cases where we knew the identity of the known cargoes, and the point made by the reviewer remains valid when it comes to the dataset as a whole. We added a comment on this effect on page 6, last paragraph.

The group of LTPs whose lipid binding profiles were characterized under physiological conditions inside Hek293 cells includes several representatives of lipocalins, a large but poorly characterized family of secretory proteins that can bind a range of hydrophobic substances. As the expression constructs used to produce these proteins in Hek293 cells contain an N-terminal affinity tag, they would fail to enter the secretory pathway and accumulate in the cytosol. Of note is that membranes of secretory organelles typically display asymmetric lipid distributions. Consequently, lipocalins mistargeted to the cytosol may fail to encounter their preferred lipid binding partners. For instance, the authors find a strong signal for PC and only a weak signal for SM associated with the lipocalin LCN1 purified from Hek293 cells (Fig. 3a). As SM almost exclusively populates the luminal leaflet of secretory organelles, a preferential binding of SM by LCN1 would be obscured by failure of the latter to enter the secretory pathway. The latter scenario could be experimentally validated using the *in vitro* approach with LCN1 produced in *E. coli*. However, this analysis is missing from the study.

We apologize for the confusion. The LTPs were not only tagged N-terminally. Those with a signal peptide have the tag at the C-terminus, so that they should enter the secretory pathway. We now clarify this point on page 1 first paragraph of the Supplementary Methods section. The position of the tag, as well as the oligo used for the cloning were listed in Extended Data Tables 1 and 2.

In cellulo, C-terminally tagged LCN1 should indeed enter the secretory pathway, where SM is available. We have rewritten this part, making it clear that the previously known ligand was phosphatidylcholine and the new ligand was sphingomyelin. We included new molecular dynamics simulations to support the new binding to sphingomyelin:

"Next, we used this resource as a starting point for molecular dynamics simulations to define the molecular determinants of lipid specificity. For example, three LTPs - a lipocalin, LCN1, and two StAR-related lipid transfer domain proteins (STARD), STARD2 and STARD10 - have distinct specificities for lipids with a choline head group. We observed that LCN1, the major lipid-binding protein in tears (a fluid rich in phosphatidylcholine and sphingomyelin), could bind to both sphingomyelin (another novel lipid in LTP transport) and phosphatidylcholine (the previously known cargo) in cellulo (Fig. 2a). In contrast, STARD2 and STARD10 were unable to mobilize sphingomyelin, either in cellulo or in vitro, but

bound to phosphatidylcholine (Fig. 2a). Using molecular dynamics simulation, we identified two aromatic amino acids in the hydrophobic cavity of LCN1 that form the binding site for the choline headgroups, where phosphatidylcholine and sphingomyelin are housed in an elongated conformation (Fig. 2c and Extended Data Fig. 4b). This contrasts with the binding site of STARD2, which imposes a bend of the phosphatidylcholine headgroup with the conserved phosphate-binding arginine (Arg78) being placed deeper in the pocket than the choline-binding aromatics (Extended Data Fig. 4a), a conformation unlikely to be adopted by the sphingosine backbone of sphingomyelin. In this case, the shape of the binding site defines specificities for lipid cargoes that share chemical similarities, but differ in structural flexibility.”

Finally, we would like to mention that lipid ionization efficiencies vary between lipid classes, making quantitative comparisons between them (in this case PC and SM) hazardous. To prevent readers from over-interpreting our data, and comparing abundance between different lipid classes, we have added a comment on how to use these data in the legend to Fig. 2a.

Referee #4:

The manuscript by Gavin et al. describes the largest functional characterization of 39 unique lipid transfer proteins in human cells. To achieve that, the authors successfully overexpressed 86 LTPs in HEK293 cells (for the *in cellulo* characterization) and 71 LTPs in *E. coli* (for the *in vitro* characterization). They then employed affinity purification and size exclusion chromatography (SEC) to purify fractions where the LTPs (bound to their cargo) elute. Afterward, they performed HPTLC followed by LC-MS lipidomics to identify the lipid bound to the specific LTPs purified. Using their computational pipeline, the authors analyzed the lipidomics datasets and identified co-elution patterns between SEC and LC-MS. Finally, they infer general properties of LTP-associated lipidome, suggest new routes of lipid transfer, generalize their properties to lipid classes, and claim new principles of sphingolipid transport. Overall, the manuscript represents a significant leap in the characterization of LTPs in human cells and offers an invaluable resource to the lipid trafficking and cell biology community. However, several cell biological, and analytical concerns must be addressed to make some of the very important claims warranted. More specifically.

We thank the reviewer for finding that the manuscript represents a significant leap in the characterization of LTPs in human cells and offers an invaluable resource to the lipid trafficking and cell biology community.

Biologically:

- The manuscript rightfully acknowledges, in several instances, that lipid transport depends on the local lipid environment of the respective LTPs. However, it utilizes the *in vitro* datasets in combination with the HEK 293 datasets to suggest a *bona fide* function of the purifiable 39 LTPs. This approach carries the inherent limitations that both *in cellulo* and *in vitro* functions, with that definition, will most likely not mimic the natural environment where these proteins are expressed and localized. This can lead to false positive and negative functional annotations, of which the rates are unknown. It is unclear how much of the reported cargo lipid for the novel LTPs will translate into a more native lipid environment where LTPs are expressed in the right organelle with the right lipid environment. A more systematic approach to report on the false positive/negative rate or the extent of the cargo dependence on the local lipid environment might be needed to address this. For example, i) the effect of subcellular localization of the LTPs, and thereby their local lipid environment, on their cargo, ii) the relationship of protein sequence or structure homology to function and propagating that knowledge to non-characterized LTPs, orthogonal validation, etc.

We agree that it would be interesting to develop systematic approaches to measure the effect of LTP subcellular localization on their ability to mobilize cargo, but this is beyond the scope of this manuscript. We believe that the resource will motivate further analyses and efforts to address these questions. Nevertheless, we have extended the work in the direction suggested by the reviewer, and included two general benchmarks. They are based on the integration of both systematic structural and functional data. The idea was to compare the *in vitro* assay with the *in cellulo* assay, and the newly identified ligands with the previously known ligands, and determine whether their quality and relevance are similar. This is described in a new section entitled “Quality and functional relevance of datasets” (pages 5-6).

- The structural benchmark is based on known or predicted structures used to estimate the volume of lipid-binding pockets in LTPs and determine how the sizes of lipid ligands observed in our assays fitted in these volumes (new Figs 1b and 1c, new Extended Data Fig. 2a and new Extended Data Table 7A). We show that the volumes occupied by previously known and newly discovered LTP ligands were very similar (p-value: 0.25), both occupying less than 42.5% of the pocket, revealing the existence of a “buffer zone”, similar to that documented in the ligand-binding cavity of enzymes.

- For the functional benchmark we measured how the lipidome of HEK293 cells is affected by a gain in LTP function (new Fig. 1d and new Extended Data Tables 7B and 7C). We postulated that a gain in LTP function (i.e. perturbing the fluxes of its cargoes) would affect the abundance of the respective cargoes or their metabolic products. And indeed, the abundance of 44% of the ligands we identified was affected by overexpression of their respective LTP, which is significantly higher than what is observed when all overexpression data are considered (22%; p-value: 1.05×10^{-5}). The set of newly discovered ligands was as often affected by overexpression of the corresponding LTP (47%) as the set of previously known cargoes (40%), supporting the idea that the two sets are of similar quality. Importantly, a significant fraction of the ligands identified in the *in vitro* assay were supported by overexpression data (36%; p-value: 2.35×10^{-2}), showing that this approach also produces relevant LTP-ligand pairs. This figure is lower than for the *in cellulo* assay (58%; p-value: 5.23×10^{-7}) and may suggest that some of these complexes do not assemble in cells. To facilitate follow-up studies, benchmark results are documented in a new Extended Data Table 8.

Overall, the benchmark brings important notions/concepts:

- The two approaches are complementary: both bring sets of high confidence LTP-ligand pairs (scoring similarly to the previously known ligands).
- Such systematic biochemical approaches to human LTPs are feasible, and the data obtained are pertinent to the lipid transfer function of LTP. In this functional benchmark, the set of newly discovered LTP-ligand pairs performed similarly to those previously known.

We've also considerably expanded the revised version, adding additional data to validate new concepts and ideas along the lines suggested by the receiver. This includes additional supporting experiments:

- Of the individual examples included in the first version of the manuscript: LTP gain-of-function assay on cells (new Fig. 2b), molecular dynamics simulations (new Figs. 2c and 6b and new Extended Data Figs. 4a, 4b and 5d), structural bioinformatics analyses (new Fig. 6d and Extended Data Fig. 6b) and new biochemical assays (new Fig. 2d). These new data are described through-out the result section (pages 7-8, 10 and 11-13).
- the general properties of the LTP-mobilized lipidome. This analysis has led to the new concept that not all species are equally mobilizable by LTPs, which show general preferences for glycerophospholipids with shorter acyl chains and 1-2 unsaturation (old Fig. 2, now Fig. 5). We have acquired new experimental evidence in support of this new concept. We demonstrate that species with shorter and mono- or bi-saturated acyl chains are the species primarily affected by gains in LTP function in HEK293 cells. These new data appear in a Fig. 5b and are described page 11 (second paragraph).
- A striking consequence of ignoring the inherent false positive/negative assumption underlying the whole experimental design is the manuscript's statement, "Not all lipid species can be easily manipulated by LTP systems." This statement ignores that 1) not all relevant lipids are detected and ii) not all LTPs are characterized (less than 50% of the initially cloned LTPs could be characterized in the manuscript). Both facts leave room for invalidating the statement and similar generalizations across the manuscript.

Our analysis broadly covers the diversity of LTPs, including nine evolutionarily distinct families (out of ten) with different structures/folds, which we believe represents a significant "sampling". Nevertheless, we have rephrased this statement "*Our analysis of 39 LTPs revealed their general*

preference for short, mono- or di- unsaturated lipids. These species are known to induce deep defects in membranes, which likely facilitate their extraction.” on page 14 (second paragraph).

- The manuscript relies on the gain of function (overexpression) as the sole mode of assigning function. A complementary approach with loss of function (knockdown or knockout of at least certain LTPs) can shed light on function in a more biological context in combination with gain of function experiments.

Our scopes are at molecular and biochemical levels. We address LTPs biochemical function *in cellulo* and *in vitro*, which is highly relevant (see above). On these molecular aspects, we have followed the reviewer's suggestions and have extended the structure-based analyses, molecular dynamic simulations, cell-based and biochemical assays to benchmark, validate and demonstrate that the dataset represents a valuable source of information (see above).

While we agree that it is important to address the cellular function of LTPs, i.e. how they contribute to cellular lipid trafficking, fluxes, metabolism or organelle function, it is beyond the scope of this manuscript. New approaches based on chemical engineering of lipids are emerging (as in André Nadler's group, doi.org/10.1101/2024.05.14.594078) and would be ideal to follow up on these questions in a systematic manner, also based on the resource described here. Following the reviewer's recommendation we have extended our data in the suggested direction, and measured the effect of gain of function of individual LTPs on the total HEK293 lipidome, a significant effort (~260 individual experiments)(see above).

Analytically;

- The manuscript hints that establishing fragmentation rules for lipids is a contribution (lines 301-302). The lipid fragmentation rules have been established for a long time for all the classes described (for some examples from recent references that used the same set of rules)

o <https://doi.org/10.1016/j.xcrm.2021.100407> with all the rules and fragmentation pathways published in the GitHub

<https://github.com/SysMedOs/AdipoAtlasScripts/tree/main/LipidIdentification>

o LipidMatch <https://doi.org/10.1186/s12859-017-1744-3>

o Lipid Blast (<https://doi.org/10.1038/nmeth.2551>)

o MSDIAL <https://doi.org/10.1038/s41587-020-0531-2>

o Lipids Standards Initiative published a list of the diagnostic fragments (neutral losses):

<https://lipidomicstandards.org/lipid-class-specific-fragments/>

As the rules are not novel and have been used in mass spectrometry-based lipidomics since its start more than 20 years ago, the manuscript should refer to the established rules and highlight the novelty they provide. It is unclear from the supplementary methods nor from GitHub which part is novel; therefore, this should be better highlighted.

Our intention was not to imply that we had invented or established new rules. We measured a series of lipid standards and determined their retention times and fragmentation patterns in order to refine the fragmentation rules in the METLIN, SwissLipids and LIPID MAPS databases to suit our analytical and experimental setting. To clear up this misunderstanding, we refer to the well-established mass fragmentation rules and clarify our intentions in the main text on page 4 (first paragraph).

- Unusual lipid species are reported, such as ceramides with carbon chain lengths of 46 and 48 and triglycerides with a total carbon chain length of 32. The manuscript should show the MS1 extracted ion chromatograms (EICs) and MS2 spectra in both cases and the rules used to verify these unusual

species. For example, what are the sphingoid base and N-acyl chain lengths for that very long ceramide? No ceramide of that length has been reported in HEK 293 cells, and a citation is missing for its previous report. Moreover, the very short triglyceride with 32 carbons in the chains must contain medium-chain fatty acids, which were not reported earlier (nor cited by the current manuscript) in HEK 293 cells under normal growth conditions.

We present the EICs in a new section of the Supplementary Methods called “Unusual ceramide and triacylglycerol species” pages 4 and 5. In a new Extended Data Fig. 5b, we show the fragmentation spectra of ceramides with carbon lengths of 46 and 48 in comparison with ceramide standards we measured as controls. The patterns of the ceramides with carbon lengths of 46 and 48 are similar to our ceramide standards with the expected mass shifts. From the spectra it is obvious that different combinations of sphingoid base and N-acyl chain coelute and are co-fragmented. By applying a color code, we unravel these chimeric spectra, also reported in column C of Extended Data Table 4. These species were not observable in the total HEK293 lipidomes. They have, however, been recorded (albeit infrequently) in spectral libraries, suggesting that they are rare or scarce. This is presented in new Extended Data Fig. 5c.

The composition of the TAG(32:0) is TAG(12:0/12:0/8:0) is reported in column C of Extended Data Table 4 and the corresponding EICs and fragmentation spectrum is now on pages 4 and 5 of the Supplementary Methods.

- The manuscript refers mainly to the sum composition level annotation of lipids. However, based on the fragmentation rules and the LC-MS method used, the authors should be able to identify lipids at the species level with the individual chain composition. It is not clear why the manuscript ignores that level of structural granularity. Especially in the case of sphingolipids, the manuscript seems to ignore the diversity on the sphingoid base, N-acyl combinations missing an opportunity to discover novel specificities (or proving the lack thereof).

We have indeed identified lipids at the species level (Extended Data Table 4). We have fragmentation spectra for the majority of the abundant lipid species (e.g. PC(34:2)).

In terms of individual fatty acid distribution (e.g. PC(18:1/16:1) or PC(18:2/16:0)), we observed coelution (or overlapping elution) of species with the same sum compositions (due to their chemical similarity and resulting similar retention times). As a result, we measured chimeric spectra, as for ceramides with carbon chain lengths of 46 and 48 (see above). Due to our top-N approach, we do not cover the elution profile of each EIC with multiple spectra and, as a result, may miss the identification of individual fatty acid distributions. For all these reasons (difficulties in interpreting chimeric spectra and the limited number of spectra for the same m/z), we have decided to report the sum composition of each species and, at the level of individual chain composition, the most abundant distributions per spectrum in the column C of Extended Data Table 4.

In Fig. 5 (old Fig. 2) we refer to species as the sum composition of the acyl chain(s), because representing individual chain composition would make the graph unreadable (due to the large number of entries on the x-axis) and the main message less visible. For similar reasons, on Figs. 6a, we display the sum composition of the sphingoid base plus N-acyl chain of sphingolipids. But to help the reader, we added information as to the size of the acyl chains in Fig. 6a.

- Normalization (min-max or sum) is performed, ignoring the fact that ionization efficiency is different among different lipid classes and follows certain trends in long vs. shorter acyl chain lengths. This leads to a situation where the same number of molecules of different lipid classes (or species) will generate significantly different numbers of ions (reflected in intensity levels or areas under the peaks for the same amount of different lipids. Considering this, it is unclear how the manuscript addresses this to

comment on specific lipid cargo of different classes bound to the same LTP. This might directly affect some interpretations as it penalized lipids with low ionization efficiency or abundance. A possible suggestion is to use ionization efficiency factors (calibration curve slopes) from pure standards spiked to a common matrix, normalizing to protein abundance and showing the intensity distribution of all lipids detected in the same fraction detected for some LTPs.

We were aware of this limitation. However, absolute quantification (using spiked in standards) was beyond the scope of this manuscript. We were aware that the effect of the ionization efficiency and the methodical setup does not allow us to make quantitative statements comparing different lipid classes. For example, we cannot assert from our data that PC is more or less abundant than PE in the STARD10 pull down. We therefore refrained from comparing the abundance of different lipid classes. When we compare the quantities of lipid species with each other (Figs. 6a and 6c), we also provide a reference to the abundance of these same species in the total lipidome; we display the enrichments.

This said, we now realize that the heat map in Fig. 2a may be misleading to non-expert readers, who may be tempted to over-interpret the data and compare different lipid classes. To remedy this, we have added a comment on how to use these data in the legend to Fig. 2a.

Referee #1 (Remarks to the Author):

While this study has been improved, my concerns remain the same. The revised manuscript now more clearly explains what the major advances of the study are and more carefully describes the scope and limits of the results. However, the study still does not provide the major conceptual insights expected of paper in Nature. The rebuttal says, "... the conceptual advances are at molecular and biochemical levels, and not so much at the cellular ones." This is reasonable but the most important questions about LTPs are how they function in cells. Understanding which lipids are bound by most of the LTPs in humans is certainly important, but, after years of work on LTPs, the primary challenge remains understanding how LTPs function in cells and integrate with cellular lipid homeostasis. Knowledge of which lipids they bind is critical but, by itself, is not a game changer. The study remains an important resource for those who want to understand how LTPs function.

We agree that understanding how LTPs function in cells, how their activities integrate with the cellular state of membranes or lipidomes, remains a challenge. We believe that to meet this challenge, it is necessary, among other things, to understand how LTPs function molecularly. They are sophisticated lipid mobilization devices. They mobilize not only their cargo, but also auxiliary lipids that serve as currencies or cofactors facilitating the uptake or release of cargo into distinct membranes. This is directly relevant to their cellular function, as it ensures the directionality of transport and its coupling to metabolism (10.1038/s41580-018-0071-5; 10.1016/j.cell.2010.12.034; 10.1016/j.cell.2013.09.056; 10.1126/science.aab1370; 10.1016/j.cell.2013.09.056; 10.1126/science.aab1370; 10.1126/science.aab1346; 10.1126/science.1233508; 10.1021/acs.jpcc.4c02398; 10.1016/j.tibs.2009.10.008; 10.1371/journal.pone.0101550). However, in most cases, the identity of the cargoes and auxiliary lipids remains unknown, and this gap limits our ability to understand how LTPs function in cells. In this context, we believe that the work represents an important resource, also acknowledged by reviewer 4, "*I am confident the communities of lipid cell biology, lipid metabolism and trafficking, and lipidomics will read the manuscript with great interest and will build several projects on its findings.*"

We clarified these notions in the introduction on page 2 (last paragraph) and page 3 (first paragraph). We also rewrote the conclusion (starting on page 13), suggesting how we believe the data could be used to advance our understanding of LTPs function in cellular context.

"Understanding how LTPs function in cells, how their activities can adapt to the state of membranes and lipidomes, remains a challenge that requires, among other things, understanding how LTPs operate at the molecular level. The capacity of LTPs to mobilize membrane lipids is essential to their cellular function, as this involves specific cargoes, but also regulatory auxiliary lipids. The widespread ability to bind to multiple classes of lipids suggests that these lipid-induced regulatory mechanisms are common. However, how a single LTP selectively mobilizes lipids of diverse structure remains poorly understood, as do the consequences of these interactions on the metabolic fate of the cargo. By defining individual LTP-lipid and lipid-lipid pairings, our work provides a foundation for future biophysical, molecular dynamics simulation and cell biology studies aimed at addressing these questions^{11,18} and should motivate the extension of these approaches, for example, to different cell types or states.

We demonstrate the feasibility of systematic studies of human LTPs, illustrate how we can integrate large-scale data and adapt concepts from systems biology to LTPs. We have just begun to study the consequences of these interactions in a cellular context, by analyzing the impact of LTP gain-of-function on cellular lipidomes. The study of the biological functions of LTPs, through systematic exploration of the consequences of their perturbation on organelle function, lipid trafficking, and metabolic fate, is the next challenge to be addressed, for which we believe this resource will serve as a basis."

Referee #2 (Remarks to the Author):

In the revised manuscript by Titeca, Chiapparino, Hennrich and coworkers, the authors have added considerable new experimental and simulation data to support their claims. They have now provided additional experimental evidence from a cell-based system as a functional benchmark. Specifically, they measured how the lipidome of HEK293 cells is affected by overexpression of the individual LTPs studied. They found that the abundance of almost half (44%) of the lipid ligands identified was affected by overexpression of their respective LTP. This is significantly higher than what is observed when all possible LTP-lipid pairs are considered. Yet, in some cases, LTP overexpression increased and in other cases decreased the amount of the ligand identified. This is specifically mentioned in the few exemplary cases discussed more thoroughly. For instance, overexpression of SEC14L2 resulted in a decrease in the identified substrate diacylglycerol, whereas overexpression of HSDL2 resulted in an increase in the identified substrate triacylglycerol. The authors have provided a new Extended Data Table 8, which summarizes in which cases significant changes in the abundance of the expected ligand were observed. This Table should also include information on whether this change was an increase or a decrease. In addition, it remains unclear if significant changes in the levels of other lipids than the expected ligands were observed upon LTP overexpressions at the whole lipidome level. If such changes were observed, it would be helpful to include the information which additional lipid species were up- or downregulated, to facilitate follow-up studies.

Following the reviewer's suggestion, we have added information regarding the direction of the observed changes to Extended Data Table 8. Information concerning all lipids affected by each overexpressed LTPs (including the fold changes and p-values) was available in Extended Data Table 7C. For clarity, we have now highlighted significant positive and negative changes in blue and green, respectively. This should make it easier to navigate this table.

According to the authors, the subcellular colocalization data that they use and refer to, cannot be made publicly available at this stage. Instead, they have listed the colocalization data for all lipid pairs in a new Extended Data Table 9A. This is unfortunately not very useful. However, according to the authors another manuscript describing these data in a more comprehensible manner will apparently be made public soon, and should remedy this.

The manuscript by Kenji Maeda's group has been submitted to BioRxiv. We have added the link to the submission (<https://doi.org/10.1101/2025.10.05.680593>) in the "Data and Code Availability Statement" section (page 19). We also quote this paper in the main text (reference 36) and the methods section (reference 26).

The experimental work is now also supported by atomic-level simulation results. Their role in the manuscript is not decisive, but they provide some additional insight into how LTPs bind certain lipids. The basis of these simulations is the same force field (CHARMM36m), which has been appropriately used in all cases studied, so the results are comparable. However, there are four key problems and weaknesses in the simulation section, as follows:

- 1) The description of the methods gives the impression that simulations have been performed primarily only once for each system (STARD2 being an exception with 3 repeats). This does not meet quality standards (for comparison, would it be acceptable for a wet-lab experiment to be performed only once). Each system should be simulated at least three times over a sufficient time scale, producing independent samples.

We thank the reviewer for giving us the opportunity to clarify this point. Replicates were run for most of the simulations reported, with the exception of STARD2-PC-O. All reported simulations have now been run in triplicate. We address these points in the SI Methods: we have added a paragraph (on

page 18, first paragraph) devoted to these topics and we provide a list of all simulations in a new Table 2 on page 18, which also contains information on the composition of the simulated systems and the duration of the simulations.

2) The manuscript implies that the LTPs studied in the simulations (e.g., STARD2) mobilize lipids and that mobilization affects both ester and ether lipids. However, mobilization is not studied in the simulations. The simulations focus only on lipid-protein interactions within the LTP cavity. Does lipid movement occur between the membrane and the LTP? To do this, one would need to carry out a massive set of simulations to observe spontaneous loading/unloading events, but doing this is possibly not feasible. Alternatively, one would need to determine the free energy profiles of loading/unloading by first pulling a lipid out of the LTP into the membrane, which would also generate a transition path, and using it, implement the corresponding loading process from the membrane to the LTP. These simulations would reveal whether the free energy barrier for lipid loading to the carrier protein is low enough in relation to the thermal energy that the mobilization is possible; and whether the ester vs. ether bond affects this process, and in other cases whether the head group/acyl chain affects the mobilization. Without this information, the simulations described in the manuscript do not tell us anything other than the ability of certain lipids to remain inside the LTP for (less than) a microsecond.

We have carefully considered the most appropriate computational approaches to provide data useful to validate several of the novel LTP-ligand complexes. We reasoned (for the reasons detailed below) that the most reliable approach was to evaluate whether the experimentally proposed LTP-lipid pairs would form thermodynamically stable complexes, which is a necessary condition for the extraction of lipid ligands from membranes to be productive. We thus focused on the bound complexes and set to identify enthalpically relevant interactions between LTPs and (known or novel) ligands, as a means to provide relevant information on the basis for ligand selectivity. This was done for LTPs and cargos for which we could formulate a hypothesis about their mode of interaction, based on chemical intuition and visual scrutiny of the protein structures and sequences. They also provide structural models in the absence of available experimental (e.g. X-ray) data from the Protein Data Bank. We clarify this on page 18, first paragraph of the SI Methods section.

We agree that simulating extraction itself is interesting and represents a kind of Holy Grail, which would allow us to characterize the molecular mechanisms of uptake. However, this remains challenging and goes beyond the scope of this manuscript. We have so far observed spontaneous uptake of known cargos only in STARD2 simulations (Talandasthi, *J Phys Chem Lett*, 2024), while no spontaneous cargo extraction has been observed in several microseconds-long simulations of STARD11 (Moqadam, *J Phys Chem B*, 2024) or STARD4 (Talandashti, *J Mol Biol*, 2024). We consider that these approaches were not tractable for validating several of the new ligands identified in this study.

The second approach suggested by the reviewer involves using, e.g. potential of mean force (PMF) to calculate the free energy required to desorb the cargo from the membrane and to bind it in the LTP pocket. These calculations are not intended to be used systematically to study multiple LTPs, each likely using different cargo extraction mechanisms. They are resource intensive and hard to implement, as they are highly sensitive to experimental settings, e.g. the composition of the membrane model (when we have little information about the lipid environment in which LTPs function). Most important, they require complex collective variables needed to correctly identify the transition state and obtain a reliable estimate of the free energy barrier (see the excellent discussions in the works by Julia Rogers and Phillip L. Geisler (doi: 10.1371/journal.pcbi.1010992, doi: 10.1021/acs.jpcc.0c04139, doi: 10.1016/j.bpj.2021.07.016)). These approaches are not suitable for validating several new protein-ligand pairs at the scale done in this study.

We realized that we have overstated the scope of the MDS on page 7 (last paragraph) of the main text and have now reformulated the following sentence:

*“Molecular simulations supported the notion that STARD2 can **mobilize** equally ether- and ester-phosphatidylcholine” (Extended Data Fig. 4a),...*”

to

*“Molecular simulations supported the notion that STARD2 can **bind** to both ether- and ester-phosphatidylcholine (Extended Data Fig. 4a),...”*

3) Continuing from the previous point, how do lipids stabilize LTPs? If the observed lipids are ligands for these LTPs, then the presence of lipids inside the LTP stabilizes the protein structure. What does the RMSD/RMSF data tell?

We thank the reviewer for this suggestion, which we have followed. We would like to underline though that the presence of a ligand in a protein structure might not always lead to less structural fluctuations, for example if the protein’s function requires a certain level of flexibility or if the ligand leads to increased exposure of hydrophobic regions. Yet RMSF provides a good indication of the stability of MD simulations and as such is a valuable measure to report.

We extracted the root mean square fluctuations (RMSF) profiles for LCN1, the STARD domain of CERT and STARD2 complexed with their previously known or novel ligands. The data are reported in a new Extended Data Fig. 4b for STARD2, and a new Figure 3 on page 22 of the SI Methods section for the other proteins. The known ligands of CERT and STARD2 (ceramide and phosphatidylcholine, respectively) indeed caused a decrease in RMSF (indicating stabilization), particularly noticeable in the gate region ($\Omega 1$ and $\Omega 4\alpha 4$). The new ligands, ether phosphatidylcholine and long DH-ceramide, respectively, had similar effects as the previously known ones. This is described on pages 7 (last paragraph) of the main text.

In the case of LCN1, the known, phosphatidylcholine (POPC) and novel, sphingomyelin (SM) cargos also had the same effect, but they tend to increase the RMSF of LCN1. This is due to the high flexibility of several of the LCN1 loops which are long and unstructured (e.g. $\beta 1\beta 2$, $\beta 5\beta 6$ and $\beta 6\beta 7$, $\beta 8\beta 9$). The high flexibility is corroborated by (1) the large structural difference that exists between the 20 conformations experimentally resolved by NMR (PDB ID 5T43) between which the backbone RMSD reaches up to 2 Å, and (2) the large standard error in the RMSF calculated for the three replicates of the apo simulations (see blue shades in Figure 3d on page 22 of the SI Methods section).

4) The simulation data provide valuable information about how LTPs could specifically bind a particular lipid head group. For instance, the manuscript reports that in LCN1 the residues W17 and Y97 would be critical for this binding with the PC head group. The immediate question is, if these residues are mutated (in simulations), will the specific binding be lost?

We thank the reviewer for their suggestion. We performed 500-ns long MD simulations of a phosphatidylcholine (PC)-loaded LCN1 variant where W17 and Y97 were substituted by alanine (see W17A/Y97A in Table 2, on page 19 of the SI Methods section). In this case, the PC was no longer stably bound in the pocket, but rapidly left its binding site in the pocket, failing to establish enthalpically favorable interactions within the pocket, interactions which are known to play a role in specificity. This is visualized in Extended Data Fig. 5b where we report the distance between the PC choline group and amino acids 17 and 97. The starting distances of the phosphocholine head group of POPC to these two residues are ca. 6 Å. In LCN1^{W17A/Y97A}, these distances increase rapidly, indicating movements of the choline head group. In contrast, in LCN1^{wt}, the head group remained ca. 6 Å from W17 and Y97 during

the 500 ns simulation indicating two cation- π interactions. This is described on page 8 (2nd paragraph) of the main text.

Referee #3 (Remarks to the Author):

In the revised manuscript, the authors introduced structural and functional benchmarks to better assess the quality and functional relevance of their system-wide analysis of LTPs to mobilize lipids. As structural benchmark, they determined to what extent the sizes of lipid ligands identified in their assays fitted in the volumes of lipid-binding pockets estimated from known or predicted LTP structures. This led them to uncover the existence of a non-occupied buffer zone, analogous to that previously reported for the ligand-binding cavity of enzymes. Additionally, they found that the volumes occupied by newly identified LTP ligands was similar to those of known ligands. As functional benchmark, they used LC-MS/MS to measure how the lipidome in HEK293 cells was affected by overexpression of individual LTPs. This revealed that cellular levels of newly identified lipid ligands were as frequently affected as those of previously known lipid ligands. These benchmarks are valuable additions to the study as they underscore the quality and functional relevance of the reported systematic analysis of lipid-binding properties of 39 LTPs. As outlined below, the revised manuscript also raises some questions and concerns, which I believe the authors should be able to address w/o additional experimental work.

1) For the functional benchmark, the authors found that a significant fraction of the ligands identified in the *in vitro* assay were supported by overexpression data (36%). This is lower than for the *in cellulo* assay (58%) but still significantly higher than what is observed when all possible LTP-lipid pairs are considered (22%; Fig. 1d), indicating that both the *in vitro* and *in cellulo* approaches yield functionally relevant LTP-ligand interactions. It would be relevant to know how the *in cellulo* and *in vitro* data sets compare with respect to the structural benchmark. This comparison is lacking from the revised manuscript.

We have added an Extended Data Fig. 2c-d that shows how the *in cellulo* and *in vitro* datasets compare in the structural benchmark. We excluded from this analysis the three outliers LTPs, LCN15, SCP2D1, and SCP2 with a particularly small pocket or their entire families. The comparison shows that the volumes occupied by the ligands *in vitro* were comparable to those *in cellulo* with p-value = 0.063 (if we exclude the three outliers) and 0.875 (if we exclude their entire families) (new Extended Data Fig. 2c-d). The description is on page 5 (third paragraph) of the main text.

2) PC was identified as putative novel lipid cargo of CERT in the *in cellulo* assay. Using a fluorescence emission assay, the authors provide experimental evidence that CERT can bind PC *in vitro*. Previous *in silico* simulations showed that PC may act as a cofactor to facilitate ceramide release (Moqadam et al., 2024; Ref. 10), providing additional proof that PC is an authentic CERT binding partner. However, whether CERT mediates PC transfer remains to be established. The observed decrease in cellular PC levels upon CERT overexpression (Fig. 2b) does not necessarily reflect the physiological relevance of the observed PC-CERT interaction (as suggested by the authors on p. 10 lines 257-260), but rather a consequence of increased consumption of PC as headgroup donor in sphingomyelin production.

The reviewer is right. We demonstrate that CERT can bind phosphatidylcholine and MD simulations support the notion that this is important for ceramide release (Moqadam et al., 2024, J Phys Chem B 128). We agree that the mobilization of phosphatidylcholine by CERT does not necessarily entail phosphatidylcholine transport. Our hypothesis is rather that phosphatidylcholine acts as a cofactor, facilitating ceramide release at places where it will be metabolized. This is consistent with the observed decrease in cellular PC and increase in DAG/SM upon CERT overexpression and reflects the physiological relevance of the observed PC-CERT interaction. We have rewritten this part to include the reviewer's suggestion and this hypothesis on page 10 (first paragraph) of the main text:

“Furthermore, overexpression of CERT resulted in a significant increase in sphingomyelin and diacylglycerol (both produced by the transfer of the phosphocholine head group from phosphatidylcholine to ceramide) and a significant decrease in phosphatidylcholine and triacylglycerol species (Fig. 2b and Extended Data Table 7C). Mobilization of phosphatidylcholine and triacylglycerol by CERT does not necessarily imply their transport, but may instead facilitate ceramide uptake/release¹¹. These metabolites are involved in the further conversion of ceramide^{37,38}. Their mobilization by CERT could couple the transport of toxic ceramide with its conversion into sphingomyelin³⁷ - or perhaps into a form of ceramide storage in lipid droplets (acylceramide)³⁸.”

3) Identification of TAG as putative novel lipid cargo of CERT is striking. This finding raises a series of questions, for instance how TAG would be accommodated in a lipid-binding pocket tuned for ceramide, a lipid with a small hydrophilic headgroup. Also, how would CERT gain access to and extract TAG from cellular membranes and how can a drop in cellular TAG levels upon CERT overexpression be explained in the context of an accelerated sphingomyelin production. I do not expect the authors to answer these questions. However, it would be helpful if they add at least a bit of context to their identification of TAG as novel lipid binding partner of CERT given that they explicitly refer to this discovery multiple times in the manuscript.

Following the reviewer’s suggestion, we have added some context and hypotheses to the identification of triacylglycerol as a novel binding partner to CERT. As with phosphatidylcholine, the binding of CERT to triacylglycerol does not necessarily imply the transport of triacylglycerol. Triacylglycerol could also act as a cofactor facilitating the release of toxic ceramide at the sites where it will be metabolized, perhaps into a storage form of ceramide (acylceramide). The capacity to be selective for ligands with large structural diversity seem to be a relatively common attribute of LTPs (not only CERT), but how this is achieved remains unclear. We believe that these data will lead to further molecular dynamics simulations aimed at elucidating the mechanism by which CERT can mobilize triacylglycerol. Due to space constraints and to avoid excessive speculation, we have merged the discussion on triacylglycerol with that on phosphatidylcholine on page 10 (first paragraph) of the main text (see point 2):

“Furthermore, overexpression of CERT resulted in a significant increase in sphingomyelin and diacylglycerol (both produced by the transfer of the phosphocholine head group from phosphatidylcholine to ceramide) and a significant decrease in phosphatidylcholine and triacylglycerol species (Fig. 2b and Extended Data Table 7C). Mobilization of phosphatidylcholine and triacylglycerol by CERT does not necessarily imply their transport, but may instead facilitate ceramide uptake/release¹¹. These metabolites are involved in the further conversion of ceramide^{37,38}. Their mobilization by CERT could couple the transport of toxic ceramide with its conversion into sphingomyelin³⁷ - or perhaps into a form of ceramide storage in lipid droplets (acylceramide)³⁸.”

4) In the Introduction (p. 2, line 51) and Discussion (p. 13, line 352) the authors refer to the crucial but largely elusive membrane-induced mechanisms that ensure vectorial transport of lipids. What mechanisms are they referring to? And how do they believe their system-wide analysis of LTP-lipid pairs will help advance our understanding of how lipids are mobilized inside cells. The manuscript leaves some room for improvement to stipulate this in more detail.

We refer to the ability of LTPs to manipulate lipids to perform vectorial transport, i.e. coordinated uptake and release. LTPs mobilize not only their cargo, but also auxiliary lipids that serve as currencies or cofactors facilitating the uptake/release of cargo into distinct membranes. This is directly relevant to their cellular function and ensures the directionality of transport and its coupling to metabolism (10.1038/s41580-018-0071-5; 10.1016/j.cell.2010.12.034; 10.1016/j.cell.2013.09.056; 10.1126/science.aab1370; 10.1016/j.cell.2013.09.056; 10.1126/science.aab1370; 10.1126/science.aab1346; 10.1126/science.1233508; 10.1021/acs.jpcc.4c02398;

10.1016/j.tibs.2009.10.008; 10.1371/journal.pone.0101550). We clarified this point on page 2 (last paragraph) and page 3 (first paragraph) of the introduction.

We also rewrote the conclusion (starting on page 13) and explicitly state how we believe the data could be exploited to advance our understanding of LTPs function in cellular context.

“Understanding how LTPs function in cells, how their activities can adapt to the state of membranes and lipidomes, remains a challenge that requires, among other things, understanding how LTPs operate at the molecular level. The capacity of LTPs to mobilize membrane lipids is essential to their cellular function, as this involves specific cargoes, but also regulatory auxiliary lipids. The widespread ability to bind to multiple classes of lipids suggests that these lipid-induced regulatory mechanisms are common. However, how a single LTP selectively mobilizes lipids of diverse structure remains poorly understood, as do the consequences of these interactions on the metabolic fate of the cargo. By defining individual LTP-lipid and lipid-lipid pairings, our work provides a foundation for future biophysical, molecular dynamics simulation and cell biology studies aimed at addressing these questions^{11,18} and should motivate the extension of these approaches, for example, to different cell types or states.

We demonstrate the feasibility of systematic studies of human LTPs, illustrate how we can integrate large-scale data and adapt concepts from systems biology to LTPs. We have just begun to study the consequences of these interactions in a cellular context, by analyzing the impact of LTP gain-of-function on cellular lipidomes. The study of the biological functions of LTPs, through systematic exploration of the consequences of their perturbation on organelle function, lipid trafficking, and metabolic fate, is the next challenge to be addressed, for which we believe this resource will serve as a basis.”

Referee #4 (Remarks to the Author):

The authors have performed a major revision of the original manuscript, adding several experiments, MD simulations, and revising the text accordingly. In this revision, they have satisfactorily addressed my concerns by incorporating a substantial amount of orthogonal data, such as HEK overexpression lipidomics data and MD simulations, which adds another dimension to the novelty and generalizability of some of their findings. They also addressed concerns regarding the mass spectrometry method, highlighting unusual species (e.g., very long-chain spingoid bases containing ceramides). They also addressed several concrete examples that showcase the potential impact of the manuscript as a resource. Although the concern regarding the cellular lipid microenvironment and its effect on LTP functions has not been addressed, the BioRxiv manuscript they refer to in the rebuttal probably complements this aspect. Reading the revised manuscript was a pleasure, and I am confident the communities of lipid cell biology, lipid metabolism and trafficking, and lipidomics will read the manuscript with great interest and will build several projects on its findings.

We thank the reviewer for the very positive review and for finding the project an important resource on which the community will build new projects.

Referee #5 (Remarks to the Author):

I co-reviewed this manuscript with one of the reviewers who provided the listed reports.

Referee #5 (Remarks on code availability):

The simulations were performed using the NAMD software package (<https://pubmed.ncbi.nlm.nih.gov/32752662/>), developed and maintained by the Tajkhorshid group at the University of Illinois at Urbana-Champaign. It is an open source, free-of-charge simulation code focused on biological molecular systems. The current version was published in 2020 and has been cited approximately 2600 times (Google Scholar), indicating its widespread use.

We thank the reviewer for their time and comments. We confirm that the simulations were performed using the open source software NAMD, a widely used software program.

Geneva, November 25th 2025

Dear _____,

We thank you for your time and effort with our manuscript. We addressed all editorial requests and confirm that we did not removed peer reviewed data making these changes. All changes are clearly visible in track change mode or highlighted in grey in the main text.

- All claims related to lipid transfer are toned down including the sentence in the abstract: "*We demonstrate some basic principles of how lipid transfer proteins work*", now changed to "*We report some basic principles of how lipid transfer proteins work*".

We systematically checked for the terms "*transfer*" / "*transport*" or "*cargoes*": and could identify five (page 3, first paragraph; page 7, 2nd paragraph; page 8, 2nd paragraph; page 13, 1st paragraph), which we have removed or replaced by "*mobilization*" or "*ligands*", respectively. All other cases were instances where these terms were used to: i) define acronyms (i.e. LTP), or to name proteins (i.e. class I phosphatidylinositol transfer proteins (PITPs)) or ii) refer to previous knowledge/literature (i.e. the protein X is known to transfer lipid Y).

- We confirm that we have not removed peer reviewed data when revising the manuscript.

- We have completed the six forms and uploaded them with the submission as "Related Manuscript Files".

- We addressed queries in the file "NATURE_email_attachment_8069167_1762971263_86", please see below.

We thank you and all reviewers for the efforts and contribution to this manuscript that has significantly improved.

Yours sincerely,

Anne-Claude on behalf of the authors